# Repulsive Latent Score Distillation for Solving Inverse Problems

**Nicolas Zilberstein**
Rice University
nzilberstein@rice.edu

**Morteza Mardani**\*
NVIDIA Inc.
mmardani@nvidia.com

**Santiago Segarra**\*
Rice University
segarra@rice.edu

## Abstract

Score Distillation Sampling (SDS) has been pivotal for leveraging pre-trained diffusion models in downstream tasks such as inverse problems, but it faces two major challenges: $(i)$ mode collapse and $(ii)$ latent space inversion, which become more pronounced in high-dimensional data. To address mode collapse, we introduce a novel variational framework for posterior sampling. Utilizing the Wasserstein gradient flow interpretation of SDS, we propose a multimodal variational approximation with a *repulsion* mechanism that promotes diversity among particles by penalizing pairwise kernel-based similarity. This repulsion acts as a simple regularizer, encouraging a more diverse set of solutions. To mitigate latent space ambiguity, we extend this framework with an *augmented* variational distribution that disentangles the latent and data. This repulsive augmented formulation balances computational efficiency, quality, and diversity. Extensive experiments on linear and nonlinear inverse tasks with high-resolution images ($512 \times 512$) using pre-trained Stable Diffusion models demonstrate the effectiveness of our approach. The code is available at GitHub.

## 1 Introduction

Diffusion models have recently achieved remarkable success in visual domains. A key application of these models is solving various inverse problems in a *plug-and-play* manner, where diffusion models act as *rich priors* to regularize the search space, ensuring the generation of plausible solutions. Variational samplers (Poole et al., 2022; Mardani et al., 2024) approach sampling as an optimization problem, providing a high degree of control and fidelity in generation. However, they encounter two significant challenges, particularly when dealing with high-dimensional data that requires diverse outputs: ($c1$) *mode collapse*, and ($c2$) *inversion of the latent space*, such as that seen in the adversarial autoencoder of Stable Diffusion (Rombach et al., 2021).

There have been a few recent attempts to address these challenges separately in the context of text-to-image/3D generation and inverse problems. To mitigate ($c1$), for text-to-3D generation, ProlificDreamer (VSD) (Wang et al., 2024) introduces data-driven dispersion with independent particles. Still, the combination of independence and unimodal approximation per particle renders an optimization that collapses to the same local minimum, limiting diversity. Collaborative Score Distillation (CSD) (Kim et al., 2023) seeks to diversify the variational approximation using Stein Variational Gradient Descent (SVGD) (Liu and Wang, 2016), but smoothing particle gradients with SVGD is problematic in high-dimensional spaces (D'Angelo and Fortuin, 2021a; Ba et al., 2021). To address ($c2$), recent samplers using latent diffusion models (Rout et al., 2024; Chung et al., 2024; Song et al., 2024) remain computationally demanding, similar to earlier pixel-based methods (Chung et al., 2022a; Song et al., 2022), due to multiple correction steps required for deviations from the image manifold, a challenge arising from adversarial training of autoencoders, akin to GAN inversion (Xia et al., 2022; Daras et al., 2021). Thus, no current solution effectively handles both mode collapse and latent space issues in inverse problems.

---

\*Equal advising.

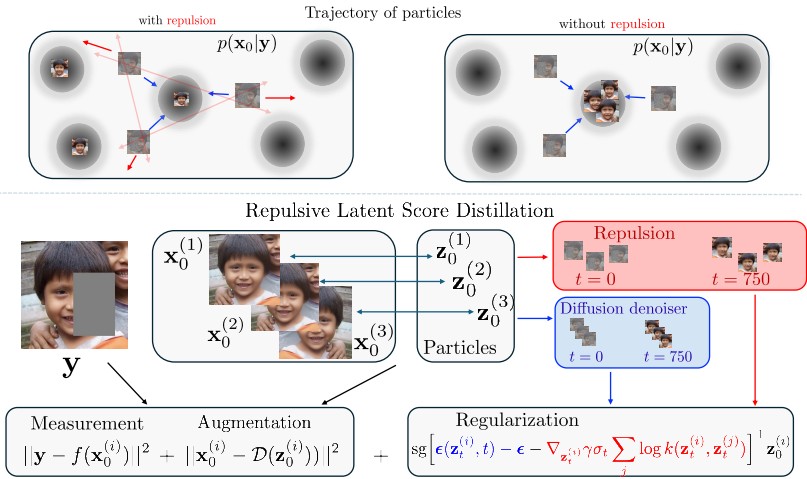

Figure 1: Illustration of Repulsive Latent Score Distillation (RLSD): It propagates a set of particles by adding noise and applying two levels of regularization: ($i$) **Denoising**, via score-matching regularization, which directs particles toward modes of the distribution $p(\mathbf{x}_0|\mathbf{y})$ (blue arrows); and ($ii$) **Repulsion**, which pushes particles apart (red arrows) to explore other regions of the posterior density. During sampling, the repulsion gradient ensures particles remain separated, leading to different modes, as shown in the upper-right box.

We hypothesize that the primary issue with ($c1$) arises from collapse in high-dimensional spaces. To address this, we employ an ensemble of interactive particles with repulsion to prevent collapse. Inspired by kernel-density estimation (D'Angelo and Fortuin, 2021b), we introduce a particle-based multimodal variational approximation that incorporates repulsive forces. These forces are defined through pairwise interactions based on similarity, such as using a radial basis kernel of DINO features (Caron et al., 2021).

To address ($c2$), we propose a variational augmented distribution that jointly optimizes the latent and data variables, similar to half-quadratic splitting (Geman and Yang, 1995). This method disentangles the latent (prior) from the data (measurement), yielding solutions with sharper details. We show that KL minimization of our augmented interactive particle approximation leads to score-matching regularization with two gradient terms: ($a$) denoising regularization along the entire diffused trajectory, and ($b$) repulsion regularization to encourage diversity in the latent diffusion's trajectory; see Fig. 1. We refer to our method as *Repulsive Latent Score Distillation* (RLSD).

We validate the advantages of RLSD through extensive experiments on both linear and nonlinear tasks, using Stable Diffusion as the prior. In diversity-critical cases such as inpainting and phase retrieval, our method provides a solid trade-off between diversity and quality. For tasks where diversity is less essential, like deblurring, our augmented formulation offers a fast solver by avoiding score Jacobian computations and performs efficiently on high-resolution ($512 \times 512$) images. Overall, RLSD combines the strengths of variational samplers (memory and compute efficiency) and posterior samplers (diversity), enabling control over speed and diversity by adjusting the scalar weights between denoising and repulsion regularizations. A detailed comparison of RLSD's properties is summarized in Table 1.

All in all, the main contributions of this paper are summarized as follows:

- We propose **Repulsive Latent Score Distillation** (RLSD), a variational posterior sampler for general inverse problems with high-resolution images (e.g., $512 \times 512$), that *trades-off diversity for quality* in a controllable fashion simply via regularization weights.
- We introduce a *repulsion regularization* to boost the diversity via an interactive particle-based variational approximation inspired by Wasserstein gradient flow.
- To handle the latent space inversion, we propose a *distribution augmentation* that decouples the latent and pixel space, rendering a two-step optimization problem.
- We perform extensive experiments for various (non)linear inverse tasks using Stable Diffusion. The results indicate the superior performance of RLSD over existing alternatives such as PLSD (Rout et al., 2024), DPS (Chung et al., 2022a) and RED-Diff (Mardani et al., 2024).

|  | Posterior samplers | | Variational samplers | | | |
| --- | --- | --- | --- | --- | --- | --- |
|  | DPS | PSLD | RED-Diff | VSD | CSD | **RLSD (Ours)** |
| Diversity (Low-dim) | ✓ | ✓ | ✗ | ✓ | ✓ | ✓ |
| Diversity (High-dim) | ✓ | ✓ | ✗ | ✗ | ✗ | ✓ |
| Latent Diffusion | ✗ | ✓ | ✗ | ✓ | ✓ | ✓ |
| Linear Inv. Problems | ✓ | ✓ | ✓ | - | - | ✓ |
| Nonlinear Inv. Problems | ✓ | - | ✓ | - | - | ✓ |
| No Score Jacobian | ✗ | ✗ | ✓ | ✓ | ✓ | ✓ |

Table 1: Comparison of our work with DPS, RED-Diff, PSLD, VSD, and CSD. Our method combines the strengths of variational samplers (no score Jacobian, computational efficiency) and posterior sampling algorithms (diversity at high dimensions). In addition, our formulation enables us to solve nonlinear inverse problems at $512 \times 512$.

## 2 RELATED WORKS

This paper is primarily related to diffusion models at its core, and two related lines of work: inverse problems and score distillation sampling.

**Diffusion models for inverse problems.** Several works have used diffusion models as priors to solve inverse problems in various domains (Daras et al., 2024; Kong et al., 2020). A recent approach termed RED-Diff (Mardani et al., 2024) uses variational inference for solving inverse problems with diffusion priors, similar to plug-and-play methods (Venkatakrishnan et al., 2013); see also Zhu et al. (2023); Zhang et al. (2021). This method employs the diffusion model as denoisers at different scales, akin to the RED framework (Romano et al., 2017). Despite successfully balancing quality and runtime, it suffers from mode collapse due to the unimodal approximation. Furthermore, optimizing directly in the pixel domain restricts the ability to leverage latent diffusion models like Stable Diffusion (Rombach et al., 2021) for solving inverse problems at high-resolution. Recent works incorporate latent diffusion models as prior (Rout et al., 2024; Chung et al., 2024; Song et al., 2024). While they partially alleviate the computational demands of pixel-domain solvers, they introduce additional steps to correct the deviations from the image manifold, which arise from the adversarial training of the autoencoder. Thus, it is still an open problem to develop methods that are fast, promote diversity, and optimize in the latent space of the diffusion model.

**Score distillation: diversity and mode collapse.** Recently, SDS enabled the use of pretrained diffusion models for text-to-3D generation (Poole et al., 2022). Although SDS provides an efficient mechanism for the aforementioned task, it often suffers from mode collapse and saturated images; see details about SDS and its formulation in Appendix E. ProlificDreamer (Wang et al., 2024) aims to fix the mode collapse using a data-driven dispersion fine-tuned at each iteration via LoRA (Hu et al., 2021). It is, however, costly, and the independence of particles hinders diversity. Recently, the authors in (Kim et al., 2023) propose to use the well-known Stein variational gradient descent (SVGD) as an update direction, which yields an interactive particle system. However, it is known that SVGD suffers from the curse of dimensionality (D'Angelo and Fortuin, 2021a). Other related works are Armandpour et al. (2023), where the authors leverage the negative prompt to eliminate undesired perspectives, and Wang et al. (2023), where an entropic regularization is proposed.

## 3 BACKGROUND

We review latent diffusion models in Section 3.1, and we briefly discuss how they are incorporated as priors to solve inverse problems in Section 3.2.

### 3.1 DIFFUSION MODELS IN THE LATENT SPACE

Diffusion models (Sohl-Dickstein et al., 2015; Ho et al., 2020; Song et al., 2021b) consist of two processes modeled using stochastic differential equations: 1) a forward process that gradually adds noise to a clean image, and 2) a reverse process that learns to generate images by iteratively denoising the diffused data. In latent diffusion models (Vahdat et al., 2021; Rombach et al., 2021), the data $\mathbf{x}_0$

is encoded into a latent space through an *encoder* $\mathcal{E}(\mathbf{x}_0) = \mathbf{z}_0$, and the forward process follows the variance-preserving SDE (Song et al., 2021b) in the latent space: $d\mathbf{z}_t = -\frac{1}{2}\beta(t)\mathbf{z}_t dt + \sqrt{\beta(t)}d\mathbf{W}_t$, for $t \in [0, T]$. Here, $\beta(t)$ is a function that defines a step size for each $t$ from 0 to $T$, and is defined as $\beta(t) := \beta_{\min} + (\beta_{\max} - \beta_{\min})\frac{t}{T}$, and $\mathbf{W}_t$ is the standard Brownian motion. The forward process is designed in such a way that the distribution of $\mathbf{z}_T$ converges to a standard Gaussian distribution. Given the forward process, the reverse process is defined as $d\mathbf{z}_t = -\frac{1}{2}\beta(t)\mathbf{z}_t dt - \beta(t)\nabla_{\mathbf{z}_t}\log p(\mathbf{z}_t) + \sqrt{\beta(t)}d\mathbf{W}_t$, where $\nabla_{\mathbf{z}_t}\log p(\mathbf{z}_t)$ is the *score function*, which is unknown. To map back to the ambient space, we pass the generated sample $\mathbf{z}_0$ through a *decoder* $\mathcal{D}(\mathbf{z}_0) = \mathbf{x}_0$.

Therefore, to solve the reverse process and use it for sampling, the score function ($\nabla_{\mathbf{z}_t}\log p(\mathbf{z}_t)$), encoder ($\mathcal{E}$), and decoder ($\mathcal{D}$) are learned by minimizing the denoising score-matching loss (Vincent, 2011). Diffused samples are generated as $\mathbf{z}_t = \alpha_t\mathbf{z}_0 + \sigma_t\boldsymbol{\epsilon}$, where $\mathbf{z}_0$ encodes $\mathbf{x}_0 \sim p_{\text{data}}(\mathbf{x})$, and $\sigma_t = 1 - e^{-\int_0^t \beta(s)ds}$, and $\alpha_t = \sqrt{1 - \sigma_t^2}$. The score function is approximated by $\boldsymbol{\epsilon}_{\boldsymbol{\theta}}(\mathbf{z}_t, t) \approx -\sigma_t\nabla_{\mathbf{z}_t}\log p(\mathbf{z}_t)$, and the score-matching loss is minimized. After training, samples are generated using samplers like DDPM (Ho et al., 2020) and DDIM (Song et al., 2020).

## 3.2 INVERSE PROBLEMS WITH DIFFUSION PRIORS

In general, *an inverse problem aims to find an unknown signal* $\mathbf{x}_0$ *given some noisy measurement* $\mathbf{y}$, related via some forward model $f(.)$,

$$\mathbf{y} = f(\mathbf{x}_0) + \mathbf{v}, \quad \mathbf{v} \sim \mathcal{N}(0, \sigma_v^2\mathbf{I}), \tag{1}$$

where the forward model is domain-dependent. In a Bayesian framework, the solution boils down to sample from the posterior $p(\mathbf{x}_0|\mathbf{y}) \propto p(\mathbf{y}|\mathbf{x}_0)p(\mathbf{x}_0)$, where $p(\mathbf{y}|\mathbf{x}_0)$ is the measurement model (1) and $p(\mathbf{x}_0)$ is the prior imposed by the diffusion model.

**Diffusion posterior sampling approaches.** These methods generate a sample from the posterior by running the reverse process (see Section 3.1) using conditional score at $t$ obtained via Bayes' rule as

$$\nabla_{\mathbf{x}_t}\log p(\mathbf{x}_t|\mathbf{y}) = \nabla_{\mathbf{x}_t}\log p(\mathbf{y}|\mathbf{x}_t) + \nabla_{\mathbf{x}_t}\log p(\mathbf{x}_t). \tag{2}$$

While the second term uses a pre-trained diffusion model, the first is intractable, as seen from $p(\mathbf{y}|\mathbf{x}_t) = \int p(\mathbf{y}|\mathbf{x}_0)p(\mathbf{x}_0|\mathbf{x}_t)d\mathbf{x}_0$. Prior works (Chung et al., 2022a; Song et al., 2022; Kadkhodaie and Simoncelli, 2021; Song et al., 2023) address this with a Gaussian approximation of $p(\mathbf{x}_0|\mathbf{x}_t)$ using Tweedie's formula $\mathbb{E}[\mathbf{x}_0|\mathbf{x}_t] = \frac{1}{\alpha_t}(\mathbf{x}_t - \sigma_t\boldsymbol{\epsilon}_{\boldsymbol{\theta}}(\mathbf{x}_t, t))$. Still, this requires the computation of the *score Jacobian*, which is computationally expensive, especially for pixel-based models at high-resolution. This can be partially addressed by using a suitable latent space (Rombach et al., 2021), alleviating the computational demands of pixel-domain solvers for high-resolution images. However, as discussed in the previous section, it introduces additional steps which arise from the adversarial training of the autoencoder (Rout et al., 2024). We defer more details to Appendix B.

**Variational inference approaches.** Recently, RED-diff was introduced in Mardani et al. (2024), which avoids computing the score Jacobian[1]. RED-diff frames the sampling problem as stochastic optimization by minimizing the KL divergence

$$q(\mathbf{z}_0|\mathbf{y}) = \underset{q(\mathbf{z}_0|\mathbf{y})}{\arg\min} \text{KL}(q(\mathbf{z}_0|\mathbf{y})||p(\mathbf{z}_0|\mathbf{y})), \tag{3}$$

where $\mathbf{x}_0 = \mathcal{D}(\mathbf{z}_0)$. When $q(\mathbf{z}_0|\mathbf{y}) \sim \mathcal{N}(\boldsymbol{\mu}_z, \sigma_z^2\mathbf{I})$, the KL minimization (3) boils down to a maximum-a-posteriori optimization that leverages the diffusion model's trajectory as a regularizer, resulting in a simple and tractable method. However, this approach shares the same limitations as score distillation regarding diversity and mode collapse. Additionally, applying this formulation directly with latent diffusion models produces blurry results (see Appendix D.8.3).

## 4 REPULSIVE VARIATIONAL DIFFUSION SAMPLING

In Section 4.1, we address $(c1)$ by introducing a repulsion mechanism, promoting diversity through a multimodal variational approximation using interactive particles. Then, in Section 4.2, we tackle

---

[1] RED-Diff was proposed in the pixel-domain. However, for the sake of clarity, here we express it with respect to the latent diffusion models.

($c2$) by proposing an augmented variational formulation. Finally, in Section 4.3 we combine both techniques to derive **Repulsive Latent Score Distillation** (RLSD), our proposed solver for inverse problems using latent diffusion models.

## 4.1 Tackling Mode Collapse: Enhancing Diversity via Repulsion

We aim to solve inverse problems characterized by the forward model in (1) by minimizing the reverse KL divergence (3). However, as explained in Section 3.2, minimizing (3) with a Gaussian variational distribution leads to a unimodal approximation of a multimodal posterior, which is problematic for highly ill-posed problems like inpainting. To circumvent this, we propose a *particle approximation* for defining a *multimodal* variational distribution. More precisely, we incorporate a repulsion force within the particles to encourage the exploration of multiple modes.

**Particle interpretation of SDS.** To facilitate the presentation, throughout this section we consider the unconditional case of (3), i.e., without measurements:

$$q(\mathbf{z}_0) = \underset{q(\mathbf{z}_0)}{\arg\min} \, \mathrm{KL}(q(\mathbf{z}_0)||p(\mathbf{z}_0)) \tag{4}$$

Following Song et al. (2021a), we rewrite (4) in terms of the diffused trajectory as

$$q(\mathbf{z}_0) = \underset{q(\mathbf{z}_0)}{\arg\min} \, \mathbb{E}_{t\sim\mathcal{U}[0,T]} \left[ \omega(t)\mathrm{KL}(q(\mathbf{z}_t)||p(\mathbf{z}_t)) \right], \tag{5}$$

where $\omega(t)$ is a weighting function and $q(\mathbf{z}_t) = \int q(\mathbf{z}_t|\mathbf{z}_0)q(\mathbf{z}_0)\mathrm{d}\mathbf{z}_0$ depends on the diffused trajectory $q(\mathbf{z}_t|\mathbf{z}_0) \sim \mathcal{N}(\alpha_t\mathbf{z}_0, \sigma_t^2\mathbf{I})$ and the *variational approximation* $q(\mathbf{z}_0)$. The optimization in (5) corresponds to score distillation, which can be formulated as a Wasserstein gradient flow (details of WGF can be found in Appendix E.1). At the particle level, the WGF is described by the following ODE

$$\mathrm{d}\mathbf{z}_{0,\tau}^{(i)} = \mathbb{E}_t \left[ \omega(t) \left( \nabla_{\mathbf{z}_{t,\tau}^{(i)}} \log p\left(\mathbf{z}_{t,\tau}^{(i)}\right) - \nabla_{\mathbf{z}_{t,\tau}^{(i)}} \log q_\tau\left(\mathbf{z}_{t,\tau}^{(i)}\right) \right) \right] \mathrm{d}\tau, \tag{6}$$

where $p\left(\mathbf{z}_{t,\tau}^{(i)}\right)$ is the target distribution, $q_\tau\left(\mathbf{z}_{t,\tau}^{(i)}\right)$ is the marginal distribution of a generic particle $i$ at time-step $\tau$, and $t$ is the noise level of the diffusion model. In a nutshell, the WGF of $\mathbf{z}_0$ in (6) is computed as an expectation over its diffused trajectory (noise levels $t$), involving the gradients of $\mathbf{z}_t$. This formulation shields light on how the particles are propagated when optimizing (5). More precisely, it becomes evident that for an *initial Gaussian variational approximation at $\tau = 0$, the marginal for all $\tau$ is also Gaussian.* The dynamic in (6) yields a deterministic trajectory where *the mode of the Gaussian variational approximation will match one of the modes of $p(\mathbf{z}_0)$.* Consequently, assuming the same initial position, *all particles will converge to the same mode.* Although this can be mitigated ad hoc by changing the particles' initial positions, we seek a more principled method.

In this context, we can enhance diversity by considering 1) a multimodal (but most likely intractable) variational distribution or 2) interactive particle systems; in this work, we focus on the second one due to its tractability. When considering an interactive set of particles, the key design factor is the coupling term, which prevents the ensemble from collapsing to the same mode. In particular, we propose using a repulsion term.

**Repulsive variational distribution.** Inspired by D'Angelo and Fortuin (2021b), we consider an ensemble of interacting particles coupled via a *repulsive force* that pushes particles away from collapsing to the same solution. In a nutshell, we introduce a repulsive force that yields the following modification of the gradient flow in (6)

$$\mathrm{d}\mathbf{z}_{0,\tau}^{(i)} = \mathbb{E}_t \left[ \omega(t) \left( \nabla_{\mathbf{z}_{t,\tau}^{(i)}} \log p\left(\mathbf{z}_{t,\tau}^{(i)}\right) - \nabla_{\mathbf{z}_{t,\tau}^{(i)}} \log q_\tau\left(\mathbf{z}_{t,\tau}^{(i)}\right) - \nabla_{\mathbf{z}_{t,\tau}^{(i)}} \mathcal{R}(\mathbf{z}_{t,\tau}^{(1)}, \cdots, \mathbf{z}_{t,\tau}^{(N)}) \right) \right] \mathrm{d}\tau, \tag{7}$$

where $N$ is the number of particles and $\mathcal{R}(\mathbf{z}_{t,\tau}^{(1)}, \cdots, \mathbf{z}_{t,\tau}^{(N)})$ is the coupling between particles such that its gradient is the repulsive force[2]. Notice that the marginal distribution in (7) at each time-step $\tau$ is given by (where $Z$ is a normalizing constant)

$$q_\tau\left(\mathbf{z}_{t,\tau}^{(1)}, \cdots, \mathbf{z}_{t,\tau}^{(N)}\right) = \frac{1}{Z}\mathcal{R}\left(\mathbf{z}_{t,\tau}^{(1)}, \cdots, \mathbf{z}_{t,\tau}^{(N)}\right) \prod_{i=1}^{N} q_\tau\left(\mathbf{z}_{t,\tau}^{(i)}\right). \tag{8}$$

---

[2]We consider here a repulsive force because we seek diversity. However, an attractive force can be considered within this same framework.

Throughout this work, we consider a pairwise kernel function $k$ such that the repulsive force adopts the form $\nabla_{\mathbf{z}_{t,\tau}^{(i)}} \mathcal{R}\left(\mathbf{z}_{t,\tau}^{(1)}, \cdots, \mathbf{z}_{t,\tau}^{(N)}\right) = \nabla_{\mathbf{z}_{t,\tau}^{(i)}} \sum_{j=1}^{N} \log\left[k\left(\mathbf{z}_{t,\tau}^{(i)}, \mathbf{z}_{t,\tau}^{(j)}\right)\right]^{\gamma}$; see the numerical experiments for the particular instances of the kernel $k$. The repulsive force allows us to consider simple and flexible variational distributions that can discover multiple modes. For a simple illustration in the Gaussian case, see Appendix D.10.1. Finally, notice that when $\gamma = 0$, we recover the i.i.d. (non-repulsive) case.

## 4.2 Tackling Latent Inversion: Augmentation of the Variational Distribution

As discussed in Section 3.2, solving (3) directly in the latent spaces yields blurry solutions. To tackle this, we propose to solve an augmented version of this problem, allowing us to decouple the data and the latent space of the diffusion model. Formally, we introduce an auxiliary variable $\mathbf{x}_0$ defined in the data (pixel) space, which yields an augmented variational distribution $q(\mathbf{z}_0, \mathbf{x}_0|\mathbf{y})$ and an augmented posterior as

$$p(\mathbf{z}_0, \mathbf{x}_0|\mathbf{y}) \propto \exp\left(-\frac{1}{2\sigma_v^2}||\mathbf{y} - f(\mathbf{x}_0)||^2 - \lambda g(\mathbf{z}_0) - \frac{1}{2\rho^2}||\mathbf{x}_0 - \mathcal{D}(\mathbf{z}_0)||^2\right), \qquad (9)$$

where $\rho$ controls the correlation between the variables $\mathbf{x}_0$ and $\mathbf{z}_0$, and $\exp\left(-\lambda g(\mathbf{z}_0)\right)$ represents the prior distribution parameterized by the latent diffusion model. Notice that the definition in (9) implies that $\mathbf{x}_0|\mathbf{z}_0 \sim \mathcal{N}(\mathcal{D}(\mathbf{z}_0), \rho^2\mathbf{I})$, i.e., the conditional distribution of the data point ($\mathbf{x}_0$) is centered at the value of the decoder applied to the latent point ($\mathbf{z}_0$). However, this is not a delta but has some variance given by $\rho^2$. It can be shown that $p(\mathbf{x}_0|\mathbf{y}, \lambda, \rho^2)$ converges in total variational to the true posterior $p(\mathbf{x}_0|\mathbf{y}, \lambda)$ when $\rho \to 0$ (Van Dyk and Meng, 2001; Vono et al., 2020) (details can be found in Appendix A.1). We can reformulate the optimization problem in (3) as

$$q(\mathbf{z}_0, \mathbf{x}_0|\mathbf{y}) = \underset{q(\mathbf{z}_0, \mathbf{x}_0|\mathbf{y})}{\operatorname{argmin}} \operatorname{KL}(q(\mathbf{z}_0, \mathbf{x}_0|\mathbf{y})||p(\mathbf{z}_0, \mathbf{x}_0|\mathbf{y})). \qquad (10)$$

When considering a diffusion model as data prior, our problem boils down to minimizing the variational lower bound, formalized in Proposition 1.

**Proposition 1** *Assuming we have access to a diffusion model $\nabla_{\mathbf{z}_t} \log p\left(\mathbf{z}_t\right)$ for the prior on $\mathbf{z}_0$, then the KL minimization w.r.t $q$ in* (10) *is equivalent to minimizing the variational bound, which can be done by solving the following optimization problem*

$$\min_{q(\mathbf{x}_0, \mathbf{z}_0|\mathbf{y})} \mathbb{E}_{q(\mathbf{z}_0|\mathbf{y})}[H(q(\mathbf{x}_0|\mathbf{z}_0, \mathbf{y}))] + \mathbb{E}_{q(\mathbf{x}_0, \mathbf{z}_0|\mathbf{y})}\left[\frac{1}{2\sigma_v^2}||\mathbf{y} - f(\mathbf{x}_0)||^2\right] + \qquad (11)$$

$$\mathbb{E}_{q(\mathbf{x}_0, \mathbf{z}_0|\mathbf{y})}\left[\frac{1}{2\rho^2}||\mathbf{x}_0 - \mathcal{D}(\mathbf{z}_0)||^2\right] + \int_0^T \tilde{\omega}(t)\mathbb{E}_{q(\mathbf{z}_t|\mathbf{y})}\left[||\nabla_{\mathbf{z}_t}\log q\left(\mathbf{z}_t \mid \mathbf{y}\right) - \nabla_{\mathbf{z}_t}\log p\left(\mathbf{z}_t\right)||_2^2\right] dt.$$

The proof is in Appendix A.2. When $\rho \to 0$, then $\mathbf{x}_0 = \mathcal{D}(\mathbf{z}_0)$ and the augmented KL optimization boils down to the objective (3). To solve the problem in Proposition 1, we need to specify the variational distribution $q(\mathbf{x}_0, \mathbf{z}_0|\mathbf{y})$; we now incorporate our result from Section 4.1.

## 4.3 Repulsive Latent Score Distillation for Solving Inverse Problems

We now derive RLSD, which integrates the techniques from Sections 4.1 and 4.2. Specifically, we apply the repulsive variational distribution introduced in Section 4.1 to instantiate the augmented variational formulation detailed in Proposition 1. By employing the particle approximation defined in (8), we define a multimodal distribution that enables a better exploration of the posterior's search space, facilitating the discovery of multiple modes and addressing ($c1$). Notably, this variational approximation yields a tractable gradient, formalized in Proposition 2.

**Proposition 2** *When considering the variational distribution defined in* (8)*, the KL minimization w.r.t $q(\mathbf{x}_0, \mathbf{z}_0|\mathbf{y})$ defined in Proposition 1 can be approximated with an ensemble of $N$ particles and admits the following gradient*

$$\frac{1}{N}\sum_{i=1}^{N}\nabla_{\mathbf{u}^{(i)}}\left[\frac{1}{2\sigma_v^2}||\mathbf{y} - f(\mathbf{x}_0^{(i)})||^2 + \frac{1}{2\rho^2}||\mathbf{x}_0^{(i)} - \mathcal{D}(\mathbf{z}_0^{(i)})||^2\right] + \nabla_{\mathbf{z}_{0,\tau}^{(i)}}\operatorname{reg}(\mathbf{z}_{t,\tau}^{(1)}, \cdots, \mathbf{z}_{t,\tau}^{(N)}) \qquad (12)$$

*for $i = 1, \cdots, N$ and where $\mathbf{u}^{(i)} = [\mathbf{x}_0^{(i)}, \mathbf{z}_0^{(i)}]$. The regularization term is given by*

$$\nabla_{\mathbf{z}_{0,\tau}^{(i)}} \mathrm{reg}(\mathbf{z}_{t,\tau}^{(1)}, \cdots, \mathbf{z}_{t,\tau}^{(N)}) = \mathbb{E}_{\boldsymbol{\epsilon},t} \left[ \lambda_t \left( \boldsymbol{\epsilon}_{\boldsymbol{\theta}}(\mathbf{z}_{t,\tau}^{(i)}, t) - \boldsymbol{\epsilon} - \nabla_{\mathbf{z}_{t,\tau}^{(i)}} \gamma \sigma_t \log \sum_{j=1}^{N} k \left( \mathbf{z}_{t,\tau}^{(i)}, \mathbf{z}_{t,\tau}^{(j)} \right) \right) \right].$$

(13)

*where $\lambda_t := \frac{T\alpha_t}{\sigma_t} \frac{d\omega(t)}{dt}$ and $\gamma \geq 0$.*

The proof can be found in Appendix A.3. The gradient defined in Proposition 2 comprises three terms: a *measurement matching term*, an *error term* measuring the discrepancy between the variable in the pixel space and the decoded latent, and a *regularization term* that combines a *score-matching regularizer* with a *diversity-promoting component*. This repulsion term acts as a second regularizer, enhancing diversity during the sampling process. Our approach is not limited to any specific latent diffusion model.

**Practical algorithm for sampling using RLSD.** The underlying optimization of the gradient update in Proposition 2, a particular case of Propositon 1, is highly non-convex (diffusion denoiser and the repulsion term) and challenging to solve. To alleviate this, we adopt a half quadratic splitting technique (Geman and Yang, 1995). The algorithm is shown in Algorithm 1; we define sg[.] as stopped-gradient operator to emphasize that the term inside it is not differentiated during the optimization step. We denote $\tilde{\rho} = \frac{\sigma_y^2}{\rho^2}$. For the weighting function $\lambda_t$ ($\omega(t)$ is embedded) and timesteps, we follow the strategy introduced in Mardani et al. (2024), where $\lambda_t = \lambda(\sigma_t/\alpha_t)$, and the timesteps follow a decreasing order (from $t_{\max}$ to $t_{\min}$); we fix $t_{\max} = T$ and $t_{\min} = 0$. Regarding *computational burden*, our final algorithm only performs one backpropagation through the decoder in the $\mathbf{z}$-step and $N$ backpropagations to compute the repulsive kernel (lines 7 and 9 in Alg. 1 are with respect to all particles). The complexity of the repulsive force depends on the number of particles as well as the domain of the kernel. Notice that the amount of particles is a hyperparameter, allowing us to control the trade-off between diversity and speed. Importantly, in contrast to previous works, we do not backpropagate through the score network.

---

**Algorithm 1** RLSD for solving inverse problems

---

**Require:** $\mathbf{y}, f(.), L, \boldsymbol{\epsilon}_{\boldsymbol{\theta}}(\mathbf{z}_t, t), \mathcal{D}(.), \{\lambda, \gamma, \tilde{\rho}, l_{r_x}, l_{r_z}\}$

1: Initialize $\{\mathbf{x}_{i,0}^0\}_{i=1}^N, \{\mathbf{z}_{i,0}^0\}_{i=1}^N$
2: **for** $\ell = 1$ to $L$ **do**
3:      $t = T - \frac{\ell}{L}T$ and $\boldsymbol{\epsilon} \sim \mathcal{N}(0, \mathbf{I})$
4:      $\lambda_t = \lambda(\sigma_t/\alpha_t)$
5:      $\mathbf{z}_{i,t}^\ell = \alpha_t \mathbf{z}_{i,0}^\ell + \sigma_t \boldsymbol{\epsilon}$
6:      $\mathcal{L}_z = \sum_{i=1}^N \|\mathbf{x}_i^\ell - \mathcal{D}(\mathbf{z}_{i,0}^\ell)\|^2 + \lambda_t \left( \mathrm{sg} \left[ \boldsymbol{\epsilon}_{\boldsymbol{\theta}}(\mathbf{z}_{i,t}^\ell, t) - \boldsymbol{\epsilon} - \gamma \nabla_{\mathbf{z}_t^{(i)}} \sigma_t \log \sum_{j=1}^N k \left( \mathbf{z}_t^{(i)}, \mathbf{z}_t^{(j)} \right) \right] \right)^\top \mathbf{z}_{i,0}^\ell$
7:      $\mathbf{z}_0^\ell = \mathrm{OptimizerStep}_{\mathbf{z}_0^\ell}(\mathcal{L}_z, l_{r_z})$
8:      $\mathcal{L}_x = \sum_{i=1}^N \|\mathbf{y} - f(\mathbf{x}_i^l)\|^2 + \tilde{\rho}\|\mathbf{x}_i^l - \mathcal{D}(\mathbf{z}_{i,0}^l)\|^2$
9:      $\mathbf{x}_0^\ell = \mathrm{OptimizerStep}_{\mathbf{x}_0^\ell}(\mathcal{L}_x, l_{r_x})$
10: **end for**
11: **return** $\{\mathbf{x}_{i,0}^L\}_{i=1}^N$

---

## 5   EXPERIMENTS

In this section, we compare RLSD against state-of-the-art (SoTA) methods for solving inverse problems using latent diffusion models. We consider 100 samples from the validation set of FFHQ (Karras et al., 2019) used in Chung et al. (2022a). We compute PSNR [dB], LPIPS, and FID as metrics. Throughout the experiments, we seek to show the following:

- Our method generates more diverse solutions, in particular for tasks like inpainting and phase retrieval,
- When diversity is not relevant, our augmented formulation generates high-quality samples and outperforms baseline methods.

**Sampling setup.**    Unless we state otherwise, we consider 1000 steps (the full denoising trajectory) for all the cases. We denote by NonAug-RLSD our repulsive method without augmentation (see Alg. 2 in Appendix D.1), and by NonRepuls-RLSD our method with augmentation but without repulsion. We consider Adam (Kingma, 2014) in the optimization steps (lines 7 and 9 in Alg. 1) and set the momentum pair $(0.9, 0.99)$. We randomly initialize variables $\mathbf{x}$ and $\mathbf{z}$ and generate a batch of $N = 4$ particles per noisy measurement. Regarding the pre-trained model, we consider Stable diffusion v2.1, although other latent diffusion models can be used. As diversity metric, we evaluate the pairwise diversity as the $1 -$ cosine similarity between the $N$ particles. Lastly, for the kernel function, we consider a RBF: $k(\mathbf{z}_i, \mathbf{z}_j) = \exp(-\frac{||g_{\text{DINO}}(\mathbf{z}_i) - g_{\text{DINO}}(\mathbf{z}_j)||^2}{h_t})$, where $h_t = m_t^2/\log N$, $m_t$ is the median particle distance (Liu and Wang, 2016) and $g_{\text{DINO}}$ is a pre-trained neural network (Caron et al., 2021). Details about implementation are in Appendix D.1, and ablation analysis in D.8.

**Baselines methods.**    As we focus on methods that leverage large pre-trained models such as Stable diffusion, we compare with the recent PSLD (Rout et al., 2024) and Latent RED-Diff. For completeness, we also include a comparison with SoTA methods in the pixel-domain, namely DPS (Chung et al., 2022a) and RED-diff (Mardani et al., 2024). Details about the implementation of each method are in Appendix D.1. Given that pixel-based diffusion solvers generate images at $256 \times 256$, we follow the strategy from Rout et al. (2024) and downsample the results generated by our sampler, which have a $512 \times 512$ resolution, to do a fair comparison.

## 5.1 INPAINTING

Inpainting, with its inherent ambiguity, provides a suitable benchmark to showcase two key aspects of RLSD: 1) high-quality reconstruction and 2) enhanced diversity achieved through the repulsion term. Additional linear inverse problems such as super resolution and deblurring are detailed in Appendix D. Specifically, we consider a box hiding half of the faces (see Fig. 2 and Appendix D.7). For the hyperparameters, we set $\lambda = 0.14$, $\tilde{\rho} = 0.075$, $l_{r_x} = 0.4$ and $l_{r_z} = 0.8$. Results in Table 2 show that *RLSD outperforms their baselines* in image quality (PSNR and FID); in particular, it outperforms PSLD, the other sampler at a resolution of $512 \times 512$. Moreover, it demonstrates that our method can trade-off diversity for quality by modifying the weight $\gamma$: while RLSD ($\gamma = 50$) achieves higher diversity than NonRepuls-RLSD, the later achieves better performance. In addition, in Table 14 we demonstrate that RLSD achieves the best performance when selecting the sample with best performance.

This highlights RLSD's ability to combine superior reconstruction with higher diversity, suggesting a *mixed strategy where some particles interact and others do not*. Indeed, this strategy (Hybrid-RLSD), where three particles interact and one particle is propagated independently of the ensemble, combines the best of both worlds, achieving the best balance between quality and diversity.

**Diversity-quality trade off.**    Fig. 2 showcases the diversity-quality trade-off. While PSLD generates four diverse samples of lower quality, Non-Repuls RLSD tends to fill the four images with very similar solutions. On the other hand, when considering RLSD ($\gamma = 50$), the results are more diverse while maintaining high quality: in images 1 and 3, the woman has the left eye hidden, while none of the generated images with NonRepuls-RLSD show this. We defer to Appendix D.10 a more exhaustive analysis of diversity-quality trade off.

## 5.2 NON-LINEAR INVERSE PROBLEMS

We consider in this case nonlinear inverse problems. Given that PSLD only works on linear inverse problems, we compare against latent DPS and latent RED-Diff.

**Phase retrieval.** We first consider phase retrieval, which deals with reconstructing the phase from only magnitude observations in the Fourier domain. Phase retrieval is known as a highly ill-posed problem, given that it is invariant to $180°$ rotation, which yields two equally probable modes. Thus, its posterior has multiple modes, which are discrete and isolated. We follow the strategy from DPS (Chung et al., 2022a), where an oversampling of rate 2 is used. We consider 6 particles for the particle variational approximation, and 6 independent particles for the non-repulsive case. Furthermore, we set $\gamma = 30$, and we only consider repulsion between $t \in [0.4T, T]$. Results are shown in Table 3. First, we observe that for this experiment, the augmented variational approximation (NonRepuls-RLSD and RLSD) entails a more unstable algorithm, and thus, not converging to good modes.

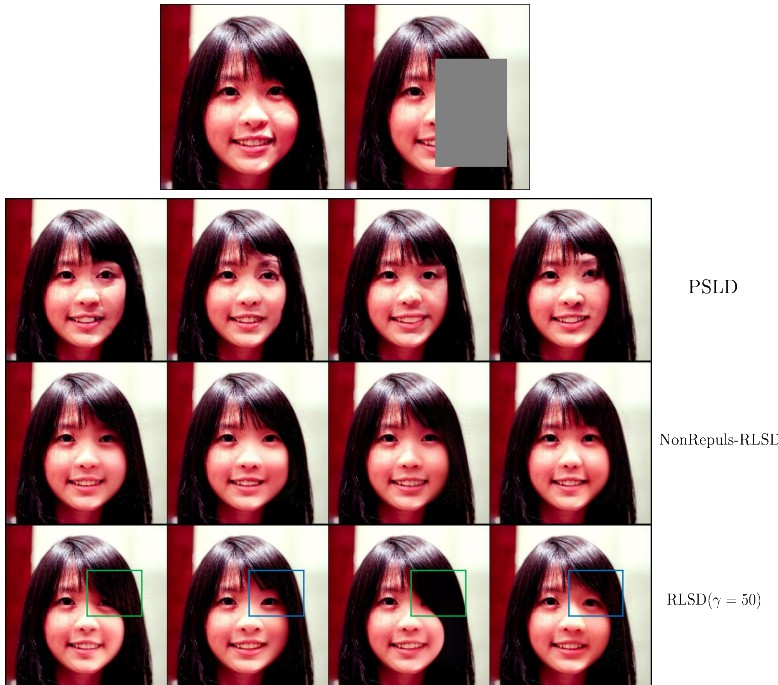

Figure 2: Ipainting half a face (top to bottom): Ground truth, measurement, PSLD, NonRepuls-RLSD, and RLSD ($\gamma = 50$). Each method generates four samples from different initializations, with RLSD incorporating repulsion. Both RLSD and NonRepuls-RLSD outperform PSLD. RLSD produces diverse results (e.g., images 1 and 3 hide the left eye), whereas NonRepuls-RLSD samples appear similar.

Table 2: Box inpainting (half face) with $\sigma_v = 0.001$ - FFHQ 512. For evaluation, we downsample the estimated images of RLSD and PSLD to 256. The best method for each metric is bolded.

| Sampler | PSNR [dB] ↑ | LPIPS ↓ | FID ↓ | Diversity |
|---|---|---|---|---|
| PSLD | 22.72 | 0.082 | 57.7 | 0.03 |
| Latent RED-diff | 23.5 | 0.15 | 92.59 | 0.009 |
| ReSample | 19.44 | 0.2 | 146.68 | - |
| RED-Diff (Pixel) | 23.1 | **0.067** | 29.79 | 0.004 |
| DPS (Pixel) | 23.4 | 0.14 | 78.88 | **0.04** |
| NonAug-RLSD ($\gamma = 50$) | 23.34 | 0.164 | 98.65 | 0.035 |
| NonRepuls-RLSD | **24.98** | 0.079 | **29.18** | 0.004 |
| RLSD ($\gamma = 50$) | 24.69 | 0.111 | 31.41 | 0.015 |
| Hybrid-RLSD | 24.72 | 0.096 | 30.48 | 0.018 |

Second, the results show that our proposed method NonAug-RLSD is more stable, and that the particle approximation effectively captures more modes. In particular, in Fig. 3 we show an example where the latent RED-diff generates 5 images that look similar, while NonAug-RLSD generates 6 samples that corresponds to different modes. This showcases that our methods indeed promote diversity, even for a nonlinear inverse problems and at a resolution of $512 \times 512$.

**High dynamic range (HDR).** We try HDR, which performs the clipping function $f(\mathbf{x}) = \text{clip}(2\mathbf{x}, -1, 1)$ on the normalized RGB pixels. HDR is known to be simpler than phase retrieval, and where diversity is not fundamental. Again, we consider Latent DPS as PSLD does not work for nonlinear inverse problems. The results are in Table 4, where RLSD outperforms all other baselines (when choosing the best sample among the group). This showcases that repulsion promotes a better exploration, and reaching better modes.

Table 3: Phase-retrieval with $\sigma_v = 0.001$ on FFHQ 512. The best method for each metric is bolded.

| Sampler | PSNR [dB] ↑ | LPIPS ↓ | FID ↓ |
|---|---|---|---|
| Latent-DPS | 14.98 | 0.618 | 291.68 |
| Latent-RED-diff ($\gamma = 0$) | 19.65 | 0.458 | 173.18 |
| NonAug-RLSD ($\gamma = 30$) | **24.21** | **0.359** | **130.09** |
| NonRepuls-RLSD | 18.33 | 0.495 | 223.14 |
| RLSD ($\gamma = 30$) | 20.43 | 0.449 | 207 |

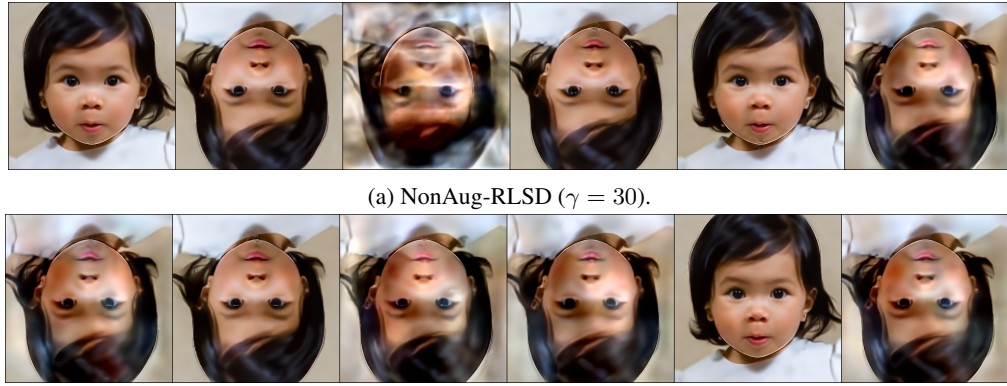

(a) NonAug-RLSD ($\gamma = 30$).

(b) Latent RED-Diff.

Figure 3: Results for Phase Retrieval. Adding repulsion between particles allows to sample from different modes (top row).

Table 4: HDR with $\sigma_v = 0.001$ on FFHQ 512. The best method for each metric is bolded.

| Sampler | PSNR [dB] ↑ | LPIPS ↓ | FID ↓ |
|---|---|---|---|
| Latent-DPS | 15.77 | 0.449 | 181.19 |
| Latent-RED-diff | 25.68 | 0.200 | 93.71 |
| NonAug-RLSD ($\gamma = 30$) | 24.84 | 0.210 | 94.15 |
| NonRepuls-RLSD | 27.72 | 0.087 | 37.57 |
| RLSD ($\gamma = 50$) | **28.13** | **0.089** | **37.32** |

## 6 CONCLUSIONS AND LIMITATIONS

In this paper, we introduce **Repulsive Latent Score Distillation (RLSD)**, a *plug-and-play* variational sampler that leverages pre-trained latent diffusion models to solve inverse problems, balancing quality, diversity, and computational efficiency. Inspired by the Wasserstein gradient flow of score distillation, **RLSD** mitigates mode collapse and latent diffusion inversion.

To tackle ($c1$) mode collapse, we introduce a particle-based variational distribution with a *repulsion mechanism* based on kernel similarity. To handle ($c2$) latent inversion (from adversarial training), we propose *distribution augmentation* to decouple latent and pixel spaces. The algorithm applies two regularizations: **denoising** to enforce the prior and **repulsion** to promote diversity.

Numerical experiments demonstrate that RLSD merges the benefits of variational samplers (memory and compute efficiency) with posterior samplers (diversity), allowing control over speed and diversity through simple regularization weights. The repulsion force significantly boosts diversity in ill-posed problems like inpainting and phase retrieval.

Our method has some limitations. Including the repulsion term increases computational demands, complicating real-time use. The chosen repulsion kernel may not be optimal under high noise levels, suggesting a need for *adaptive kernel learning*. Moreover, the method introduces additional hyperparameters, requiring better coupling for noise levels and deriving repulsion weights based on the forward operator.

## ACKNOWLEDGEMENT

Research was sponsored by National Science Foundation (CCF 2340481) and the Army Research Office (Grant Number W911NF-17-S-0002). NZ was partially supported by a Ken Kennedy Institute 2024–25 Ken Kennedy-HPE Cray Graduate Fellowship. The views and conclusions contained in this document are those of the authors and should not be interpreted as representing the official policies, either expressed or implied, of the Army Research Office or the U.S. Army or the U.S. Government. The U.S. Government is authorized to reproduce and distribute reprints for Government purposes notwithstanding any copyright notation herein.

## REPRODUCIBILITY STATEMENT

In this work, we have taken several steps to ensure the reproducibility of our results. We provide a comprehensive description of the methodology in Section 4 of the main text and the full algorithm in pseudocode are in Algorithms 1 and 2. In addition to this, in Section 5 as well in Appendix D.1 we give details of all the hyperparameters for each experiment. Additionally, we have included all necessary proofs for theoretical claims in Appendix A. For the experiments, we gave details of all the datasets that we used and how we compute the metrics. We include a link to our repository in the abstract. In the README.md file in the repository it is explained the steps to run the code. Lastly, in Appendix D.10.3 we included four .gif files, which might require a pdf reader that can reproduce gifs.

## ETHICS STATEMENT

Our method has the potential to cause unintended negative consequences if not handled responsibly. Key ethical and societal risks include the amplification of biases, difficulties in verifying the authenticity of generated content, which could contribute to misinformation, and the economic impact on creative professionals. Additionally, there are concerns over the misuse of this technology for harmful purposes, privacy issues related to the datasets used, cultural insensitivity, and potential intellectual property conflicts surrounding AI-generated creations. Addressing these risks necessitates the development of strong ethical standards, regulatory frameworks, and safeguards to ensure fairness, privacy protection, and respect for cultural and intellectual property rights. Therefore, it is essential that RLSD and other generative models are applied with a clear understanding of their limitations, and that outcomes are validated carefully to reduce these risks.

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

# A  TECHNICAL PROOFS

## A.1  DATA AUGMENTATION

In Section 4.2 we consider an augmented variational distribution $q_\rho(\mathbf{x}_0, \mathbf{z}_0|\mathbf{y})$ such that

$$q_\rho(\mathbf{x}_0|\mathbf{y}) = \int q_\rho(\mathbf{x}_0, \mathbf{z}_0|\mathbf{y})\mathrm{d}\mathbf{z}_0. \tag{14}$$

Therefore, we seek a joint distribution such that Property 1 holds.

**Property 1** *For all $\mathbf{x}_0 \in \mathbb{R}^N$, it holds $\lim_{\rho\to 0} \pi_\rho(\mathbf{x}_0) = \pi(\mathbf{x}_0)$.*

Notice that our approach using data augmentation resembles some recent methods introduced in the bibliography. In Zhu et al. (2023), the authors propose to decouple data and diffusion model. More recently, a Bayesian version of the RED method was proposed in Faye et al. (2024). This method shares similarities with our method, in the sense that both are augmented versions that resemble RED. However, our method has three main differences: 1) it leverages latent diffusion models, which allows us to solve large-scale inverse problems ($512 \times 512$ and beyond), 2) it uses the diffused trajectory to regularize the solution, and 3) it promotes diversity via the coupling term.

## A.2  PROOF OF PROPOSITION 1

We expand the KL objective as follows

$$\mathrm{KL}(q(\mathbf{z}_0, \mathbf{x}_0|\mathbf{y})\|p(\mathbf{z}_0, \mathbf{x}_0|\mathbf{y})) = \int q(\mathbf{z}_0, \mathbf{x}_0|\mathbf{y}) \log \frac{q(\mathbf{z}_0, \mathbf{x}_0|\mathbf{y})}{p(\mathbf{z}_0, \mathbf{x}_0|\mathbf{y})}\mathrm{d}\mathbf{z}_0\mathrm{d}\mathbf{x}_0 \tag{15}$$

$$= \int q(\mathbf{z}_0, \mathbf{x}_0|\mathbf{y}) \log \frac{q(\mathbf{x}_0|\mathbf{z}_0, \mathbf{y})q(\mathbf{z}_0|\mathbf{y})p(\mathbf{y})}{p(\mathbf{y} \mid \mathbf{x}_0)p(\mathbf{x}_0|\mathbf{z}_0)p(\mathbf{z}_0)}\mathrm{d}\mathbf{z}_0\mathrm{d}\mathbf{x}_0$$

$$= \underbrace{\int q(\mathbf{z}_0, \mathbf{x}_0|\mathbf{y}) \log q(\mathbf{x}_0|\mathbf{z}_0, \mathbf{y})\mathrm{d}\mathbf{z}_0\mathrm{d}\mathbf{x}_0}_{(i)}$$

$$- \underbrace{\int q(\mathbf{z}_0, \mathbf{x}_0|\mathbf{y}) \log p(\mathbf{y} \mid \mathbf{x}_0)\mathrm{d}\mathbf{z}_0\mathrm{d}\mathbf{x}_0}_{(ii)}$$

$$- \int q(\mathbf{z}_0, \mathbf{x}_0|\mathbf{y}) \underbrace{\log p(\mathbf{x}_0|\mathbf{z}_0)\mathrm{d}\mathbf{z}_0\mathrm{d}\mathbf{x}_0}_{(iii)}$$

$$+ \underbrace{\int q(\mathbf{z}_0, \mathbf{x}_0|\mathbf{y}) \frac{q(\mathbf{z}_0|\mathbf{y})}{p(\mathbf{z}_0)}\mathrm{d}\mathbf{z}_0\mathrm{d}\mathbf{x}_0}_{(iv)} + \log p(\mathbf{y}).$$

Based on the augmented posterior, we have for $(ii)$ and $(iii)$ that

$$(ii) = \int q(\mathbf{z}_0, \mathbf{x}_0|\mathbf{y}) \log p(\mathbf{y} \mid \mathbf{x}_0)\mathrm{d}\mathbf{z}_0\mathrm{d}\mathbf{x}_0 = \mathbb{E}_{q(\mathbf{x}_0, \mathbf{z}_0|\mathbf{y})}\left[\frac{1}{2\sigma_v^2}\|\mathbf{y} - f(\mathbf{x}_0)\|^2\right] \tag{16}$$

and

$$(iii) = \int q(\mathbf{z}_0, \mathbf{x}_0|\mathbf{y}) \log p(\mathbf{x}_0 \mid \mathbf{z}_0)\mathrm{d}\mathbf{z}_0\mathrm{d}\mathbf{x}_0 = \mathbb{E}_{q(\mathbf{x}_0, \mathbf{z}_0|\mathbf{y})}\left[\frac{1}{2\rho^2}\|\mathbf{x}_0 - \mathcal{D}(\mathbf{z}_0)\|^2\right]. \tag{17}$$

Regarding the first term, we can write as

$$(i) = \int q(\mathbf{z}_0|\mathbf{y})\left[q(\mathbf{x}_0 \mid \mathbf{z}_0, \mathbf{y}) \log q(\mathbf{x}_0|\mathbf{z}_0, \mathbf{y})\right]\mathrm{d}\mathbf{z}_0\mathrm{d}\mathbf{x}_0 = \mathbb{E}_{q(\mathbf{z}_0|\mathbf{y})}\left[H(q(\mathbf{x}_0 \mid \mathbf{z}_0, \mathbf{y})\right]. \tag{18}$$

Finally, the last term can be obtained by following theorem 2 in Song et al. (2021a), assuming that the score is learned exactly, namely $\epsilon_\theta(\mathbf{z}_t; t) = -\sigma_t\nabla_{\mathbf{z}_t} \log p(\mathbf{z}_t)$, and under some mild assumptions on the growth of $\log q(\mathbf{z}_t \mid \mathbf{y})$ and $p(\mathbf{z}_t)$ at infinity, we have

$$\mathrm{KL}(q(\mathbf{z}_0|\mathbf{y})\|p(\mathbf{z}_0)) = \int_0^T \frac{\beta(t)}{2}\omega(t)\mathbb{E}_{q(\mathbf{z}_t|\mathbf{y})}\left[\|\nabla_{\mathbf{z}_t} \log q(\mathbf{z}_t \mid \mathbf{y}) - \nabla_{\mathbf{z}_t} \log p(\mathbf{z}_t)\|_2^2\right]dt \tag{19}$$

over the denoising diffusion trajectory $\{\mathbf{z}_t\}$ for positive values $\{\beta(t)\}$. This essentially implies that a weighted score-matching over the continuous denoising diffusion trajectory is equal to the KL divergence.

### A.3    PROOF OF PROPOSITION 2

The first two terms are straightforward. We focus here on the regularization term. The regularization term in Proposition 1 corresponds to the score matching loss defined in Song et al. (2021a) For general weighting schemes $\omega(t)$, we have the following Lemma from Song et al. (2021a)

**Lemma 1** *The time-derivative of the* KL *divergence at timestep $t$ obeys*

$$\frac{d\,\mathrm{KL}\,(q\,(\mathbf{z}_t\mid\mathbf{y})\,\|p\,(\mathbf{z}_t))}{dt} = -\frac{\beta(t)}{2}\mathbb{E}_{q(\mathbf{z}_t\mid\mathbf{y})}\left[\|\nabla_{\mathbf{z}_t}\log q\,(\mathbf{z}_t\mid\mathbf{y}) - \nabla_{\mathbf{z}_t}\log p\,(\mathbf{z}_t)\|^2\right].$$

Then, under the condition $\omega(0) = 0$, the integral in Proposition 1 can be written as (see Song et al. (2021a))

$$\int_0^T \frac{\beta(t)}{2}\omega(t)\mathbb{E}_{q(\mathbf{z}_t\mid\mathbf{y})}\left[\|\nabla_{\mathbf{z}_t}\,\log q\,(\mathbf{z}_t\mid\mathbf{y}) - \nabla_{\mathbf{z}_t}\log p\,(\mathbf{z}_t)\,\|^2\right]dt$$

$$= -\int_0^T \omega(t)\frac{d\mathrm{KL}\,(q\,(\mathbf{z}_t\mid\mathbf{y})\,\|p\,(\mathbf{z}_t))}{dt}dt$$

$$\overset{(a)}{=} \underbrace{-\omega(t)\mathrm{KL}\,(q\,(\mathbf{z}_t\mid y)\,\|p\,(\mathbf{z}_t))]_0^T}_{=0} + \int_0^T \omega'(t)\mathrm{KL}\,(q\,(\mathbf{z}_t\mid\mathbf{y})\,\|p\,(\mathbf{z}_t))\,dt$$

$$= \int_0^T \omega'(t)\mathbb{E}_{q(\mathbf{z}_t\mid\mathbf{y})}\left[\log\frac{q\,(\mathbf{z}_t\mid\mathbf{y})}{p\,(\mathbf{z}_t)}\right]dt,$$

where $\omega'(t) := \frac{d\omega(t)}{dt}$. The equality holds because $\omega(t)\mathrm{KL}\,(q\,(\mathbf{z}_t\mid\mathbf{y})\,\|p\,(\mathbf{z}_t))]_0^T$ is zero at $t = 0$ and $t = T$. This is because $\omega(t) = 0$ by assumption at $t = 0$, and $x_T$ becomes a pure Gaussian noise at the end of the diffusion process which makes $p\,(\mathbf{z}_T) = q\,(\mathbf{z}_T\mid\mathbf{y})$ and thus $\mathrm{KL}\,(q\,(\mathbf{z}_T\mid\mathbf{y})\,\|p\,(\mathbf{z}_T)) = 0$.

Now, we consider our proposed variational distribution defined in (8). with $N$ particles and the pairwise kernel. For each particle $i$, we apply the forward diffusion $\mathbf{z}_t^{(i)} = \alpha_t\mathbf{z}_0^{(i)} + \sigma_t\boldsymbol{\epsilon}$, which yields the distribution $q\left(\mathbf{z}_t^{(i)}\mid\mathbf{y}\right) = \mathcal{N}\left(\alpha_t\mathbf{z}_0^{(i)}, \sigma_t^2 I\right)$, and thus $\nabla_{\mathbf{z}_t}\log q_t\,(\mathbf{z}_t\mid\mathbf{y}) = -\left(\mathbf{z}_t^{(i)} - \alpha_t\mathbf{z}_0^{(i)}\right)/\sigma_t^2 = -\frac{\epsilon^{(i)}}{\sigma_t}$. By applying the re-parameterization trick, we obtain

$$\nabla_{\mathbf{z}_0^{(i)}}\mathrm{reg}(\mathbf{z}_t^{(1)},\cdots,\mathbf{z}_t^{(N)}) = \tag{20}$$

$$\int_0^T \omega'(t)\mathbb{E}_{\epsilon\sim\mathcal{N}(0,1)}\left[\left(-\nabla_{\mathbf{z}_t^{(i)}}\gamma\log\sum_{j=1}^N k\left(\mathbf{z}_t^{(i)},\mathbf{z}_t^{(j)}\right) + \nabla_{\mathbf{z}_t^{(i)}}\log q_t(\mathbf{z}_t^{(i)}\mid\mathbf{y}) - \nabla_{\mathbf{z}_t}\log p(\mathbf{z}_t^{(i)})\right)^\top\frac{d\mathbf{z}_t^{(i)}}{d\mathbf{z}_0^{(i)}}\right]dt$$

$$\int_0^T \omega'(t)\mathbb{E}_{\epsilon\sim\mathcal{N}(0,1)}\left[\left(-\nabla_{\mathbf{z}_t^{(i)}}\gamma\log\sum_{j=1}^N k\left(\mathbf{z}_t^{(i)},\mathbf{z}_t^{(j)}\right) - \frac{\epsilon}{\sigma_t} + \frac{\epsilon_{\boldsymbol{\theta}}(\mathbf{z}_t^{(i)};t)}{\sigma_t}\right)^\top\alpha_t\mathbf{I}\right]dt$$

$$\int_0^T \omega'(t)\frac{\alpha_t}{\sigma_t}\mathbb{E}_{\epsilon\sim\mathcal{N}(0,1)}\left[\left(-\nabla_{\mathbf{z}_t^{(i)}}\gamma\sigma_t\log\sum_{j=1}^N k\left(\mathbf{z}_t^{(i)},\mathbf{z}_t^{(j)}\right) - \epsilon + \epsilon_{\boldsymbol{\theta}}(\mathbf{z}_t^{(i)};t)\right)\right]dt.$$

We can rearrange terms to arrive at the following compact form

$$\nabla_{\mathbf{z}_0^{(i)}}\mathrm{reg}(\mathbf{z}_t^{(1)},\cdots,\mathbf{z}_t^{(N)}) = \tag{21}$$

$$\mathbb{E}_{t\sim\mathcal{U}[0,T],\epsilon\sim\mathcal{N}(0,1)}\left[\lambda_t\left(-\nabla_{\mathbf{z}_t^{(i)}}\gamma\sigma_t\log\sum_{j=1}^N k\left(\mathbf{z}_t^{(i)},\mathbf{z}_t^{(j)}\right) - \epsilon + \epsilon_{\boldsymbol{\theta}}(\mathbf{z}_t^{(i)};t)\right)\right]$$

for $\lambda_t := T\omega'(t)\alpha_t/\sigma_t$. When considering the measurement matching term and the error between the ambient and the augmented variable, we obtain $\lambda_t := T\omega'(t)\alpha_t/\sigma_t 4\sigma_v^2\rho^2$.

# B  DIFFUSION FOR INVERSE PROBLEMS

Diffusion models are powerful generative models. Therefore, they have been used as deep generative priors to solve inverse problems. Given a pre-trained diffusion model, this involves running the backward process using a guidance (likelihood) term that incorporates the measurement information. Formally, we can sample from the posterior $p(\mathbf{x}_0|\mathbf{y})$ by running (22)

$$d\mathbf{x}_t = -\frac{1}{2}\beta(t)\mathbf{z}_t dt - \beta(t)\left[\nabla_{\mathbf{x}_t}\log p(\mathbf{x}_t) + \nabla_{\mathbf{x}_t}\log p(\mathbf{y}|\mathbf{x}_t)\right] + \sqrt{\beta(t)}d\mathbf{W}_t. \tag{22}$$

Early studies used Langevin dynamics for linear problems (Kadkhodaie and Simoncelli, 2021; Kawar et al., 2021; Laumont et al., 2022; Zilberstein et al., 2024; 2022), while others used DDPM (Kawar et al., 2022; Chung et al., 2022c;b; Ho et al., 2022). However, approximating the guidance term remains a challenge. Previous works addressed this with a Gaussian approximation of $p(\mathbf{x}_0|\mathbf{x}_t)$ around the MMSE estimator via Tweedie's formula, increasing computational burden (Chung et al., 2022a; Song et al., 2022; Kadkhodaie and Simoncelli, 2021; Song et al., 2023). These methods crudely approximate the posterior score, especially for non-small noise levels. One of the most effective methods is DPS (Chung et al., 2022a), which assumes:

$$p(\mathbf{y}|\mathbf{x}_t) \approx p(\mathbf{y}|\hat{\mathbf{x}}_0 := \mathbb{E}[\mathbf{x}_0|\mathbf{x}_t]) = \mathcal{N}\left(\mathbf{y}|f(\mathbb{E}[\mathbf{x}_0|\mathbf{x}_t]), \sigma_t^2\mathbf{I}\right).$$

Essentially, DPS approximates the likelihood with a unimodal Gaussian distribution center around the MMSE estimator $\mathbb{E}[\mathbf{x}_0|\mathbf{x}_t]$. Under this approximation, the term $p(\mathbf{y}|\mathbf{x}_t)$ boils down to the gradient of a multivariate Gaussian. Although it achieves impressive results, the unimodal approximation is far from optimal. Furthermore, its adaptation to use a latent diffusion model is not straightforward, as explained in (Rout et al., 2024). Recent works (Rout et al., 2024; Song et al., 2024; Kim et al., 2024; Chung et al., 2024) extended this by sampling from the latent space of diffusion models but still face limitations due to the intractable model likelihood. In particular, PSLD incorporates an additional term to guide the reconstruction towards a fixed point of the autoencoder process. This yields the following guidance score, where a *gluing* term is added to circumvent the discontinuity issues at the boundary.

$$\nabla_{\mathbf{z}_t}\log p(\mathbf{y}|\mathbf{z}_t) = \nabla_{\mathbf{z}_t}\log p\left(\mathbf{y}|\hat{\mathbf{x}}_0 = \mathcal{D}\left(\mathbb{E}[\mathbf{z}_0|\mathbf{z}_t]\right)\right) + \tag{23}$$
$$\gamma_t\nabla_{\mathbf{z}_t}\left\|\mathbb{E}[\mathbf{z}_0|\mathbf{z}_t] - \mathcal{E}\left(\mathcal{A}^T\mathbf{y} + \left(\mathbf{I} - \mathcal{A}^T\mathcal{A}\right)\mathcal{D}\left(\mathbb{E}[\mathbf{z}_0 \mid \mathbf{z}_t]\right)\right)\right\|^2,$$

where

$$p(\mathbf{y}|\mathbf{z}_t) \approx \mathcal{N}\left(\mathbf{y}|f(\mathcal{D}(\mathbb{E}[\mathbf{z}_0|\mathbf{z}_t])), \sigma_t^2\mathbf{I}\right).$$

Notice that while the gluing term is effective for linear inverse problems, it cannot handle non-linear cases.

# C  DISCUSSION ON VARIATIONAL AUGMENTED DISTRIBUTION

In Section 4.2 we introduced an augmented variational distribution instead of a variational formulation in the latent space directly. This decision stems from the observation that optimizing in the latent space often produces blurry reconstructions We hypothesize that this is due to the nonlinearity of the decoder $\mathcal{D}(.)$ and the adversarial training of the encoder-decoder pair. Specifically, the autoencoder tends to *compress* fine details, resulting in reconstructions that capture high-level semantics but fail to reproduce the fine-grained features.

To address this issue, we correct deviations from the image manifold during optimization by using an augmented variational formulation. This introduces a coupling term between $\mathbf{x}$ and $\mathbf{z}$ to account for these deviations. First, we define the true augmented posterior distribution. We write $\mathbf{x}_0 = \mathcal{D}(\mathbf{z}_0) + \sigma_z\epsilon$, where $\sigma_z$ is the variance of the posterior of the decoder. Intuitively, when optimizing (10), we

are optimizing the following joint target posterior $p(\mathbf{x}, \mathbf{z}|\mathbf{y})$, which has the following two conditionals associated

$$p(\mathbf{x}|\mathbf{z}, \mathbf{y}) \sim \mathcal{N}(\mu(\mathbf{z}, \mathbf{y}), \mathbf{\Sigma}(\mathbf{z})) \tag{24}$$

$$p(\mathbf{z}|\mathbf{x}) \propto p(\mathbf{x}|\mathbf{z})p(\mathbf{z}) \tag{25}$$

where $\mathbf{\Sigma}(\mathbf{z}) = \frac{1}{\sigma_v^2}\mathbf{A}^\top\mathbf{A} + \frac{1}{\sigma_z^2}\mathcal{D}(\mathbf{z})^\top\mathcal{D}(\mathbf{z})$ and $\mu(\mathbf{z}, \mathbf{y}) = \mathbf{\Sigma}(\mathbf{z})^{-1}\left(\frac{1}{\sigma_v^2}\mathbf{A}^\top\mathbf{y} + \frac{1}{\sigma_z^2}\mathcal{D}(\mathbf{z})\right)$, and $p(\mathbf{z})$ is the diffusion prior. Therefore, the variational inference in the augmented formulation aims to approximate the first Gaussian with another Gaussian $q(\mathbf{x}|\mathbf{z}, \mathbf{y}) \sim \mathcal{N}(\boldsymbol{\mu}_x, \boldsymbol{\sigma}_x^2\mathbf{I})$, with $\boldsymbol{\sigma}_x \to 0$, i.e., a MAP estimate, and the second one with the particle approximation, promoting diversity throughout the diffused trajectory.

This analysis also helps explain why NonAug-RLSD outperforms its augmented variant in phase retrieval. Phase retrieval is a nonlinear inverse problem, making alters (24) intractable: we cannot express the mean and covariance es explained above. Consequently, the augmentation renders a more difficult problem, which might explain why it is unstable.

# D ADDITIONAL EXPERIMENTS

## D.1 IMPLEMETATION DETAILS OF BASELINES

To facilitate reproducibility, we share an anonymous link of our source code`https://file.io/iQNq3U5GpsY6`. If the paper is accepted, we will publish in a public repository.

**PSLD/Latent-DPS.** We use the origianl code from Rout et al. (2024). We use the version with Stable Diffusion, and we select hyperparameters as detailed in the paper.

**RED-Diff.** We use the implementation from the original paper (Mardani et al., 2024). We follow the same weighting scheme, and we use $\lambda = 0.25$ and $l_r = 0.1$. As pretrained models, for FFHQ we use the model from Chung et al. (2022a), while for ImageNet we use the one from Dhariwal and Nichol (2021).

**Latent RED-Diff.** This method is the same as RED-Diff but using a latent diffusion model (as explain in Section 3.2.

**DPS.** We use the implementation from the original paper (Chung et al., 2022a). We follow their configuration of hyperparameters. We use the same pretrained models as RED-Diff.

**FPS-SMC(Dou and Song, 2024).** We use the implementation from the original paper. We follow their configuration of hyperparameters. We use the same pretrained models as RED-Diff.

**ΠGDM.** We use the implementation from the original paper (Song et al., 2022). We follow their configuration of hyperparameters. We use the same pretrained models as RED-Diff.

**ReSample** We use the implementation from the original paper (Song et al., 2022). We follow their configuration of hyperparameters. We use LDM-VQ-4 trained on FFHQ. We tried using Stable Diffusion (the original implementation does not support it), but we got worst results.

**NonAug-RLSD.** This method corresponds to our variant using the particle-based variational approximation (8) and without augmentation. For clarity, we show it in Alg. 2.

---

**Algorithm 2** Non-augmented RLSD for solving inverse problems

---

**Require:** $\mathbf{y}, f(.), L, \boldsymbol{\epsilon}_{\boldsymbol{\theta}}(\mathbf{z}_t, t), \mathcal{D}(.), \{\lambda, \gamma, \tilde{\rho}, l_{r_z}\}$

    **for** $l = 1$ to $L$ **do**

        Initialize $\{\mathbf{z}_{i,0}^0\}_{i=1}^N$

        $t = T - \frac{\ell}{L}T$ and $\boldsymbol{\epsilon} \sim \mathcal{N}(0, \mathbf{I})$

        $\lambda_t = \lambda(\sigma_t/\alpha_t)$

        $\mathbf{z}_{i,t}^\ell = \alpha_t \mathbf{z}_{i,0}^\ell + \sigma_t \boldsymbol{\epsilon}$

        $\mathcal{L}_z = \sum_{i=1}^n \|\mathbf{y} - f(\mathcal{D}(\mathbf{z}_i^l))\|^2 + \lambda_t \left( \mathrm{sg}\left[ \boldsymbol{\epsilon}_{\boldsymbol{\theta}}(\mathbf{z}_{i,t}^\ell, t) - \boldsymbol{\epsilon} - \gamma \nabla_{\mathbf{z}_t^{(i)}} \sigma_t \log \sum_{j=1}^N k\left(\mathbf{z}_t^{(i)}, \mathbf{z}_t^{(j)}\right) \right] \right)^\top \mathbf{z}_{i,0}^\ell$

        $\mathbf{z}_0^\ell = \mathrm{OptimizerStep}_{\mathbf{z}_0^\ell}(\mathcal{L}_z, l_{r_z})$

    **end for**

    **return** $\{\mathbf{x}_{i,0}^L = \mathcal{D}(\mathbf{z}_{i,0}^L)\}_{i=1}^n$

---

**RLSD.** For our augmented formulation, we use Stable diffusion trained in the LAION (Schuhmann et al., 2022) dataset as its pre-trained model. For the kernel function, we consider a RBF $k(\mathbf{z}_i, \mathbf{z}_j) = \exp(-\frac{\|g_{\mathrm{DINO}}(\mathbf{z}_i) - g_{\mathrm{DINO}}(\mathbf{z}_j)\|^2}{h_t})$ where $h_t = m_t^2/\log N$, $m_t$ is the median particle distance (Liu and Wang, 2016) and $g_{\mathrm{DINO}}$ is a pre-trained neural network (Caron et al., 2021). Notice that NonRepuls-RLSD corresponds to $\gamma = 0$.

Regarding the similarity metric, we consider the cosine similarity in the range of DINO defined as follow

$$\mathrm{Sim}(\mathbf{x}_1, \cdots, \mathbf{x}_N) = \frac{1}{n(n-1)} \sum_{i \neq j} \frac{g_{\mathrm{DINO}}(\mathbf{x}_i)^T g_{\mathrm{DINO}}(\mathbf{x}_j)}{\|g_{\mathrm{DINO}}(\mathbf{x}_i)\|_2 \|g_{\mathrm{DINO}}(\mathbf{x}_j)\|_2}, \tag{26}$$

Base on this metric, we define diversity as

$$\mathrm{Div}(\mathbf{x}_1, \cdots, \mathbf{x}_N) = 1 - \mathrm{Sim}(\mathbf{x}_1, \cdots, \mathbf{x}_N). \tag{27}$$

Lastly, we consider an decreasing annealing schedule. It has been notice in previous works (Mardani et al., 2024; Zhu et al., 2024) that a decreasing timestep works better than sampling uniformly at random. As a consequence, we consider this scheme.

### D.2 SUPER RESOLUTION

We consider super resolution from $\times 8$ downsampled images. In this case we use $\lambda = 0.2$, $\tilde{\rho} = 0.05$, $l_{r_x} = 0.4$ and $l_{r_z} = 0.6$. The results are shown in Table 5. For additional comparison, we compare also with solvers that generate samples at $256 \times 256$. For this comparison, similar to Section 5.1, we downsample the result of RLSD to 256 and compare at that resolution. The results are in Table 6. This example illustrates the key difference between RLSD and PSLD for solving inverse problems. To be more specific, PSLD generates images of faces that have the typical artifacts when sampling with Stable Diffusion. On the other hand, RLSD leverages Stable Diffusion as multiple denoisers at different scale.

Table 5: SR $\times 8$ with $\sigma_v = 0.001$ - FFHQ 512. The best method for each metric and experiment is bolded.

| Sampler | PSNR [dB] ↑ | LPIPS ↓ | FID ↓ |
|---|---|---|---|
| PSLD | 24.82 | 0.314 | 81.31 |
| Latent RED-diff | 26.07 | 0.439 | 76.07 |
| NonRepuls-RLSD | **28.39** | **0.286** | **65.42** |

For completeness, we also compare with RED-Diff when solving SR $\times 4$, as it has the same resolution as input than RLSD with SR $\times 8$ (RED-Diff handles images of size 256x256). The results is Table 7.

Table 6: SR $\times$ 8 with $\sigma_v = 0.001$ - FFHQ 256. The best method for each metric and experiment is bolded.

| Sampler | PSNR [dB] $\uparrow$ | LPIPS $\downarrow$ | FID $\downarrow$ |
|---|---|---|---|
| NonRepuls-RLSD | **28.4** | **0.149** | **65.42** |
| RED-Diff | 25.69 | 0.264 | 104.59 |
| DPS | 23.83 | 0.175 | 90.45 |
| $\Pi$GDM | 24.44 | **0.128** | 82.01 |

Table 7: SR $\times$ 8 with $\sigma_v = 0.001$ - FFHQ 256. The best method for each metric and experiment is bolded.

| Sampler | PSNR [dB] $\uparrow$ | LPIPS $\downarrow$ | FID $\downarrow$ |
|---|---|---|---|
| NonRepuls-RLSD | 28.4 | **0.149** | **65.42** |
| RED-Diff (SR $\times$ 4) | **28.91** | 0.157 | 78.41 |
| DPS (SR $\times$ 4) | 27.14 | 0.128 | 71.8 |
| FPS-SMC (SR $\times$ 4) | 27.36 | 0.21 | 120.49 |

### D.3 MOTION DEBLURRING

We consider Motion Blurring. In this case we use $\lambda = 0.007$, $\tilde{\rho} = 0.01$, $l_{r_x} = 0.4$ and $l_{r_z} = 0.3$, and $L = 500$. In particular, we follow Chung et al. (2022a), where we convolve the image with a $61 \times 61$ motion kernel that is randomly sampled with intensity $0.3^2$. The results are shown in Table 8. For additional comparison, we compare also with solvers that generate samples at $256 \times 256$. For this comparison, similar to Section 5.1, we downsample the result of RLSD to 256 and compare at that resolution. The results are in Table 9.

Table 8: Motion Blurring with $\sigma_v = 0.001$ - FFHQ 512. The best method for each metric and experiment is bolded.

| Sampler | PSNR [dB] $\uparrow$ | LPIPS $\downarrow$ | FID $\downarrow$ |
|---|---|---|---|
| PSLD | 25.17 | 0.389 | 135.22 |
| Latent RED-diff | 27.85 | 0.329 | 118.09 |
| NonRepuls-RLSD | **30.4** | **0.23** | **56.79** |

Table 9: Motion Blurring with $\sigma_v = 0.001$ - FFHQ 256. The best method for each metric and experiment is bolded.

| Sampler | PSNR [dB] $\uparrow$ | LPIPS $\downarrow$ | FID $\downarrow$ |
|---|---|---|---|
| NonRepuls-RLSD | **30.47** | **0.095** | **56.79** |
| RED-Diff | 30.27 | 0.15 | 103.17 |
| DPS | 24.7 | 0.22 | 90.45 |
| ReSample | 26.82 | 0.115 | 72.74 |

### D.4 INPAINTING WITH MASKED BOX (HALF-FACE)

We consider here additional baselines for box inpainting (half-face). See results in Table 10.

Table 10: Box inpainting (half face) with $\sigma_v = 0.001$ - FFHQ 256. The best method for each metric and experiment is bolded.

| Sampler | PSNR [dB] ↑ | LPIPS ↓ | FID ↓ |
|---|---|---|---|
| NonAug-RLSD ($\gamma = 50$) | 23.34 | 0.164 | 98.65 |
| NonRepuls-RLSD | **24.98** | 0.079 | **29.18** |
| RLSD ($\gamma = 50$) | 24.69 | 0.111 | 31.41 |
| Hybrid-RLSD | 24.72 | 0.096 | 30.48 |
| ΠGDM | 23.74 | 0.077 | 33.8 |
| FPS-SMC | 24.91 | 0.086 | 59.59 |

## D.5 INPAINTING WITH RANDOM MASK

We consider random inpainting where we drop $80\%$ of the pixels. In this case we use $\lambda = 0.009$, $\tilde{\rho} = 0.08$, $l_{r_x} = 0.4$ and $l_{r_z} = 0.8$. The numerical results are shown in Table 11, and visual examples are shown in Fig. 9.

Table 11: Random inpainting with (80%) mask and with ($\sigma_v = 0.001$) - FFHQ 512. In bold is the best method for each metric and experiment.

| Sampler | PSNR [dB] ↑ | LPIPS ↓ | FID ↓ |
|---|---|---|---|
| PSLD | 28.53 | 0.212 | 65.14 |
| NonRepuls-RLSD | **30.56** | **0.145** | **41.11** |

Table 12: Random inpainting with (80%) mask and with ($\sigma_v = 0.001$) - FFHQ 256. In bold is the best method for each metric and experiment.

| Sampler | PSNR [dB] ↑ | LPIPS ↓ | FID ↓ |
|---|---|---|---|
| NonRepuls-RLSD | **30.56** | 0.073 | **41.11** |
| RED-Diff | 28.55 | 0.074 | 62.87 |
| DPS | 26.48 | 0.15 | 95.44 |
| ReSample | 27.49 | **0.062** | 54.42 |

## D.6 INPAINTING FREE MASK

We use the free masks $(10\% - 20\%)$ from Saharia et al. (2022). For this experiment, we consider ImageNet (Russakovsky et al., 2015) to demonstrate that our method outperforms its baselines on other datasets. We consider the $\lambda = 0.15$, $\tilde{\rho} = 0.15$, $l_{r_z} = 0.8$ and $l_{r_x} = 0.4$, and we consider 500 steps (instead of the full trajectory of 1000 steps) for both RLSD and PSLD. For PSLD, we use the parameters from their experiments with box inpainting (similar to the case of half face). The quantitative results are shown in Table 13, and qualitative results in Fig. 14.

Table 13: Free mask (Saharia et al., 2022) with ($\sigma_v = 0.001$) - ImageNet 512. In bold is the best method for each metric and experiment.

| Sampler | PSNR [dB] ↑ | LPIPS ↓ | FID ↓ |
|---|---|---|---|
| PSLD | 22.56 | 0.154 | 69.43 |
| NonRepuls-RLSD | **26.77** | **0.075** | **60.53** |

**Results when considering the best sample across particles.** If instead of focusing on the average performance, we focus on the performance achieved by the best image among the particle ensemble,

then the conclusion is different in favor of the RLSD; see Table 14. This can be explained as follows: while some of the modes obtained with RLSD might not be as good as NonRepuls-RLSD, others might be better.

Table 14: Box inpainting (half face) with $\sigma_v = 0.001$ - FFHQ 512. We consider the best particle across each batch of them.

| Sampler | PSNR [dB] ↑ | LPIPS ↓ | FID ↓ |
|---|---|---|---|
| PSLD (mean) | 21.34 | 0.10 | 57.7 |
| PSLD (max) | 22.72 | 0.082 | 57.7 |
| NonRepuls-RLSD (mean) | 24.98 | 0.079 | **29.18** |
| NonRepuls-RLSD (max) | 25.82 | 0.071 | 29.18 |
| RLSD ($\gamma = 50$) (mean) | 24.69 | 0.111 | 31.41 |
| RLSD ($\gamma = 50$) (max) | **25.84** | **0.069** | 27.84 |

## D.7 QUALITATIVE RESULTS

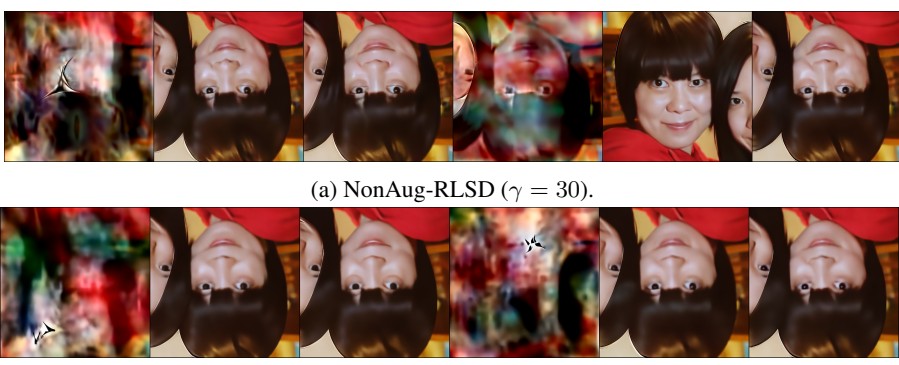

(a) NonAug-RLSD ($\gamma = 30$).

(b) Latent RED-Diff.

Figure 4: Phase Retrieval. Adding repulsion allows to sample from different modes.

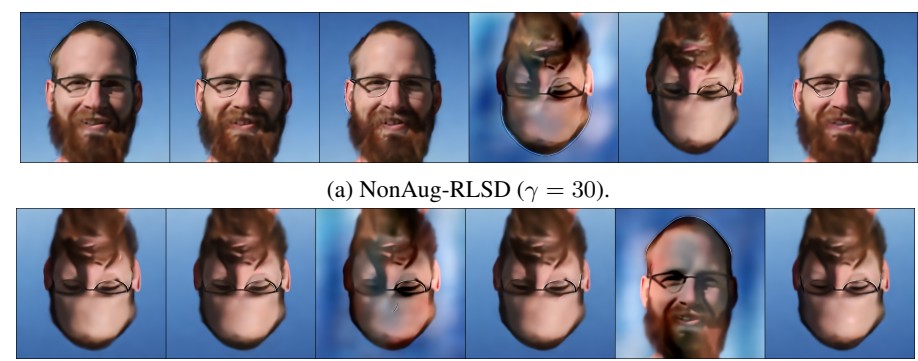

(a) NonAug-RLSD ($\gamma = 30$).

(b) Latent RED-Diff.

Figure 5: Phase Retrieval. Adding repulsion allows to sample from different modes

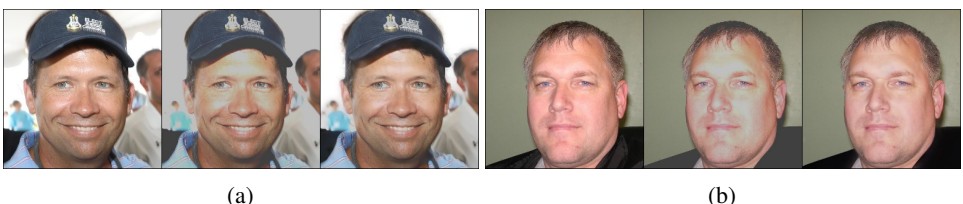

(a)          (b)

Figure 6: Two qualitative examples of HDR. Both a) and b) corresponds to NonRepuls-RLSD, and each one has Ground-truth; Measurement; Estimation. NonRepuls-RLSD generates an image of high-fidelity in a nonlinear problem at $512 \times 512$.

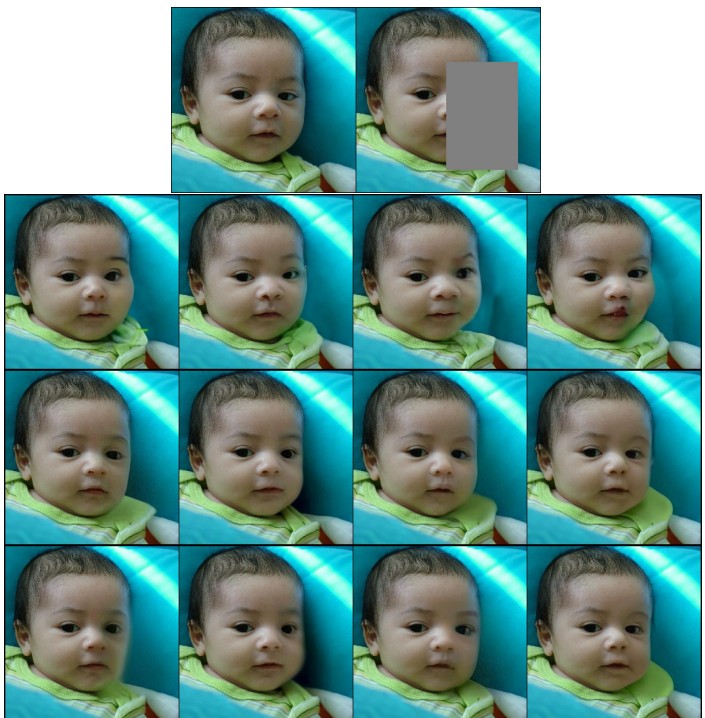

Figure 7: Inpainting half face using (from top to bottom): Ground truth and Measurement, PSLD, NonRepuls-RLSD, and RLSD ($\gamma = 50$). We generate four samples for each method from a different initialization; for RLSD, they interact through the repulsion term. First, NonRepuls-RLSD, and RLSD outperforms PSLD for all four images. Second, while images 1 and 3 from RLSD (last row) look different, the images 1 and 3 of NonRepuls-RLSD are similar; this illustrates that RLSD promotes diversity.

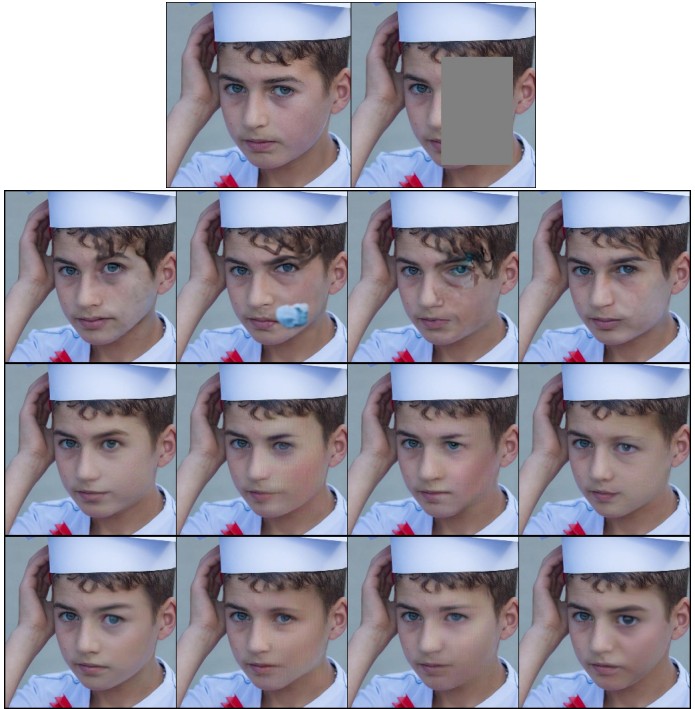

Figure 8: Inpainting half face using (from top to bottom): Ground truth and Measurement, PSLD, NonRepuls-RLSD, and RLSD ($\gamma = 50$). We generate four samples for each method from a different initialization; for RLSD, they interact through the repulsion term. First, NonRepuls-RLSD, and RLSD outperforms PSLD for all four images. Second, image 4 from RLSD (last row) looks different (has brown eye), while the four samples of NonRepuls-RLSD have blue eye; this illustrates that RLSD promotes diversity.

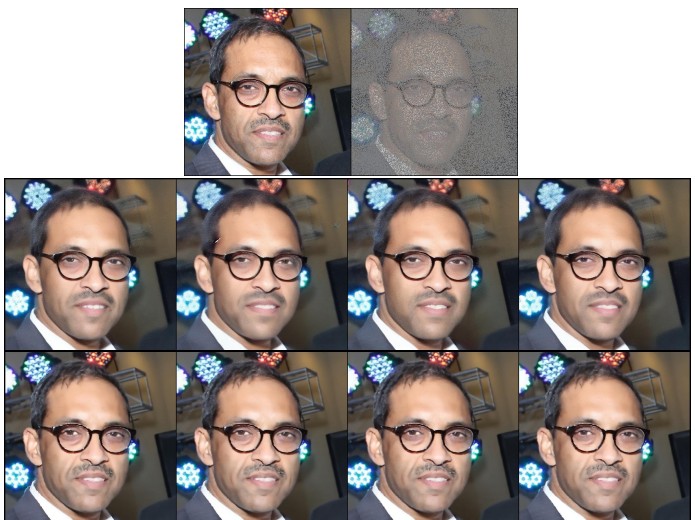

Figure 9: Random inpainting (80%): PSLD (top) and NonRepuls-RLSD (down), and four different samples for each method. Notice that NonRepuls-RLSD generates a better reconstruction of the background.

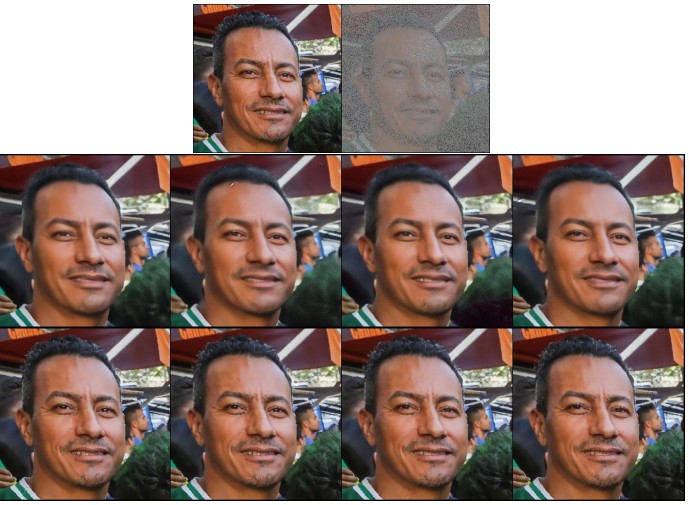

Figure 10: Random inpainting (80%): PSLD (top) and NonRepuls-RLSD (down), and four different samples for each method. Notice that NonRepuls-RLSD generates a sharper reconstruction of the face.

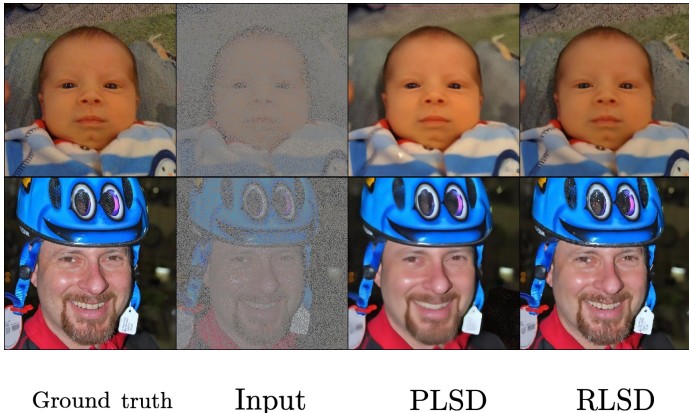

Ground truth     Input     PLSD     RLSD

Figure 11: Additional examples for random inpainting. Notice that RLSD have more details in the background for the first row, while for the second row it can reconstruct the details of the white label.

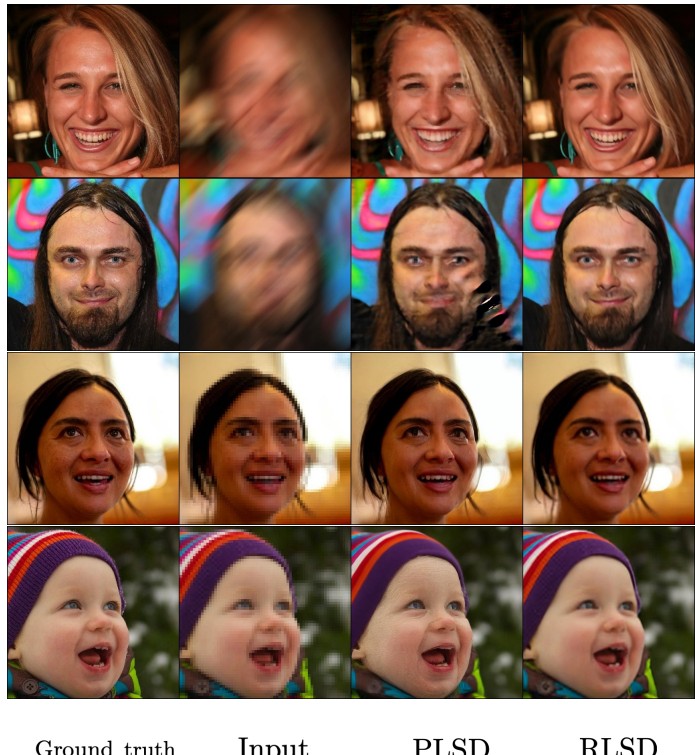

Ground truth   Input   PLSD   RLSD

Figure 12: Qualitative examples of different inverse problems for PSLD (third column) and NonRepuls-RLSD (forth column): *Rows 1 and 2* are *motion blurring*, *Rows 3 and 4* are $SR \times 8$ For motion deblurrring, RLSD clearly outperforms PLSD (PSLD reconstruct a broken image. For $SR \times 8$, PSLD introduces artifacts, which are typical when sampling faces with Stable Diffusion.

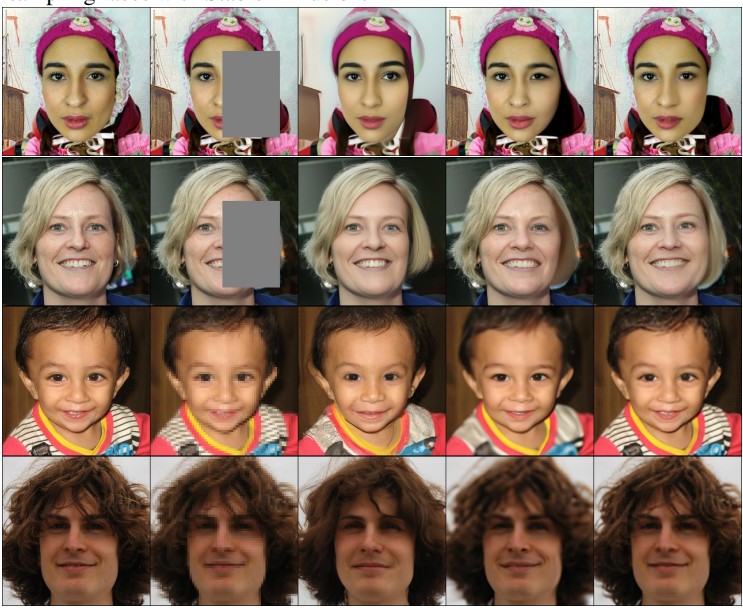

Ground truth   Input   DPS   RED-Diff   RLSD

Figure 13: Qualitative examples of different inverse problems for DPS (third column), RED-Diff (forth column) and NonRepuls-RLSD (fifth column): *Rows 1 and 2* are *half face inpainting*, *Rows 3 and 4* are $SR \times 8$. Notice that for $SR \times 8$, both DPS and RED-Diff generates images that look different than the original. This is expected considering that both operate at a resolution of $256 \times 256$. This demostrates the advantage of using a high-resolution model.

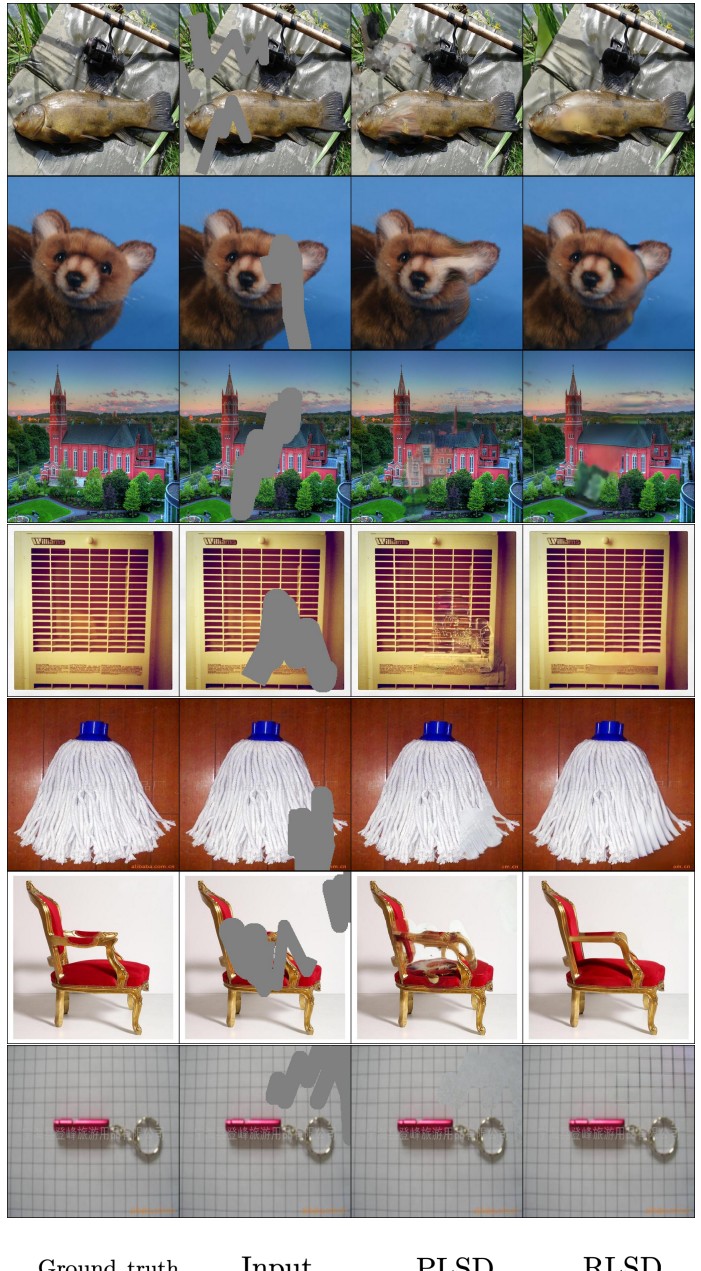

Ground truth   Input   PLSD   RLSD

Figure 14: Qualitative examples of Inpainting with free mask on Imagenet. We consider PSLD (third column) and NonRepuls-RLSD (forth column). For rows 2, 4-7, RLSD outperforms PSLD by a large margin. Furthermore, notice that RLSD can reconstruct backgrounds that might be difficult, such as the one row 4. On the other hand, RLSD struggles to reconstruct fine-details, as it is shown in row 3. We expand on this in Appendix D.9.

### D.8    ABLATION OF RLSD

#### D.8.1    RUNNING TIME/NUMBER OF STEPS

We ablate the running time between RLSD and PSLD when generating two samples for super resolution ×8. We ran the experiments on the same NVIDIA A100 GPU with 80GB. When running the full trajectory (1000 steps), the running time of RLSD without particles is 8.8 minutes, while PSLD is 9.2 minutes. When running 200 steps, the running time of RLSD without particles is 1.75 minutes, while PSLD is 1.9 minutes. The results for this case are shown in Fig. 15. Notice that although the running time difference between RLSD and PSLD is not large, our formulation leverages the diffused trajectory as a denoiser. Therefore, we require fewer steps to obtain a high-fidelity estimation; for instance, for super resolution we need just 200 steps. This showcases that our method works fine with just a few number of steps; see also Fig. 16.

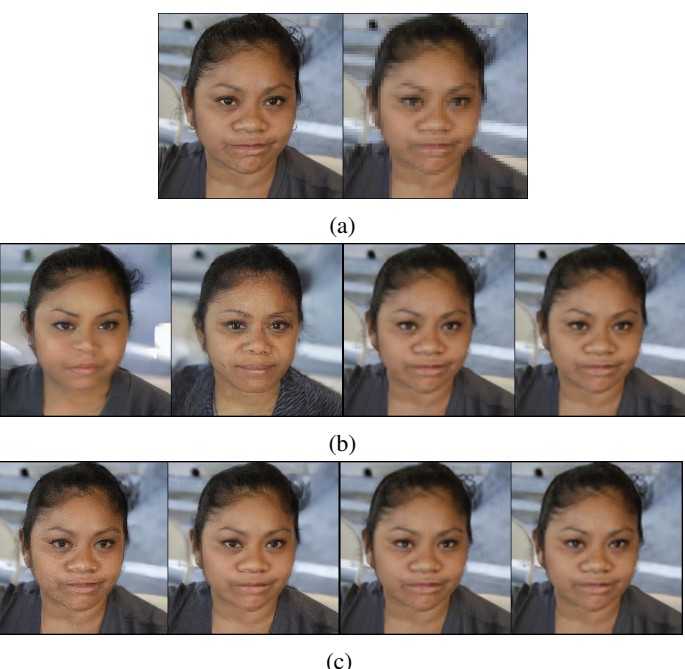

Figure 15: Reconstruction for different number of steps. a) Ground-truth (left) and Measurement (right). b) The two on the left are the reconstruction using PSLD with 200 steps (running time = 1.9 min), while the two of the right are with RLSD ($\gamma = 0$) (running time = 1.75 min). c) The two on the left are the reconstruction using PSLD with 1000 steps (running time = 9.2 min), while the two of the right are with RLSD ($\gamma = 0$) (running time = 8.8 min).

Compared to ReSample Song et al. (2024), RLSD is also computationally more efficient. Specifically, ReSample involves an inner optimization within the sampling. Consequently, when using LDM (256x256 resolution), it takes 4 minutes in one A100 GPU to generate 1 image. Moreover, when we tried Stable Diffusion, it took 16 minutes (considering the same hyperparameter configuration). We hypothesize that ReSample takes a long time with Stable Diffusion due to the optimization step, where it might struggle to converge at every step. While this might be improved, it demonstrates that the method needs more fine-tuning than ours. On the other hand, our method, using Stable Diffusion as prior, can generate 4 samples with just 200 steps in less than 4 minutes (without repulsion, 1 image per minute) and 5 minutes (with full repulsion).

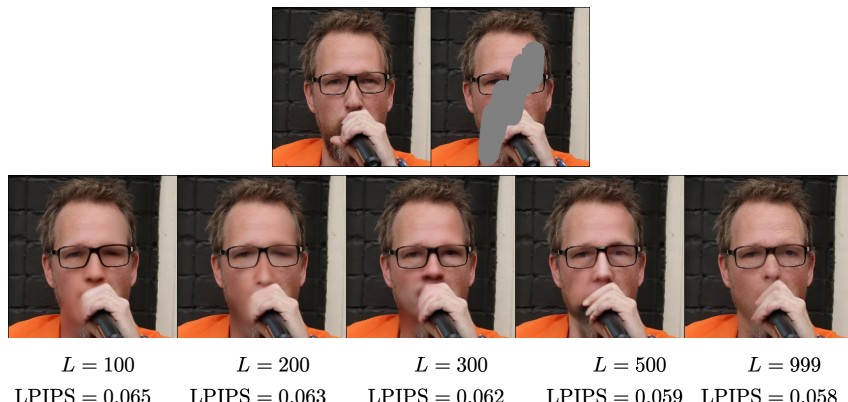

| $L = 100$ | $L = 200$ | $L = 300$ | $L = 500$ | $L = 999$ |
| :---: | :---: | :---: | :---: | :---: |
| LPIPS = 0.065 | LPIPS = 0.063 | LPIPS = 0.062 | LPIPS = 0.059 | LPIPS = 0.058 |

Figure 16: Reconstruction as a function of $L$ (number of steps in the optimization). Increasing $L$ modifies the number of denoisers that are used (the limits of the interval are the same, and it changes the step-size). Also, we need to decrease $l_{r_z}$ (from 1 to 0.8) as well the coupling term $\tilde{\rho}$, from 0.15 to 0.1.

### D.8.2 REPULSION TERM

**Effect of $\gamma$.** We showcase an example of the effect of $\gamma$. We consider $N = 4$ particles and 200 steps, $\tilde{\rho} = 0.1, \lambda_t = 0.15, l_{r_x} = 0.4$ and $l_{r_z} = 1$. It is important to remark that when considering fewer steps, we need a higher $\gamma$ (compared to the example in Section 5.1; the reason why this happens is due to the annealing factor in the repulsion term (13), given by $\alpha(t)$. The example for $\gamma = 0, 100, 150, 200$ is shown in Fig. 17. Also, in Table 15 we present quantiative results.

Table 15: Ablation $\gamma$ for inpainting with free-mask.

| $\gamma$ | **PSNR [dB]** ↑ | **LPIPS** ↓ | FID | Diversity ↑ |
| :--- | :---: | :---: | :---: | :---: |
| 0 | 30.03 | 0.064 | 22.62 | 0.004 |
| 25 | **30.04** | **0.064** | **22.21** | **0.005** |
| 50 | 30 | 0.064 | 22.31 | 0.005 |

**Effect of repulsion on a fixed interval.** Depending on the downstream task, we can include repulsion only on a fixed interval. For instance, this is the case of Phase Retrieval, where modes are discrete and isolated. Therefore, once each particle get around one of the mode (different), we can turn-off the repulsion, which yields a faster solver. For the case of inpainting, we show in Fig. 18 an ablation changing $t_{\text{repul}}$ where $t \in [0, t_{\text{repul}}]$ and in Fig. 19 when $t \in [t_{\text{repul}}, T]$ (with $T = 200$).

We observe that for diversity, it is more important to have repulsion at the beginnig of the sampling, which corresponds to the higher noise levels. Intuitively, it is more important to impose repulsion in the high-level semantics of the image instead of the fine-details. However, adding some repulsion later in the sampling process might improve some details (see the case $t_{\text{repul}} \in [100, 200]$ in Fig. 19).

**Effect of number of particles.** Lastly, we do an ablation when considering more particles in the repulsion term. Given four particles, we consider three settings: $N = 0$ (four independent particles), $N = 2$ (two independent particles and two interacting particles) and $N = 4$ (four interacting particles). We use again the free mask (Saharia et al., 2022), and $L = 200$. In Table 16 we show quantitative results, while in Figs. 20 and 21 we show two qualitative results. In both cases for $N = 2$, the last two columns correspond to the two i.i.d. particles. Consequently, those two images can change when considering $N = 4$, while the first two columns can change when moving from $N = 0$ to two. In particular, in Fig. 20 with $N = 2$ we incearse diversity, while in Fig. 21 we need at least $N = 4$.

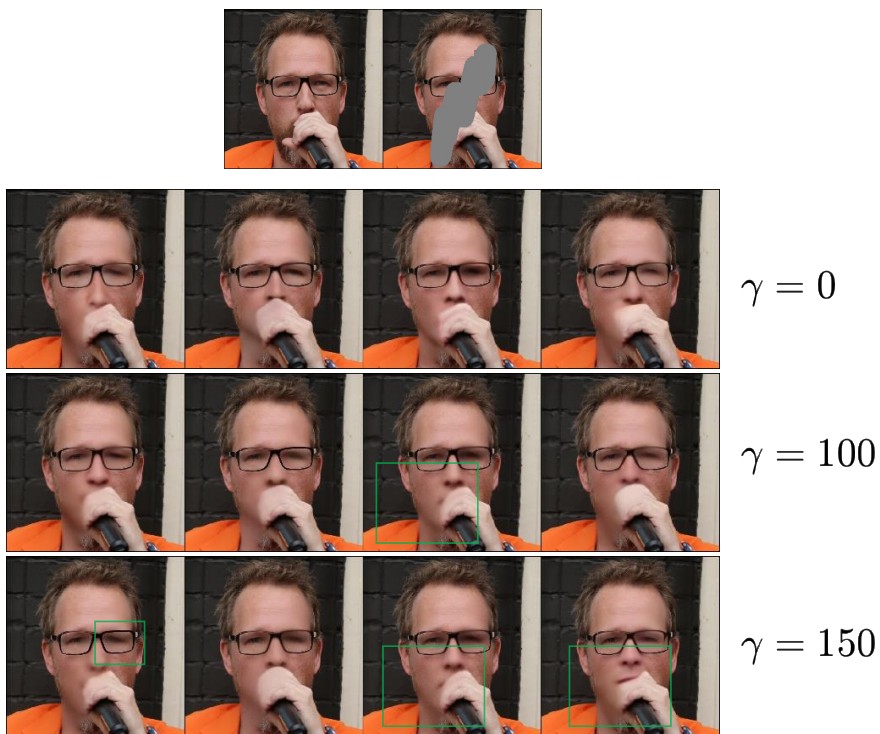

Figure 17: Reconstruction as a function of $\gamma$ (number of repulsion), for $L = 200$ and $N = 4$. Increasing $\gamma$ enhances diversity: for $\gamma = 150$, 2 out of 4 samples show the mouth, while for $\gamma = 0$ all samples have the mouth hidden.

Table 16: Ablation when changing the number of particles in the repulsion term with $\sigma_v = 0.001$ - FFHQ 512.

| Sampler | PSNR [dB] ↑ | LPIPS ↓ | FID ↓ | Diversity |
|---|---|---|---|---|
| NonRepuls-RLSD ($N = 0$) | **30.03** | 0.064 | **65.42** | 0.002 |
| RLSD ($N = 2$) | 29.99 | 0.064 | 71.8 | 0.004 |
| RLSD ($N = 4$) | 29.99 | **0.063** | 69.91 | 0.005 |

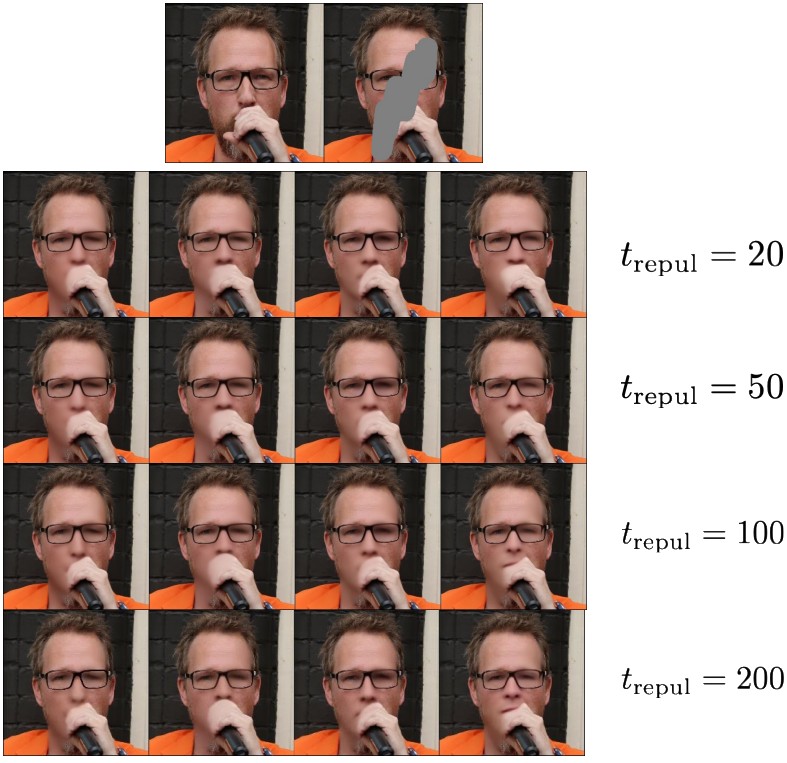

Figure 18: Reconstruction for $\gamma = 150$, for $L = 200$, $N = 4$ and repulsion between $0$ and $t_{\mathrm{repul}}$: when $t_{\mathrm{repul}} = T$, we have repulsion in all the trajectory, while $t_{\mathrm{repul}} = 0$ corresponds to NonRepuls-RLSD.

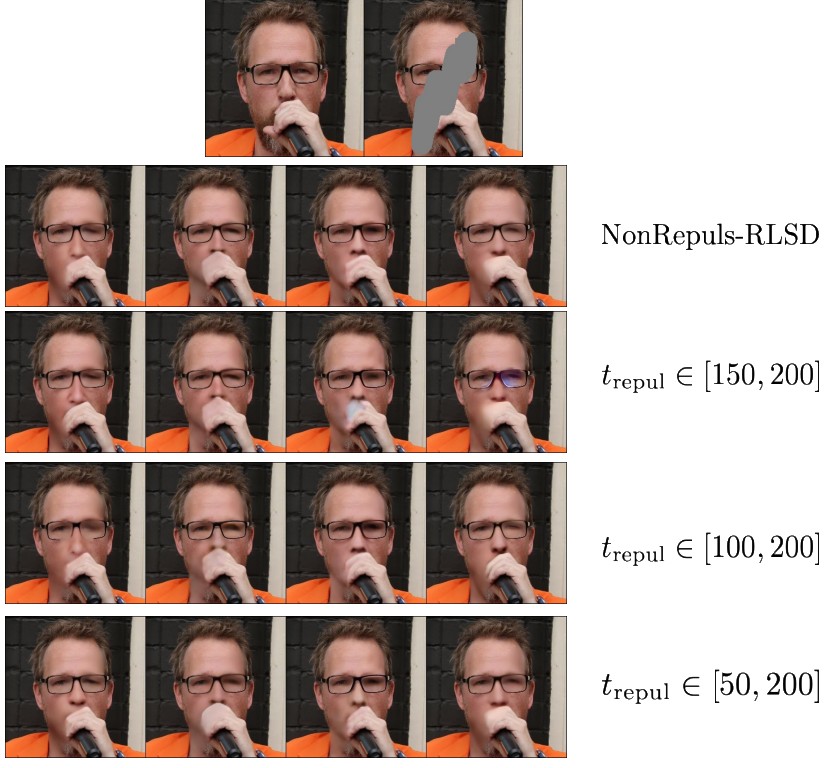

Figure 19: Reconstruction for $\gamma = 150$, for $L = 200$, $N = 4$ and repulsion for $t \in [t_{\mathrm{repul}}, 200]$; notice that $t_{\mathrm{repul}} = T$ corresponds to NonRepuls-RLSD.

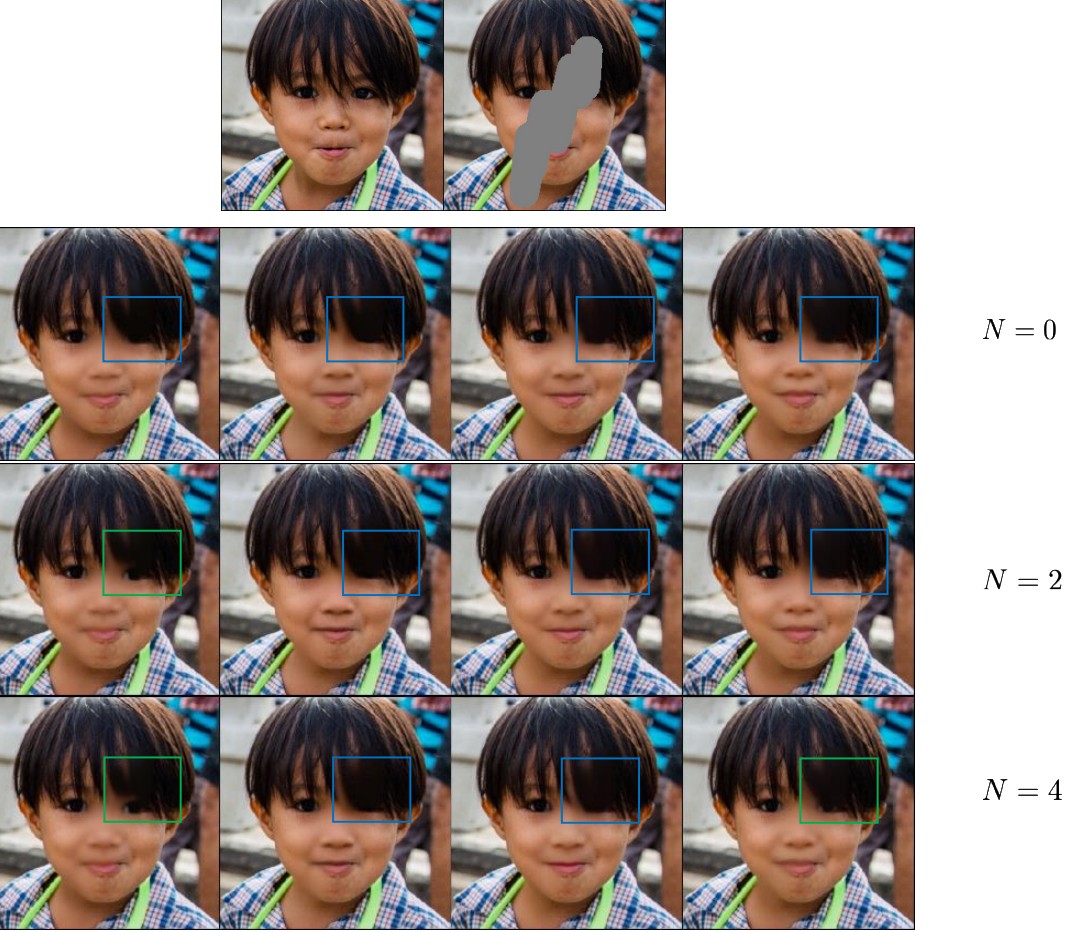

Figure 20: Reconstruction when increasing the number of particles in the repulsion, for $N = 0, 2$ and $4$. Notice that $N = 0$ corresponds to the NonRepuls-RLSD, $N = 2$ to Hybrid case, and $N = 4$ to RLSD.

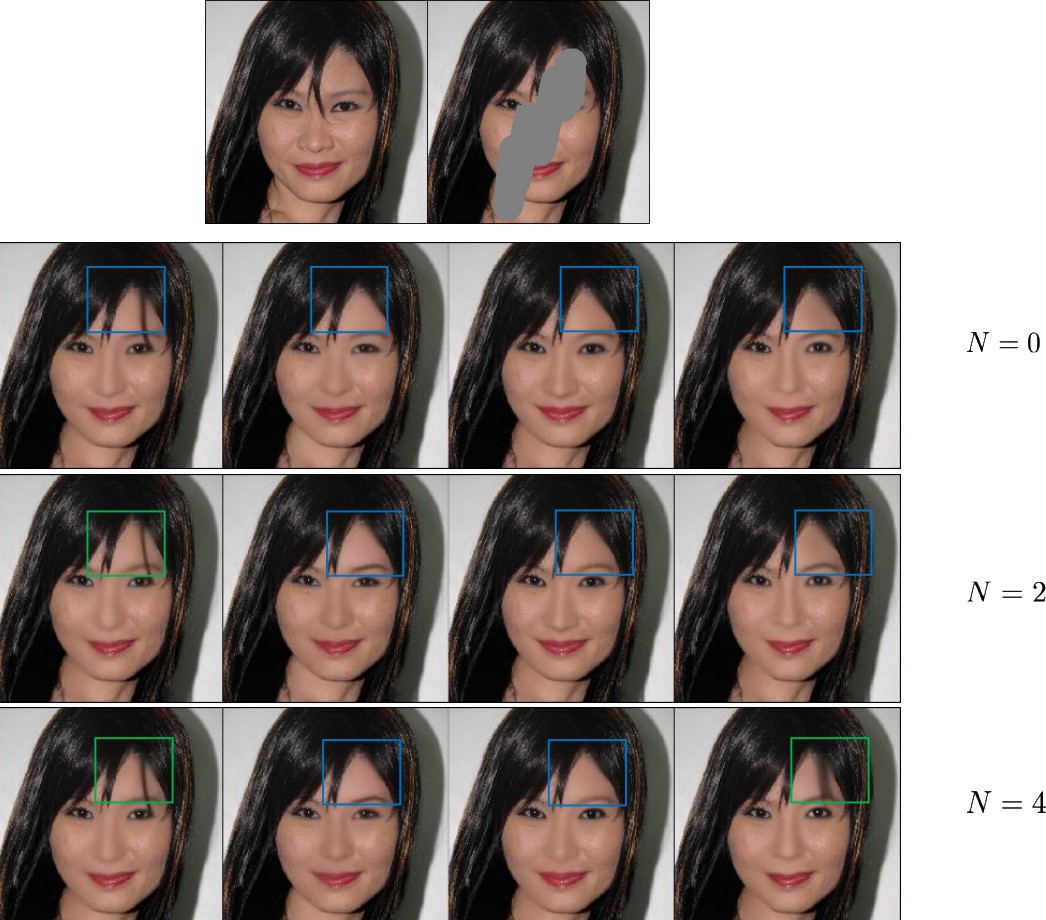

Figure 21: Reconstruction when increasing the number of particles in the repulsion, for $N = 0, 2$ and $4$. Notice that $N = 0$ corresponds to the NonRepuls-RLSD, $N = 2$ to Hybrid case, and $N = 4$ to RLSD. In this case, we need full repulsion to increase diversity w.r.t. the NonRepuls-RLSD case.

### D.8.3   COMPARISON BETWEEN NONAUG-RLSD VS RLSD

We compare in Fig. 22 NonAug-RLSD vs RLSD. Notice that RLSD achieves a sharper reconstructed image; in particular, the shirt for the case of NonAug-RLSD has not fine-detail, while RLSD has.

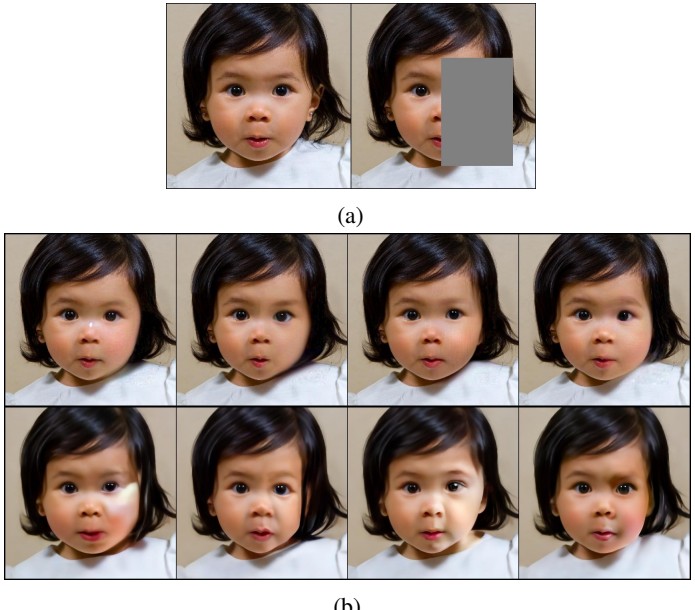

Figure 22: Comparison between RLSD (top row of Fig. b) and NonAug-RLSD (last row of Fig. b). RLSD achieves a sharper reconstructed image; in particular, the shirt for the case of NonAug-RLSD has not fine-detail, while RLSD has.

### D.8.4   COMPARISON WITH IMAGES FROM IMAGENET BETWEEN RLSD AND RED-DIFF WITH DIFFUSION PRIOR TRAINED ON FFHQ

Here we show some examples of reconstruction using out-of-distribution images; we use samples from ImageNet (Russakovsky et al., 2015). In Fig. 23 we show an example from ImageNet, where we compare RLSD with RED-diff using FFHQ. Clearly, the performance of RLSD is better than RED-diff. This is expected given that the diffusion model of RED-diff is with FFHQ. However, this demonstrates that using more powerful diffusion model as prior enables to deploy our model for different type of images.

### D.8.5   COUPLING PARAMETER

We do an ablation of the coupling parameter $\tilde{\rho}$. We consider three values: $\tilde{\rho} = \{0.03, 0.07, 0.12\}$. The qualitative results for this case are shown in Fig. 24, and the quantiative in Table 17 As expected, increasing the coupling parameter yields a more diverse set of images. The reason why this happens is because the prior (through $\mathbf{z}$) has more weight when estimating the image $\mathbf{x}$. However, this increase in diversity is penalized by a lower performance. On the other hand, a lower $\rho$ generates four images that look very similar. It is important to note that our repulsion term is a better trade-off in terms of diversity/quality.

Table 17: Ablation $\rho$ for inpainting with free-mask - $\gamma = 0$

| $\rho$ | PSNR [dB] ↑ | LPIPS ↑ | FID | Diversity ↑ |
|---|---|---|---|---|
| 0.03 | 29.94 | 0.069 | 25.57 | 0.01 |
| 0.07 | **30.03** | **0.064** | **22.62** | 0.002 |
| 0.12 | 28.6 | 0.065 | 23.25 | **0.006** |

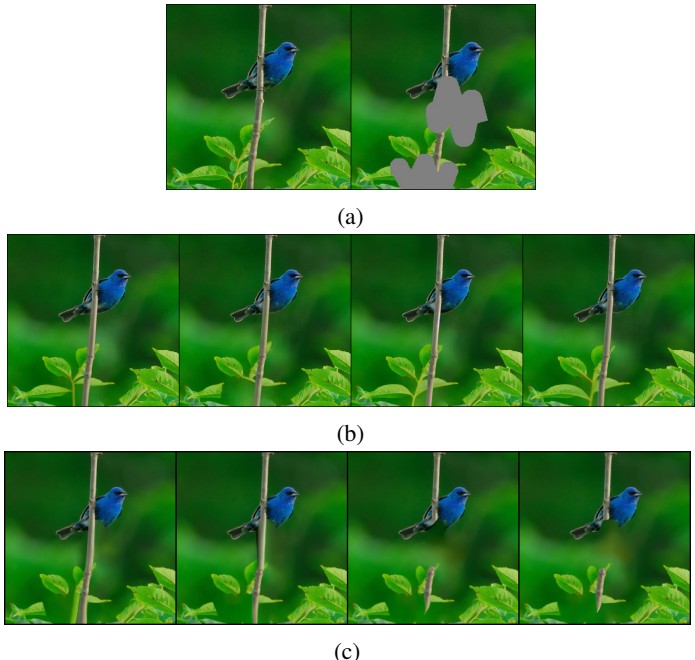

Figure 23: Results of the reconstruction of an image from the validation set of ImageNet using RLSD (b) and RED-diff with diffusion prior trained on FFHQ (c). This example showchases that using a large-pre trained model such as Stable Diffusion enables to use our method with images from very different classes (Imagenet and FFHQ).

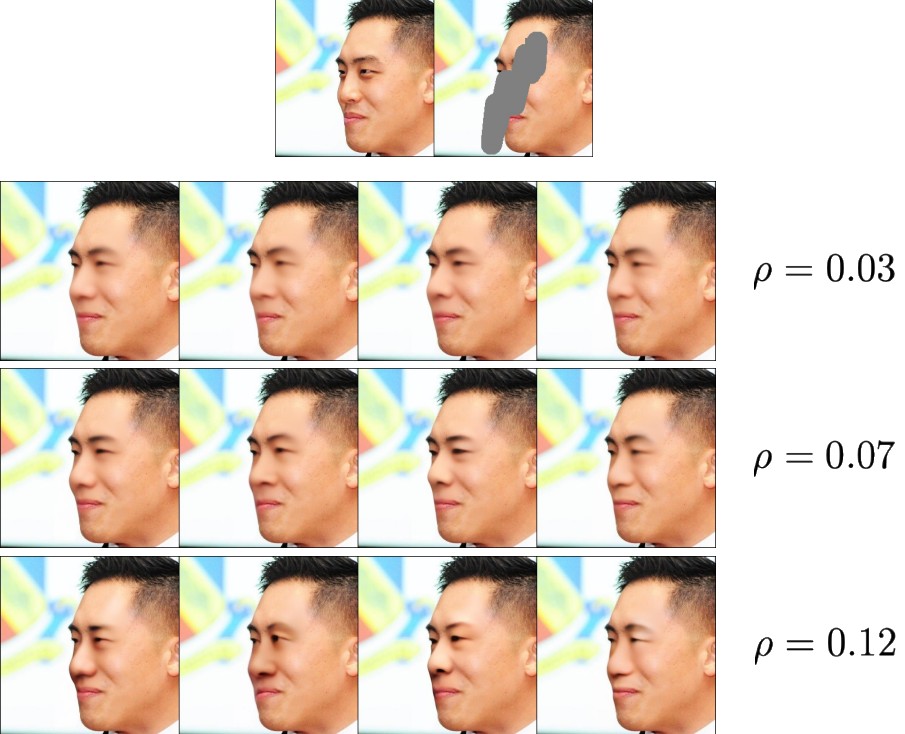

Figure 24: Reconstruction as a function of $\rho$ (coupling parameter). Increasing $\rho$ enhances diversity but reduces quality: for $\rho = 0.12$, the 4 samples looks different but with a low quality, while with a lower $\rho$, the diversity is lower the performance is better.

### D.9    LIMITATIONS AND FAILURE SETTINGS

Throughout the experiments, we demonstrated that RLSD outperforms other baselines, particularly PSLD. Additionally, we show that NonAug-RLSD is capable of solving nonlinear inverse problems at high resolutions. However, RLSD still faces challenges in certain settings and inverse problems. Below, we highlight some of these cases, which we aim to address in future work.

**Lack of fine details in ImageNet.**    When performing inpainting on ImageNet, we observe that our method struggles to reconstruct fine details.

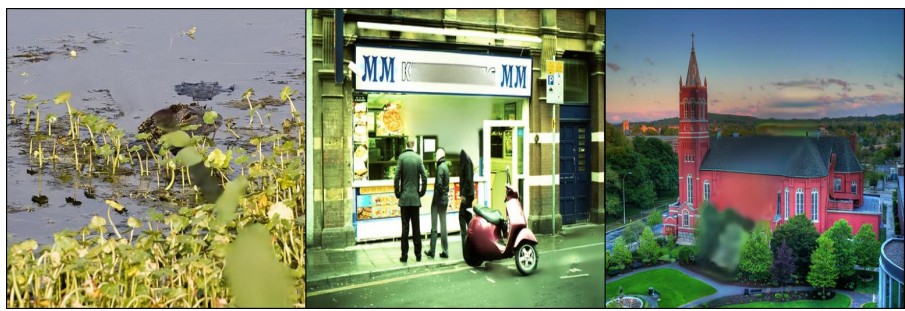

Figure 25: Example of a setting where our method struggles to have a high-quality reconstruction. In particular, RLSD generate images without small details: for instance, the building does not have windows in the walls.

While this might be circumvent by increasing the weight of the prior, our simulations did not work.

**Full-face on FFHQ.**    When we seek to reconstruct the face of a person, we observe that our method might struggle based on the background. For instance, the samples from Fig. 26 look blurry and unnatural. On the other hand, in Fig. 27 we observe a case that has a more natural reconstruction. Still, it has a lack of details in some parts of the face (eyebrow for example).

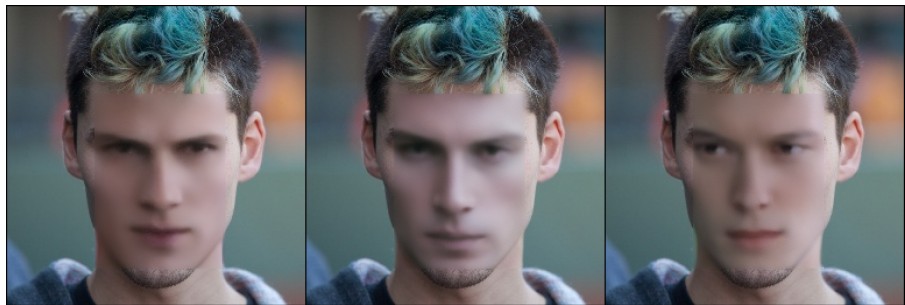

Figure 26: Example of a setting where our method struggles to have a high-quality reconstruction. In particular, RLSD generate a face that looks unnatural.

When applying the same box to ImageNet, we observe the same as it is shown in Fig. 28.

**Repulsion too high.**    Lastly, an important setting where RLSD fails is when $\gamma$ is too high. As an example, we show in Fig. 29 a case where $\gamma = 200$, and the same setting as in Appendix D.8. While in Appendix D.8.2 we illustrated that increasing $\gamma$ enhances diversity, if $\gamma$ is too high, then the reconstruction fails.

**Fixed resolution given by the diffusion prior.**    The dimension of the images generated with the diffusion prior is constrained by the existing pretrained diffusion priors (currently, Stable Diffusion provides resolutions up to $512 \times 512$). This is still considred a high dimensional data, compared with existing methods that work with resolutions up to $256 \times 256$. Extending our framework to handle higher dimensions than the one of the diffusion prior is an interesting research direction, where ideas from compositional generation might be helpful. We leave this as future work.

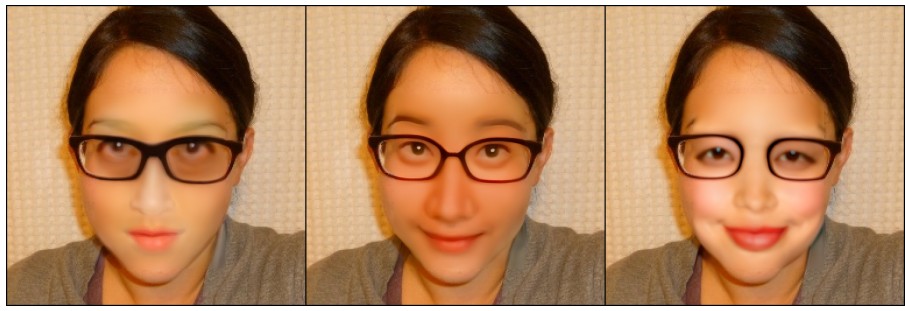

Figure 27: Example of a more natural reconstruction for full-face example. In particular, RLSD generate a face that looks more natural than Fig. 26. Still, the eyebrow looks blurry.

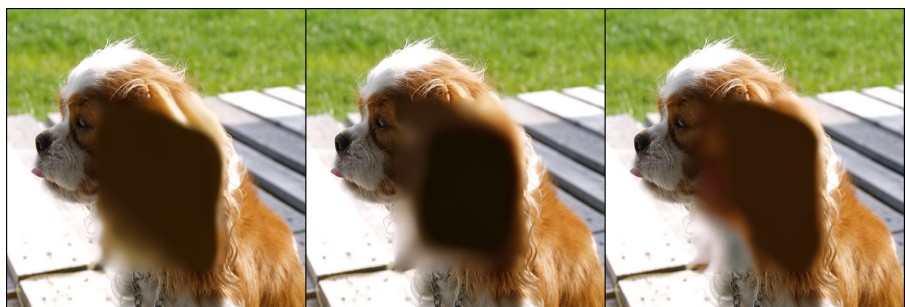

Figure 28: Example from ImageNet, where we see that there is a lack of details.

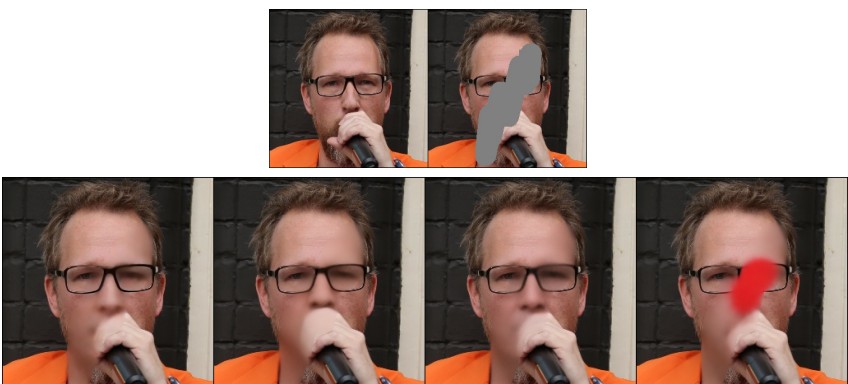

Figure 29: Example of failure when considering $\gamma$ too high. We consider $N = 4$ (the second row), $L = 200$, and the same setting as in Appendix D.8.

### D.10 TRADE-OFF BETWEEN DIVERSITY AND QUALITY IN UNCONSTRAINED GENERATION

To demonstrate the generality of RLSD and our particle-based variational approximation, we also include additional experiments when considering unconstrained sampling: we include results for the unconstrained case, i.e., text-to-image and text-to-3D. Details about Score distillation and the methods used in this section can be found in Appendix E.

#### D.10.1 TOY EXAMPLE WITH A BIMODAL GAUSSIAN DISTRIBUTION

We consider here a toy example to showcases how the $\gamma$ parameter (the amount of repulsion) dictates the trade-off between diversity and quality. We consider a mixture of two Gaussians of parameters $\mathcal{N}_1([1, 0]^\top, 0.005\mathbf{I})$ and $\mathcal{N}_2([-1, 0]^\top, 0.005\mathbf{I})$, and we consider two settings:

1. Two independent Gaussians with $\sigma \to 0$ where we fit the mean parameters

2. Two dependent Gaussians with $\sigma \to 0$ where we fit the mean parameters and coupled via an Euclidean RBF kernel.

For each setting, we compute 200 realizations with the same seed. In Fig. 30a we show how many realizations suffers from mode collapse ($\gamma = 0$ corresponds to the first setting), while in Figs. 30b and 30c we show a realization for $\gamma = 1$ and $\gamma = 2000$, where we observe that the "quality" of the estimation for this high value is poorer compared to $\gamma = 1$.

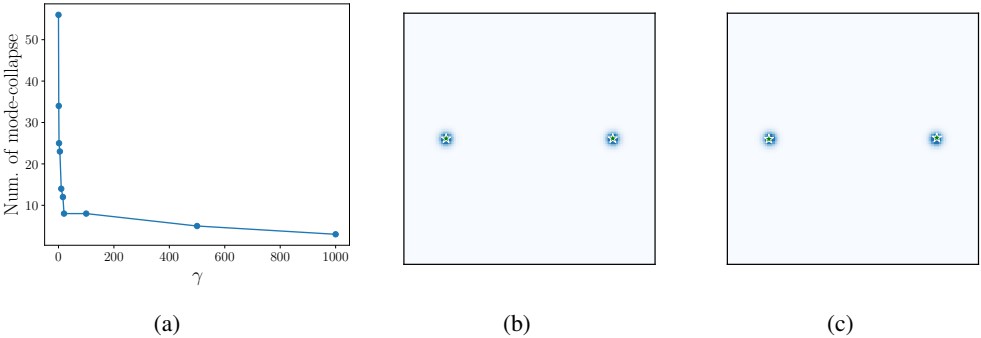

(a)        (b)        (c)

Figure 30: a) Number of realizations that has a particle collapse to the same mode as a function of the amount of repulsion that we consider. Estimation when considering b) $\gamma = 1$, c) $\gamma = 2000$.

### D.10.2   TEXT-TO-IMAGE GENERATION: TRADE-OFF PLOT AND ABLATION OF KERNEL FUNCTION

**Implementation details.** Here we describe the details of the experiments on 2D images with ProlificDreamer (VSD) (Wang et al., 2024), which is described in Appendix E. VSD is a particular case of score distillation for unconstrained sampling. We optimize 500 optimization steps, follow their setup, and we train a U-Net from scratch to estimate the variational score; we used the code from the public repository `https://github.com/yuanzhi-zhu/prolific_dreamer2d`. We consider ADAM optimizer and set the learning rate of particle images is 0.03 and the learning rate of U-Net is 0.0001. We consider only 4 particles as it is enough to promote diversity (in particular when considering DINO as feature extractor). The images and parameters are initializad at random. We run all the experiments in a single NVIDIA A100 GPU of 80GB.

**Trade-off plot.** We study the trade-off between diversity and quality when increasing the amount of repulsion. We consider 75 images from the COCO dataset and compute average qualities and diversities across all images. We also include a comparison with stochastic sampling using the Euler discretization of the backward process, with 30 steps (denoted by 'Ancestral'). This serves as an upper bound on the performance of distillation techniques. We consider 4 particles for both the distillation optimization and the sampling via Euler. The results are shown in Fig. 31, where we compute FID (lower is better), Aesthetic (higher is better) and CLIP (higher is better) scores as a function of a diversity score (27); a higher diversity score corresponds to more diverse image generation. For the plot, we fix all the hyperparameters, and we sweep $\gamma$ between 0 (ProlificDreamer) and 40, and we consider Ancestral sampling as an upper bound in terms of trade-off. Clearly, adding repulsion increases the diversity, while slightly decreasing the quality metrics for low values of $\gamma$. However, for these lower values of $\gamma$, the diversity is still far from the stochastic sampling. When increasing the values of $\gamma$ above 30, we close the gap in terms of diversity at the cost of decreasing the quality (FID and Aesthetic) as well as the text-alignment (CLIP).

**Using different domains/distances for the repulsion force** We compare the RBF when considering different domains, namely Euclidean, DINO and LPIPS.

1. **RBF in the Euclidean domain.** We compare between DINO and RBF for a repulsion with $\gamma = 10$ with two qualitative examples in Figs. 32 from COCO validation set (Lin et al., 2014) constructed in (Jain et al., 2022).

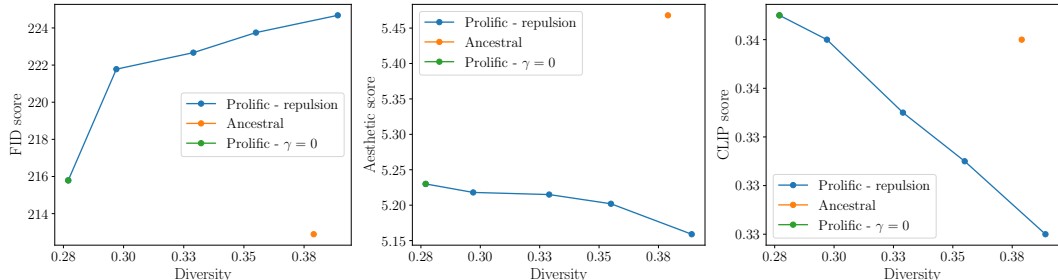

Figure 31: We consider the trade-off between diversity and quality for text-to-2D image generation using score distillation. We consider $\gamma = \{0, 10, 20, 30, 40\}$. From left to right: Diversity (27) vs left) FID, middle) Aesthetic score and right) CLIP score. We consistently observe that adding repulsion increases diversity, while slightly decreasing the quality of the images. Notice that we need a repulsion with $\gamma > 30$ to reach the diversity level of stochastic sampling using Euler.

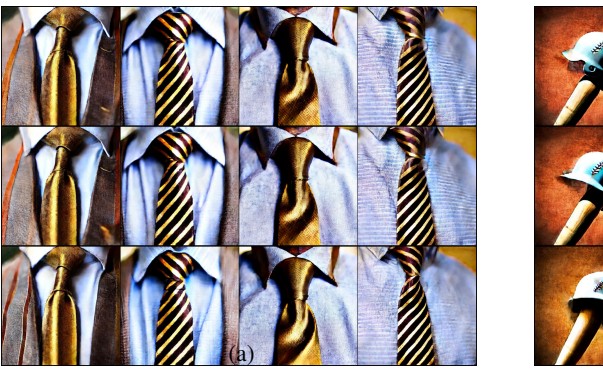 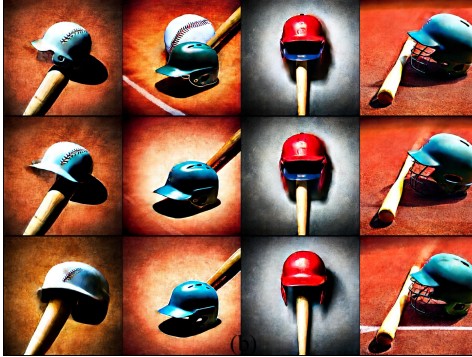

Figure 32: Comparison between RBF with Euclidean distance and DINO. We generate samples using the prompt a) "a gold tie is tied under a brown dress shirt with stripes..", b) "a baseball bat with a batting helmet upsidedown.". From the top to bottom: $\gamma = \{0, 10(\text{Euclidean}), 10(\text{DINO})\}$ with CFG = 7.5 and 500 steps.

2. **RBF using LPIPS.** We also tried LPIPS as metric in RBF. However, we observe that when combined with ProlificDreamer, the generated images have some artifacts or are not well aligned with the text prompt; two examples are shown in Figs. 33.

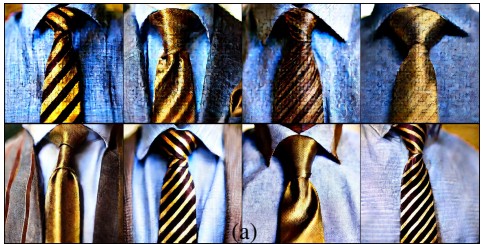 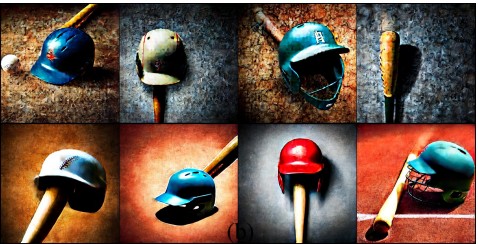

Figure 33: Comparison between RBF with LPIPS (top row) and DINO (bottom row). We generate samples using the prompt a) "a gold tie is tied under a brown dress shirt with stripes..", b) "a baseball bat with a batting helmet upsidedown.". From the top to bottom: $\gamma = \{10(\text{LPIPS}), 10(\text{DINO})\}$ with CFG = 7.5 and 500 steps.

### D.10.3 TEXT-TO-3D GENERATION

Lastly, we demonstrate how our method improves the mode collapse phenomena in text-to-3D generation. Our main motivation is to show that including the repulsion entails a more diverse set of scenes when changing the seed.

**Implementation details.**    We consider DreamFusion (Poole et al., 2022) as base method and the implementation from the Threestudio framework (Guo et al., 2023). All 3D models are optimized for 10000 iterations using Adam optimizer with a learning rate of 0.01, and we use the same configuration as DreamFusion for the rendering. For the NeRF architecture we use an MLP, and we use the same configuration from the setting in Threestudio. For the diffusion model, we use DeepFloyd-IF-XL-v1.0 with a guidance scale of 20. For the repulsion, we use a repulsion weight of $\gamma = 200\sigma_t$. We use one Nvidia A100 of 80GB, and we consider a batch of 20 particles; this presents a limitation for using more computationally-demand methods such as ProlificDreamer. For the RBF, we use DINO.

**Results.**    In Figs. 34 and 39 we show two qualitative examples. For each case, we show two different views of the generated object. Furthermore, we include the corresponding .gif files for the prompt "An ice cream sundae" in Figs. 31 to 34. Notice that with repulsion we can generate two different colors of glass for the ice cream (dark and white), something that we could not achieve without repulsion. Furthermore, while the case without repulsion generates two scenes that have a similar perspective, our method generates two scenes that show the ice cream at different distances. Lastly, notice that the quality of the generated scene is bounded by the performance of the base method (in this case, DreamFusion). Therefore, the results look saturated, similar to what happen with SDS.

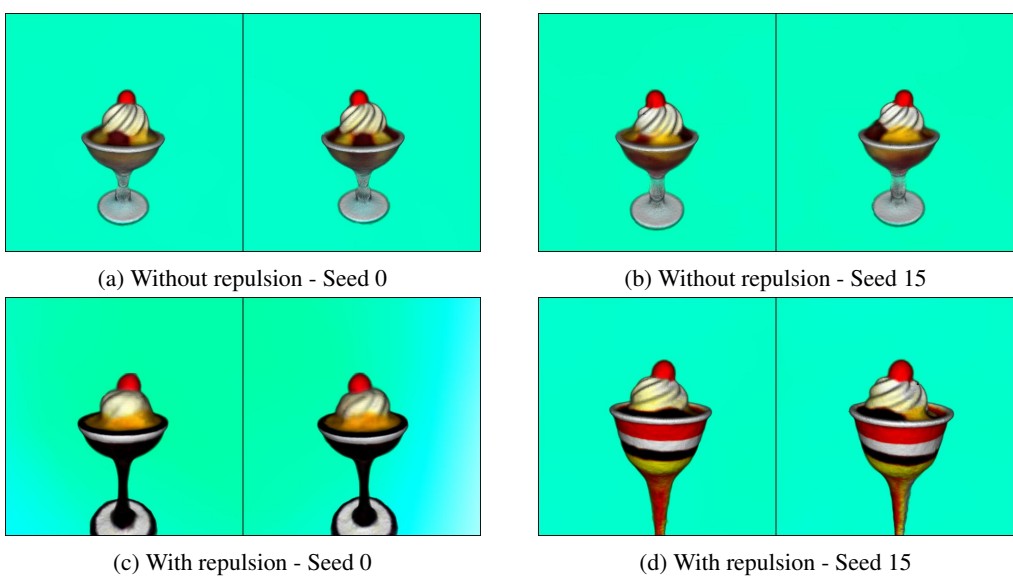

| (a) Without repulsion - Seed 0 | (b) Without repulsion - Seed 15 |
|---|---|
| (c) With repulsion - Seed 0 | (d) With repulsion - Seed 15 |

Figure 34: Text-to-3D generation using DreamFusion with the prompt "An ice cream sundae", and considering a batch of 20 samples (particles) a, b) without repulsion with seed 0 and seed 15, c, d) with repulsion with seed 0 and 15 respectively. For each case, we show two different views of the object. Clearly, the two cases without repulsion look very similar, generating the same type of ice cream sundae. On the other hand, adding repulsion increase diversity of the scene (a different glass, and color). However, this comes at the cost of less details in the ice cream.

Figure 35: Without repulsion - Seed 0                     Figure 36: With repulsion - seed 0

Figure 37: Without repulsion - seed 15

Figure 38: With repulsion - seed 15

(a) Without repulsion - Seed 1

(b) Without repulsion - Seed 35

(c) With repulsion - Seed 1

(d) With repulsion - Seed 35

Figure 39: Text-to-3D generation using DreamFusion with the prompt "a bulldozer made out of toy bricks", and considering a batch of 20 samples (particles) a, b) without repulsion with seed 1 and seed 35, c, d) with repulsion with seed 1 and 34 respectively. For each case, we show two different views.

## E  SCORE DISTILLATION

Score distillation sampling (Poole et al., 2022) was the first proposed method to optimize a generator by using distillation. Consider a pretrained score function $\epsilon_{\boldsymbol{\theta}}(\mathbf{x}_t, t) \approx -\sigma_t \nabla_{\mathbf{x}_t} \log p(\mathbf{x}_t)$ representing a distribution of interest $p_{\boldsymbol{\theta}}$. The idea of *score distillation sampling* (SDS) is to train a generator $g_{\boldsymbol{\phi}}$, such that the output of the generator given an input $\mathbf{m}$ is a sample $\mathbf{x}_0 = g_{\boldsymbol{\phi}}(\mathbf{m}) \sim p_{\boldsymbol{\theta}}$. This is achieved by optimizing the distillation loss (Poole et al., 2022)

$$\mathcal{L}_{\text{SDS}}(\mathbf{x}_0 = g_{\boldsymbol{\phi}}(.)) = \mathbb{E}_{t \sim \mathcal{U}[0,T], \boldsymbol{\epsilon} \sim \mathcal{N}(0,\mathbf{I})} \left[ \omega(t) \frac{\sigma_t}{\alpha_t} \text{KL} \left( q(\mathbf{x}_t | \mathbf{x}_0 = g_{\boldsymbol{\phi}}(.)) \, \| \, p_{\boldsymbol{\theta}}(\mathbf{x}_t) \right) \right], \quad (28)$$

where $\omega(t)$ is a weighting function and $q(\mathbf{x}_t | \mathbf{x}_0 = g_{\boldsymbol{\phi}}(.))$ is the variational distribution; when $g_{\boldsymbol{\phi}}(.) = \phi$, i.e., identity mapping, then we are in the case our proposed method in Section 4.

Although its remarkable success in generating 3D scenes, these methods suffer from a mode collapse problem. This is mainly driven by the choice of the (reverse) KL divergence as the loss in (28) and the fact that a *unimodal variational distribution q is used*. In particular, they need to consider a high CFG in the classifier-free guidance score (Ho and Salimans, 2021)

$$\epsilon_{\boldsymbol{\theta}}^{w} (\mathbf{z}_t; c, t) = \epsilon_{\boldsymbol{\theta}} (\mathbf{z}_t; c = \varnothing, t) + w (\epsilon_{\boldsymbol{\theta}} (\mathbf{z}_t; c, t) - \epsilon_{\boldsymbol{\theta}} (\mathbf{z}_t; c = \varnothing, t)) \quad (29)$$

where $\varnothing$ indicates a null condition and the $w$ is the CFG weight; they consider $w = 100$.

The mode collapse phenomenon is clear when observing SDS through the lens of gradient flows 4.1, which entails a mode-seeking optimization. Therefore, if we want to avoid mode collapse, we need to modify either the loss function (the divergence) or the variational approximation. While the former renders intractable formulations, the latter can be modified easily. Therefore, previous works focused on modifying the variational distribution. See, for instance, ProlificDreamer (Wang et al.,

2024), and more recently, in Luo et al. (2024); also other works (Katzir et al., 2024; Wang et al., 2023) have studied this problem, and proposed alternative approaches to circumvent this problem. In ProlificDreamer, the authors consider a particle approximation of the variational distribution by randomizing the parameters $\boldsymbol{\theta}$, which yields the following update rule

$$\nabla_{\boldsymbol{\theta}}\mathcal{L}_{\text{VSD}} = \mathbb{E}_{t,\epsilon,c}\left[\omega(t)\left(\epsilon_\phi(\mathbf{x}_t; c, t) - \sigma_t\nabla_{\mathbf{x}_t}\log q(\mathbf{x}_t|c)\right)\frac{\partial\mathbf{x}_0}{\partial\boldsymbol{\theta}}\right]. \tag{30}$$

Notice that $\sigma_t\nabla_{\mathbf{x}_t}\log q(\mathbf{x}_t|c)$ is unknown, so they fine-tune a pre-trained diffusion model for each particle (which represents each rendered image). Therefore, they incorporate a second optimization problem that minimizes the DSM loss; they parametrize the score network using LoRA (Hu et al., 2021). However, adding an auxiliary neural network does not necessarily promote diversity.

### E.1 Gradient Flow Perspective of Score Distillation

We can interepret the variational diffusion sampling optimization procedure as a Wasserstein gradient flow when we constraint to Bures–Wasserstein manifold.

**Wasserstein gradient flow.** WGF describes how probability distributions change over time by minimizing a functional on $\mathcal{P}(\mathbb{R}^n)$, representing the space of probability distributions over $\mathbb{R}^n$ with finite second moments. Denoted as $(\mathcal{P}(\mathbb{R}^n), W_2)$, this space employs the Wasserstein-2 distance as its metric, termed the Wasserstein space. Before delving into the WGF, we define the Wasserstein gradient of a functional $\mathcal{F}(q)$ as

$$\nabla_{W_2}\mathcal{F}(q) = \nabla_{\mathbf{x}}\frac{\delta\mathcal{F}(q)}{\delta q}. \tag{31}$$

where $\frac{\delta\mathcal{F}(q)}{\delta q} = \lim_{\epsilon\to 0}\frac{\mathcal{F}(q+\epsilon\sigma)-\mathcal{F}(q)}{\epsilon}$ is the first variation defined for any direction in the tangent space of $\mathcal{P}$. Given this definition and two boundary conditions $\rho_0 = q_0(\mathbf{x})$ and $\rho_\infty = p(\mathbf{x})$, we can define a path of densities $q_t$ where its evolution is described by the Liouville equation (also known as continuity equation)

$$\frac{\partial q_\tau}{\partial\tau} = \text{div}\left(q_t\nabla_{W_2}\mathcal{F}(q_\tau)\right) \tag{32}$$

At the particle level, for a given particle $\mathbf{x}_\tau \sim q_\tau$ in $\mathbb{R}^n$, the gradient flows defines a dynamical system drive a vector a field $\{v_\tau\}_{\tau\geq 0}$ in the Euclidean space $\mathbb{R}^n$ given by

$$d\mathbf{x}_\tau = v_\tau(\mathbf{x}_\tau)\,d\tau = -\nabla_{W_2}\mathcal{F}(q_t)(\mathbf{x}_\tau)\,d\tau.$$

Therefore, this ODE describes the evolution of the particle $\mathbf{x}_\tau$ where the associated marginal $q_\tau$ evolves to decrease $\mathcal{F}(q_\tau)$ along the direction of steepest descent according to the continuity equation in (32).

Notice that the WGF is defined in a continuous domain for $\tau$. We can discretized via the following movement minimization scheme with step size $h$, also known as the Jordan-Kinderlehrer-Otto (JKO) scheme (Jordan et al., 1998),

$$q_{k+h} = \underset{q\in\mathcal{P}(\mathbb{R}^n)}{\text{argmin}}\left\{\mathcal{F}(q) + \frac{1}{2h}W_2^2(q, q_k)\right\}, \tag{33}$$

Notice that the JKO scheme has two terms: the first seeks to minimize the functional $\mathcal{F}(q)$ and the second is a regularization term, that penalize to stay close to $q_k$ in Wasserstein-2 distance as much as possible. It can be shown that as $h \to 0$, the limiting solution of (33) coincides with the path $\{q_\tau\}_{\tau\geq 0}$ defined by the continuity equation (32).

Throughout this work, we consider the KL divergence as the function, i.e, $\mathcal{F}_{\text{kl}}(q) = \text{KL}(q\|p)$. Then, the Liouville equation boils down to the Fokker-Planck equation

$$\frac{\partial q_\tau}{\partial\tau} = \text{div}\left(q_\tau\left(\nabla_{\mathbf{x}}\log q_\tau - \nabla_{\mathbf{x}}\log p\right)\right) \tag{34}$$

where the Wasserstein gradient is $\nabla_{W_2}\mathcal{F}_{\text{kl}}(q_\tau) = \nabla_{\mathbf{x}}\log(q_\tau/p)$ and the probability flow ODE follows (6). Lastly, although here we focus on the Wasserstein metric, we can consider other metrics that yield different flows (Chen et al., 2023). This is a future avenue to explore.

**Bures-Wasserstein gradient flow.** We now show that RED-diff (Mardani et al., 2024) can be derived from the gradient flow perspective when considering the flow in in the Bures-Wasserstein space $(\mathcal{BW}(\mathbb{R}^n), W_2)$, i.e., the subspace of the Wasserstein space consisting of Gaussian distributions.

The Wasserstein-2 distance between two Gaussian distributions $q = \mathcal{N}(\mu_q, \Sigma_q)$ and $p = \mathcal{N}(\mu_p, \Sigma_p)$ has a closed form,

$$W_2^2(q, p) = \|\mu_q - \mu_p\|_2^2 + \mathcal{B}^2(\Sigma_q, \Sigma_p),$$

where $\mathcal{B}^2(\Sigma, \Sigma_p) = \mathrm{tr}\left(\Sigma + \Sigma_p - 2\left(\Sigma^{\frac{1}{2}}\Sigma_p\Sigma^{\frac{1}{2}}\right)^{\frac{1}{2}}\right)$ is the squared Bures distance. By restricting the JKO scheme in (33) to the Bures-Wasserstein space, the authors in Lambert et al. (2022) showed that the discretization entails a solution given by the limiting curve $\{q_t : \mathcal{N}(\mu_t, \Sigma_t)\}_{t \geq 0}$. Therefore, the gradient flow of the KL divergence in the Bures-Wasserstein space boils down to the evolution of the means and covariance matrices of Gaussians, described by the following ODEs:

$$\frac{\mathrm{d}\mu_\tau}{\mathrm{d}\tau} = \mathbb{E}_{\mathbf{x} \sim q_\tau}\left[\nabla_\mathbf{x} \log \frac{p(\mathbf{x})}{q_\tau(\mathbf{x})}\right], \tag{35}$$

$$\frac{\mathrm{d}\Sigma_\tau}{\mathrm{d}\tau} = \mathbb{E}_{\mathbf{x} \sim q_\tau}\left[\left(\nabla_\mathbf{x} \log \frac{p(\mathbf{x})}{q_\tau(\mathbf{x})}\right)^T (\mathbf{x} - \mu_\tau)\right] + \mathbb{E}_{\mathbf{x} \sim q_\tau}\left[(\mathbf{x} - \mu_\tau)^T \nabla_\mathbf{x} \log \frac{p(\mathbf{x})}{q_\tau(\mathbf{x})}\right]. \tag{36}$$

In essence, RED-diff (and DreamFusion as well) follows the ODE corresponding to the mean with the diffusion as regularizer and the likelihood as measurement term. The key of the proposed methods is that the variational distribution uses the diffused trajectory (and therefore, the denoiser) as regularizer.

## F COMPARISON WITH OTHER PARTICLE METHODS USING DIFFUSION MODELS

### F.1 STEIN VARIATIONAL GRADIENT DESCENT (SVGD)

SVGD is a deterministic particle-based variational inference method (Liu, 2017), and is the core method of (Kim et al., 2023). Through the lens of gradient flow, it optimizes the same functional (KL divergence) but considers a different metric induced by the Stein operator (Duncan et al., 2023). In particular, given a pairwise repulsion as $\mathcal{R}(\mathbf{z}_{t,\tau}^{(1)}, \cdots, \mathbf{z}_{t,\tau}^{(N)}) = k\left(\mathbf{z}_{t,\tau}^{(i)}, \mathbf{z}_{t,\tau}^{(j)}\right)$, the update direction in SVGD for the particle $\mathbf{z}_{t,\tau}^{(i)}$ is given by

$$\sum_{j=1}^{N} k\left(\mathbf{z}_{t,\tau}^{(i)}, \mathbf{z}_{t,\tau}^{(j)}\right) \nabla_{\mathbf{z}_{t,\tau}^{(j)}} \log p\left(\mathbf{z}_{t,\tau}^{(j)}\right) - \nabla_{\mathbf{z}_{t,\tau}^{(j)}} k(\mathbf{z}_{t,\tau}^{(j)}, \mathbf{z}_{t,\tau}^{(j)}). \tag{37}$$

When comparing the gradient update (37) with (7)), the main difference resides in how is computed the gradient of the log target: while in SVGD, all the ensembles follow the same averaged gradient direction weighted by the similarity kernel function, in our method each particle uses its score direction. This difference in the score has important implications: SVGD suffers from the curse of dimensionality and poor exploration of the space in high-dimensional multimodal distributions, suffering the mode collapse phenomenon. Some alternatives have been proposed, like adding an annealing schedules (D'Angelo and Fortuin, 2021a; Ba et al., 2021), but the method still discovers only a few modes in multimodal distributions. Conversely, the repulsive method achieves a better exploration, and therefore, it finds more modes.

### F.2 PARTICLE GUIDANCE

In the context of sampling, a recent work termed Particle guidance (Corso et al., 2024) proposed a method to incorporate an interactive particle sampler based on diffusion. They proposed a guidance term that couples a set of particles to guide the backwards diffusion process towards different modes of the target distribution. Although they leverage a similar idea, the scope of our work is different. Here we focus on diffusion model in the context of distillation, and therefore, as a regularizer, which allows to deploy in constrained problems like inverse problems, or unconstrained cases like text-to-3D.

### F.3 COPULA MODELS

Copula models Nelsen (2006); Tran et al. (2015) are methods for defining variational distributions that incorporate dependencies between variables. While typically this is done for modeling the relationship between each dimension, in this case, we use the copula model to define dependencies between a batch of particles and promote diversity via repulsion.

## G VARIATIONAL SAMPLERS FOR INVERSE PROBLEMS

Variational inference methods Blei et al. (2017) are one of the most important techniques for sampling from intractable distributions. In a nutshell, VI defines a parametric distribution where the parameters are learned via stochastic gradient methods. A key difference between variational samplers and others, such as those based on the Monte Carlo Markov Chain, is that VI defines a parametric distribution. Thus, once the parameters are learned, we can easily generate samples.

In inverse problems, different variational samplers have been proposed in Knoll et al. (2011); Portilla et al. (2003); Kobler et al. (2017). The difference between these approaches lies in the definition of the variational distribution. For instance, Kobler et al. (2017) leverages the algorithm-unrolling framework, while Portilla et al. (2003) uses a mixture of Gaussians in the wavelet domain. More recently, the authors in Tonolini et al. (2020) proposed a two-stage semi-supervised method for learning variational samplers: a first stage that learns the forward model and a second stage that trains a conditional variational auto-encoder that learns to solve the inverse problem.

Related to our work, both Feng et al. (2023) and Mardani et al. (2024) combine variational inference with diffusion priors. Our method has three main differences with Mardani et al. (2024): 1) we define our variational inference problem in a latent space, allowing us to exploit the latent diffusion model, 2) we consider a multi-modal variational distribution with a repulsion term to couple the particles, enabling a better exploration, and 3) we decouple the data and prior term to handle the challenges of latent inversion. On the other hand, Feng et al. (2023) also considers a variational perspective using diffusion priors. First, they consider pixel-based diffusion models, which simplifies the formulation. Second, they incorporate the diffusion prior by computing the log probabilities: this requires solving the underlying ODE and estimating a divergence, which is computationally expensive. Lastly, they consider a normalizing flow as a variational distribution, i.e., they define a distribution $q_\phi(\mathbf{x}|\mathbf{y})$ parameterized by $\phi$; in particular, they consider RealNVP (Dinh et al., 2017). While this differs from our definition in (8), our framework allows incorporating a normalizing flow as a variational distribution. Exploring this type of parametric distribution opens an interesting research direction, where we define an amortized distribution that can generate solutions in only a few steps; we leave this as future work A pseudo-code version of this method is shown in Alg 3.

---

**Algorithm 3** Score-Based Diffusion Models as Principled Priors for Inverse Imaging Feng et al. (2023)

---

**Require:** $\mathbf{y}, f(.), s_{\boldsymbol{\theta}}(\mathbf{x}_t, t)$
  **for** $n = 1$ to $N_{iter}$ **do**
    Compute $\log p_\theta(\mathbf{x}_0) = \log p_T(\mathbf{x}_T) + \int_0^T \nabla \cdot \left( -\frac{1}{2}\beta(t)\mathbf{z}_t \mathrm{d}t - \beta(t)s_{\boldsymbol{\theta}}(\mathbf{x}_t, t) \right) \mathrm{d}t$ via ODE solver and divergence estimation
    Minimize $D_{\mathrm{KL}}\left( q_\phi \| p_\theta(\cdot \mid \mathbf{y}) \right)$
  **end for**
  **return** $q_\phi(\mathbf{x}|\mathbf{y})$

---

Lastly, it is important to note the connection of RLSD with plug-and-play methods Kamilov et al. (2023); Zhang et al. (2021). However, none of these works incorporate latent diffusion model as denoisers.

