# OpenReview forum: "Repulsive Latent Score Distillation for Solving Inverse Problems"
_ICLR.cc/2025/Conference — ICLR 2025 Poster_

### Official Review · Reviewer_Xthg · 2024-10-28

**Soundness:** 3
**Presentation:** 3
**Contribution:** 2
**Rating:** 6
**Confidence:** 4

**Summary:**

The paper proposes (Repulsive Latent Score Distillation) RLSD, a new method leveraging pre-trained diffusion models for inverse problems. RLSD uses a repulsive term in its variational sampling process, which aims to address mode collapse by promoting diversity. Experimental results suggest that RLSD may offer improvements in both quality and diversity for tasks like image inpainting.

**Strengths:**

The RLSD approach is novel, with a theoretically sound foundation in Wasserstein gradient flow and variational sampling. Additionally the repulsion term does appear to promote diversity.

**Weaknesses:**

While comparisons with a few models are presented, the paper does not address how RLSD performs against the broader set of models for inverse problems using diffusion prior methods. For example how about comparisons with DDNM, Pi-GDM, FPS, FPS-SMC, ect.? Without these comparisons it makes it difficult to gauge just how much of an improvement this approach is compared to similar approaches.

**Questions:**

How sensitive is RLSD to the initialization of particle positions?

In cases where diversity is essential, what strategies do you suggest to improve convergence to distinct modes?

Can RLSD achieve similar performance when the number of particles is reduced? If so, what modifications, if any, would be required?

---

> ### Author Response · Authors · 2024-11-21
>
> Thank you for reviewing our paper. We address your concerns below, and added to Appendix D in the pdf.
> ### **Q1: Additional comparison with baselines.**
> Following your suggestion, we have included additional comparisons with $\Pi$GDM and FPS-SMC for half-face inpainting and super-resolution ($\times$ 8).
> We report these additional results in Appendix D.2-D.4 and also here.
> Notice that both new baseline methods can be applied to _linear_ inverse problems, and in the case of $\Pi$GDM for some non-linear inverse problems like JPEG, but not to phase retrieval or HDR, limiting their application to challenging inverse problems.
>
> Table 5 (half-face inpanting)
> |**Sampler**|**PSNR [dB]↑**|**LPIPS↓**|**FID↓**|
> |-|-|-|-|
> |NonAug-RLSD (γ=50)|23.34|0.164|98.65|
> |NonRepuls-RLSD|**24.98**|_0.079_|**29.18**|
> |RLSD (γ=50)|24.69|0.111|31.41|
> |Hybrid-RLSD|_24.72_|0.096|30.48|
> |πGDM|23.74|0.077|33.8|
> |FPS-SMC|24.91|0.086|59.59|
>
> Table 6 (super-resolution)
> |**Sampler**|**PSNR [dB]↑**|**LPIPS↓**|**FID↓**|
> |-|-|-|-|
> |NonRepuls-RLSD|**28.4**|0.149|**65.42**|
> |πGDM|24.44|**0.128**|82.01|
> |FPS-SMC|27.36|0.21|120.49|
>
> Also, in response to Reviewer 2, we included ReSample (a recent method working with latent diffusion models) in all the experiments we have in the appendix.
> ### **Q2: How sensitive is RLSD to the initialization of particle positions?**
> This is a great point.
> We consider a random initialization (from a standard Gaussian distribution).
> While changing the initial point (by using other seeds or other distributions) renders different solutions, we saw that, on average, we get similar results in terms of diversity/quality.
>
> We tried other initializations, such as $f^{-1}\mathbf{y}$ (when $f$ is linear and invertible), but this is counterproductive for promoting diversity as we tend to start close to a local minimum, and it takes more time to explore the search space. This is particularly relevant in phase retrieval.
> ### **Q3: Strategies to improve convergence to distinct modes**
> Below, we give details of two strategies that might help to improve convergence.
>
> 1. **Adaptive $\gamma$**
>     - The simplest way is by increasing its value (strength of repulsion).
>     While this might keep the particles far away from each other, it can also generate a bad trajectory. In other words, it might diverge.
>     - A more sophisticated way is to define an adaptive $\gamma$.
>     Doing this, one could enforce a strong repulsion at the beginning and gradually decrease it to converge to a good mode.
>     While we did not include an experiment using an adaptive $\gamma$, the experiments in Appendix D.8 using repulsion in a  truncated interval go in this direction.
> 2. **Adaptive kernel**
>     - Use an adaptive kernel based on noise level. This resembles an annealing process, where the width of the RBF will depend on the noise level. By incorporating more information in the kernel's definition, the repulsion will help to improve convergence to distinct modes.
>     - Furthermore, if one has access to a dataset of points, we could aim to learn a kernel that promotes diversity.
> ### **Q4: Similar performance when the number of particles is reduced**
> This is an excellent question.
> Diversity is attained from the interaction between particles. To achieve good performance with a small number of particles, we can explore the different degrees of freedom in the design of this repulsion force. In particular, we can focus on and modify three aspects:
> 1. **Kernel function**.
> In the paper, we consider the radial basis function (RBF) to be a width that depends on the distance between particles.
> A possible modification would be using the adaptive kernel mentioned in point 2 in response to your Q3, either by rescaling the width to account for the small number of particles or using a learned kernel.
> 2. **Kernel domain**.
> The domain where the repulsion is computed is critical for having a well-defined repulsion.
> In this context, we use latent diffusion models + DINO to avoid the curse of dimensionality.
> Thus, this allows for the achievement of diversity with a relatively small number of particles (four).
> If we want to achieve more diversity, other domains might be explored, such as LPIPS; however, our ablation in Appendix D.10.2 suggests that LPIPS does not render a good repulsion for a small number of particles.
> 3. **Differentiable treatment**.
> We can use a hybrid approach, where only a few particles are affected by the repulsion force.
> In this case, some particles will promote diversity and others will converge to the modes that would have been discovered without repulsion.
> This additional degree of freedom (the number of particles affected by repulsion) can be useful in maintaining high performance even for varying numbers of total particles.
> We expand on this approach in the response to W1 and W2 of Reviewer 3.

---

> > ### Comment · Reviewer_Xthg · 2024-11-22
> >
> > Thank you for addressing my concerns and providing additional details and experimental results.
> > The added experiments, especially the baseline comparisons, provide stronger evidence for RLSD's contributions. However, I still believe that certain aspects, like adaptive mechanisms and the limitations of baselines, could benefit from further elaboration. With this being said, I am inclined to raise my score to 6.
> >
> > Thank you again for your thorough responses and efforts to improve the manuscript.

---

> ### Author Response · Authors · 2024-11-22
>
> Thank you for your positive comment and for considering our response in your assessment of our manuscript!

---

### Official Review · Reviewer_XHxy · 2024-10-29

**Soundness:** 2
**Presentation:** 3
**Contribution:** 2
**Rating:** 6
**Confidence:** 4

**Summary:**

This paper focus on solving two challenges in inverse problem solving with latent score distillation: mode collapse and latent space inversion. To solve mode collapse, the authors add a repulsive regularization loss to push different samples away from each other. To improve the latent space inversion, the authors define an augmented joint distribution $p(z_0, x_0 | y)$ and propose a two-step optimization algorithm. The authors demonstrate their model on FFHQ 512 dataset using the inpainting, phase retrieval and HDR tasks.

**Strengths:**

The paper presents a clear and well-structured introduction to the concepts of repulsive regularization and augmented variational distribution. The motivation behind these methods is well-founded, and the theoretical derivation is both detailed and easy to understand.

**Weaknesses:**

My major concern is that the effectiveness of the two proposed techniques in this paper does not seem to be fully supported by the experiments. The proposed techniques do not consistently yield good results under all conditions. In more detail:

1. Table 2,3,4 show that there is no single setting that can consistently perform the best (or almost the best) across all the three tasks in this paper. For example, NonRepuls-RLSD works the best in HDR, but it performs badly  (second-to-worst) in Phase retrieval. NonAug-RLSD shows best results in Phase-retrieval, but it performs bad in HDR and inpainting. And the full RLSD model has never achieved the best results across the three tasks.

2. While directly enforcing dissimilarity between particles can intuitively enhance their diversity, pushing particles apart may also drive them away from the true data manifold and affect the condition $y = f(x) + v$. Therefore, the strength of this term should be highly influenced  the landscape of the underlying distribution—specifically, whether it is indeed highly multimodal. This likely explains why incorporating repulsive regularization does not always yield improved results.

3. The two proposed techniques introduce several new hyperparameters that require tuning (for example, $N, \gamma, \rho, l$). While some quantitative results are presented in D7, a comprehensive qualitative ablation study examining the sensitivity of performance to different parameter selections might be needed.

**Questions:**

1. In the Sampling setup, the proposed method employs Stable Diffusion v2.1. Do the baseline methods (PLSD, latent RED-Diff, latent DPS) also use Stable Diffusion v2.1? It appears that in the original paper, PLSD used Stable Diffusion v1.5. Additionally, it may not be fair to directly compare methods using Stable Diffusion as a prior with those using pixel-level diffusion models as a prior, as the pixel-level diffusion models are smaller in size and trained on less data.

2. In the RLSD, N=4 particles are generated per noisy measurement. How are these four different results used to calculate the metrics reported in Tables 2, 3, and 4? Are the metrics calculated by selecting the best result among the four, or are they averaged across all four results?

**Details Of Ethics Concerns:**

No ethics concerns

---

> ### Author Response · Authors · 2024-11-21
>
> Thank you for your detailed review and constructive feedback. Please find the comments addressed below. All new experiments are in blue in Appendix D.1 to D.9 in the updated pdf.
> ## **Weaknesses**
> ### **W1: Full RLSD model does not consistently achieve the best results across all tasks**
> We thank the reviewer for raising this excellent point. The added diversity gained from the repulsion between particles can indeed affect the _average_ performance of the images. However, there are three important points to notice:
> 1. If instead of focusing on the average performance, we focus on the **performance achieved by the best image among the ensemble of particles**, then the conclusion differs in favor of the full RLSD; see Table 4. In particular, the best image generated by RLSD with repulsion ($\gamma = 50$) achieves better PSNR and LPIPS metrics than the best image generated when repulsion is ignored.
> This improvement can be explained as follows: while some modes obtained with RLSD might not be as good as NonRepuls-RLSD, the additional diversity obtained by repulsion can allow for the exploration of better modes.
>
> Table 4 (Box inpainting):
> |**Sampler**|**PSNR [dB]↑**|**LPIPS↓**|**FID↓**|
> |-|-|-|-|
> | PSLD (mean)|21.34|0.10|57.7|
> | PSLD (max)|22.72|0.082|57.7|
> | NonRepuls-RLSD(mean)|24.98|0.079|**29.18**|
> | NonRepuls-RLSD(max)|_25.82_| _0.071_|29.18|
> | RLSD (γ = 50)(mean)| 24.69|0.111|31.4|
> | RLSD (γ = 50)(max)|**25.84**|**0.069**|31.41|
>
> 2. Even if one is interested in a good average performance while boosting diversity, then one can further explore the hybrid approach mentioned in Section 5.1. More precisely, given N particles, we can have one or two that are not affected by repulsion, so that these will discover the same modes that the non-repulsive RLSD would. We can then use the remaining particles with repulsion to promote the discovery of (potentially better) modes. In this way, we can have an additional degree of freedom (the number of particles without repulsion) to interpolate between the non-repulsive and the full RLSD, enabling the user to find a sweet spot for their problem at hand.
> 3. Lastly, how much the gain in diversity affects the performance depends on intrinsic aspects of the problem at hand, which brings us to your next point.
> ### **W2: Strength of repulsion influence the landscape of the distribution**
> Indeed, this is true. For instance, in tasks such as phase retrieval (highly multimodal with discrete modes), how much strength is considered is fundamental to discovering different -- and good -- modes.
>
> Furthermore, if the strength is too high, the solution might diverge, as illustrated in Limitations (Appendix D.9). We believe the hybrid approach mentioned in the previous point can provide a well-balanced trade-off to guarantee a good solution while allowing exploration to promote diversity in other samples.
> ### **W3: Additional ablation w.r.t. hyperparemeters**
> Besides the ablation of $\gamma$ and $L$ in Appendix D.8.2, we added one comparing the number of particles $N$ in the repulsion (Figures 20, 21 in the updated pdf). Given 4 particles, we considered 3 scenarios:
> 1. All particles independent ($N=0$),
> 2. Two particles interacting, two independent ($N=2$),
> 3. All particles interacting ($N=4$).
>
> Additionally, we include an ablation for the unconstrained case in Appendix D.10.2.
> ## **Questions**
> ### **Q1: Version of Stable Diffusion and comparison with pixel-based models**
> We tried both versions of Stable Diffusion (v1.5 and v2.1) for RLSD and PSLD, and in each case, we chose the version that entails the best performance.
>
>   - PSLD: We tried with Stable Diffusion v2.1, but we did not get good results even after fine-tuning. Hence, we kept v1.5.
>   - RLSD: Achieved better performance with v2.1 (although it still outperforms PSLD when using v1.5).
>
> This analysis yields an important implementation remark to be made. PSLD's implementation does not use Hugging Face's diffuser library. Thus, we needed to adapt the code to run with Stable Diffusion v2.1. By contrast, our code leverages the diffuser library, making the code transparent to the version of Stable Diffusion and making the transition between different Stable Diffusion versions completely seamless.
>
> Regarding the comparison with pixel-space methods, there is no official implementation for FFHQ trained on 512 x 512 resolutions.
> While we agree that this might not be 100% fair, we believe this is one of the strengths of our method: it generates better and higher-resolution images, while being as fast as the samplers working with images of resolution 256 x 256.
> ### **Q2: How metrics are calculated**
> As explained in the response to your W1, we averaged across all particles to get the results with the exception of phase retrieval, where the average is not useful. However, considering the best image can also provides important insights. In particular, when looking at Table 4, we observe that RLSD is the best one.

---

> > ### Comment · Reviewer_XHxy · 2024-11-24
> >
> > I would like to thank the authors for their responses. However, the current results still do not sufficiently address my concerns regarding the effectiveness and general applicability of the proposed techniques:
> >
> > 1. The proposed method performs inconsistent across different tasks: The authors show that in box inpainting task, if best sample is picked, then RLSD works better than non-repulsive RLSD. However, in phase retrieval (Table 3), NonAug-RLSD achieves the best results, whereas in HDR (Table 4), NonRepuls-RLSD performs the best. This variability suggests that the users may have to introduce task-specific modifications to optimize performance for certain scenarios, raising concerns about the generalizability of the proposed techniques across diverse tasks.
> >
> > 2. Lack of Quantitative Ablation Studies: While I appreciate the qualitative results included in the paper, quantitative ablation studies are still absent for key hyper-parameters such as $\gamma$, $\rho$. Although qualitative visualizations provide insight into the effects of these parameters in specific cases, quantitative evaluations are crucial for assessing their overall impact. Quantitative analyses would shed light on the sensitivity of the proposed method to hyper-parameter tuning and provide insight into the complexity of tuning these parameters when applying the techniques to a new task.
> >
> > In all, while the proposed techniques are conceptually interesting, the experimental results presented in the paper are not sufficiently convincing. As such, I am inclined to maintain my current rating.

---

> ### Author Response · Authors · 2024-11-28
>
> Thanks again for your review.
> We run more experiments to further improve the results and address your concerns.
>
> ### **Q1: Full RLSD model does not consistently achieve the best results across all tasks**
> We reran the HDR and Phase Retrieval experiments with improved fine-tuning. Also, we ran inpainting with free mask for the ablation.
>  The results, especially for HDR, confirm the effectiveness of RLSD.
>
> **HDR**.
> We provide below the results for HDR. Table 1 is mean among four particles and Table 2 is with max among the four ones.
>
> Table 1 (HDR with mean among four particles)
> |**Sampler**|**PSNR [dB]↑**|**LPIPS↓**|**FID↓**|
> |-|-|-|-|
> |NonRepuls-RLSD|**27.46**|**0.088**|37.57|
> |RLSD($\gamma=50$)|27.39|0.089|37.32|
>
> Table 2 (HDR with max to choose the sample)
> |**Sampler**|**PSNR [dB]↑**|**LPIPS↓**|**FID↓**|
> |-|-|-|-|
> |NonRepuls-RLSD|27.72|0.087|37.57|
> |RLSD($\gamma=50$)|**28.05**|**0.086**|**37.32**|
>
> We observe that RLSD achieves higher PSNR and lower LPIPS when choosing the best sample among 4 particles, confirming its advantage in HDR reconstruction.
>
> **Inpainting Free Mask**.
> In free-mask inpainting, RLSD ($\gamma = 25$) also outperforms NonRepuls-RLSD (see corresponding Table 4 in Ablation $\gamma$ below). These findings align with our results for inpainting with box masks that we shared before, further demonstrating RLSD's robustness across different mask configurations.
>
> **Limitations**.
> Unlike HDR and inpainting, phase retrieval presents challenges where RLSD does not outperform NonRepuls-RLSD. In particular, neither approach achieves satisfactory results compared to the non-augmented case. We hypothesize that this is due to the limited information that provides the measurement compared to the other experiments, making the optimization w.r.t. $\mathbf{x}$ unstable.
>
> **Conclusion**.
> RLSD demonstrates superior performance over NonRepuls-RLSD in HDR and inpainting (both box mask and free mask). These results emphasize RLSD's effectiveness in solving diverse and challenging inverse problems.
>
> ### **Q2: Ablation**
>
> We conducted an additional ablation study to analyze the impact of the coupling parameter $\tilde{\rho}$ and the repulsion term $\gamma$ on inpainting with free mask.
> In addition, a qualitative result that illustrates the effect of $\tilde{\rho}$ has been included in Appendix D.8.5.
>
>
> **Ablation $\tilde{\rho}$**.
> The coupling parameter $\tilde{\rho}$ controls the strength of the prior during the optimization of $\mathbf{x}$.
> The results below indicate that increasing $\tilde{\rho}$ enhances diversity, but degrades performance metrics such as PSNR and LPIPS.
> Notice that $\tilde{\rho}=0.07$ is the one we used for the experiments in the main apper
>
> Table 3 (Ablation $\tilde{\rho}$ for inpainting with free-mask - $\gamma = 0$)
> |**$\tilde{\rho}$**|**PSNR [dB]↑**|**LPIPS↓**|**FID↓**|**Diversity↑**|
> |-|-|-|-|-|
> |0.03|29.94|0.069|25.57|0.001|
> |0.07|**30.03**|**0.064**|**22.62**|0.002|
> |0.12|28.6|0.065|23.25|**0.006**|
>
> Observations:
> - A smaller $\tilde{\rho}$ balances prior influence and optimization, resulting in higher performance (PSNR and LPIPS).
> - Increasing $\tilde{\rho}$ improves diversity but sacrifices reconstruction quality.
>
> **Ablation $\gamma$**.
> The repulsion term $\gamma$ controls the strength of the repulsion effect during sampling.
> The results are below
>
> Table 4 (Ablation $\gamma$ for inpainting with free-mask.)
> |**$\gamma$**|**PSNR [dB]↑**|**LPIPS↓**|**FID↓**|**Diversity↑**|
> |-|-|-|-|-|
> |0|30.03|0.064|22.62|0.004|
> |25|**30.04**|**0.064**|**22.21**|**0.005**|
> |50|30|0.064|22.31|0.005|
>
> Observations:
> - For this experiment, increasing $\gamma$ improves diversity, with diminishing returns for values above 25. However, as mentioned in the limitations (Appendix D.9), too much repulsion can have a negative impact.
> - RLSD with $\gamma = 25$ outperforms NonRepuls-RLSD in both diversity and performance metrics, highlighting the repulsion term's dual benefit. Thus, this configuration achieves the best balance, improving both diversity and performance compared to NonRepuls-RLSD
>
>
> **Summary of the ablation**
>
> - Increasing $\tilde{\rho}$ enhances diversity but degrades performance.
> - The repulsion term significantly improves diversity and can also boost performance.
> - In general, repulsion is more effective than increasing the coupling to improve diversity while maintaining good performance metrics.
> - RLSD is robust with respect to the hyper parameters, as in all the cases the performance is still good.

---

> > ### Comment · Reviewer_XHxy · 2024-11-28
> >
> > Thank you for the new results. I appreciate the extra experiments. May I ask why FID is not reported in the tables (given FID is reported in all other tables)?

---

> > > ### Author Response · Authors · 2024-11-28
> > >
> > > We are glad you appreciate the extra experiments. We didn’t include FID in these tables because we just wanted to showcase the advantages of RLSD, and the ablation. Therefore, we thought PSNR and LPIPS might be enough, but we agree that FID should be included to be consistent.
> > >
> > > We just added the FID metric in the tables in the previous comments, and we hope this additional metric helps to improve the results further. Lastly, please let us know if there is anything else that can help to improve our work.

---

> > > > ### Comment · Reviewer_XHxy · 2024-11-29
> > > >
> > > > Thanks for the reply. After reviewing the newly added experimental results, I have increased my score to 6.

---

> > > > > ### Author Response · Authors · 2024-11-29
> > > > >
> > > > > Thanks a lot for the great feedback and suggestions for improving the manuscript. We will incorporate these new results in the paper.

---

### Official Review · Reviewer_GZBS · 2024-10-31

**Soundness:** 3
**Presentation:** 3
**Contribution:** 3
**Rating:** 6
**Confidence:** 3

**Summary:**

This paper presents Repulsive Latent Score Distillation (RLSD), a method that addresses the challenges of mode collapse and latent space inversion in Score Distillation Sampling (SDS) for high-dimensional data. RLSD introduces a repulsion mechanism to promote diversity among solutions and an augmented variational distribution to disentangle latent and data spaces. Inspired by the Wasserstein gradient flow, RLSD balances computational efficiency, quality, and diversity in inverse problems like inpainting phase retrieval, and HDR.

**Strengths:**

I appreciate the overall strong contribution of this paper. It is well-written and easy to follow. The proposed method introduces a repulsion mechanism to enhance diversity among solutions and an augmented variational distribution to separate latent and data spaces. Experiments on both linear and nonlinear inverse problems demonstrate the method’s effectiveness.

**Weaknesses:**

1. A recent baseline published in ICLR 2024 [1], which also uses a latent-diffusion-based approach for inverse problem-solving, is missing.

2. I found the coverage of linear inverse problems to be limited. Previous works like DPS, PLSD and RED-DIFF typically include four types of linear inverse problems—such as inpainting, super-resolution, Gaussian deblurring, and motion deblurring—which would strengthen the analysis.


[1] Solving Inverse Problems with Latent Diffusion Models via Hard Data Consistency. Bowen Song, Soo Min Kwon, Zecheng Zhang, Xinyu Hu, Qing Qu, Liyue Shen. ICLR 2024.

**Questions:**

1. A recent baseline published in ICLR 2024 [1], which also uses a latent-diffusion-based approach for inverse problem-solving, is missing. Can you clarify the advantages of your method over this baseline? I suggest that a direct comparison would be beneficial.

2. The variety of linear inverse problems in the main text is limted. I reviewed the appendix for additional experiments on other tasks. Why not include these results in the main text? A fair and complete comparison with PSLD, Latent RED-Diff, RED-Diff, DPS, and [1] across various linear inverse problems would strengthen the analysis.

3. Since Algorithm 1 involves two optimizer steps, could you clarify the computational efficiency of the proposed method?


[1] Solving Inverse Problems with Latent Diffusion Models via Hard Data Consistency. Bowen Song, Soo Min Kwon, Zecheng Zhang, Xinyu Hu, Qing Qu, Liyue Shen. ICLR 2024.

---

> ### Author Response · Authors · 2024-11-21
>
> Thank you for reviewing our paper.
> We provide answers to your three questions below; you can find also in Appendix D.1 to D.7 in blue in the updated pdf.
> ### **Q1: Comparison with ReSample**
> We used their implementation, which only includes an implementation with LDM trained on FFHQ.
> For completeness, we also adapted the code to handle Stable Diffusion.
> We report quantitative results for half-face inpainting, and motion deblurring and random inpainting in the response to Q2.
>
> Table 1 (Half face inpainting):
> |**Sampler**|**PSNR [dB]↑**|**LPIPS↓**|**FID↓**|
> |-|-|-|-|
> |PSLD|22.72|0.082|57.7|
> |ReSample|19.44|0.2|146.68|
> |NonRepuls-RLSD|**24.98**|_0.079_|**29.18**|
> |RLSD (γ=50)|24.69|0.111|31.41|
> |Hybrid-RLSD|_24.72_|0.096|30.48|
>
> We can summarize the following findings when comparing with ReSample:
> 1. **Performance**. RLSD **performs better** than ReSample across the different inverse problems; ReSample only outperforms in one metric (LPIPS) in one experiment (rand. inpainting). It is worth saying that we also tried Stable Diffusion as a diffusion prior for ReSample, but the results were worse than those with LDM.
> 2. **Computational complexity**. RLSD is **computationally more efficient**. Specifically, ReSample involves an inner optimization within the sampling. Consequently, when using LDM (256x256 resolution), it takes 4 minutes in one A100 GPU to generate 1 image. Moreover, when we tried Stable Diffusion, it took 16 minutes (considering the same hyperparameter configuration). We hypothesize that ReSample takes a long time with Stable Diffusion due to the optimization step, where it might struggle to converge at every step. While this might be improved, it demonstrates that the method needs more fine-tuning than ours.
> On the other hand, our method, using Stable Diffusion as prior, can **generate 4 samples with just 200 steps in less than 4 minutes (without repulsion, 1 image per minute) and 5 minutes (with full repulsion)**.
>
> It is worth noting that ReSample performs particularly poorly in half-face inpainting. We fine-tuned hyperparameters (step size and the hard consistency step), but the results in Table 1 are the best we could get. We have contacted the authors of [1], and they told us that, for this task, there is a step in their algorithm that is not used. However, they haven't made that script publicly available. This example demonstrates that our method generalizes well to different inverse problems, while ReSample implementations might change based on the task.
> ### **Q2: Breadth of Inverse problems considered**
> We added all the missing baselines for motion deblurring and random inpainting. We report these results in Table 2 and 3. Overall, we experimented with deblurring, inpainting, super resolution (linear), and phase retrieval, HDR (non linear), covering a wide range of inverse problems.
>
> Table 2 (Motion Blurring):
> |**Sampler**|**PSNR [dB]↑**|**LPIPS↓**|**FID↓**|
> |-|-|-|-|
> |NonRepuls-RLSD|**30.52**|**0.093**|**56.79**|
> |RED-Diff|30.27|0.15|103.17|
> |DPS|24.7|0.22|90.45|
> |ReSample|26.82|0.115|72.74|
>
> Table 3 (Random inpainting (80\%)):
> |**Sampler**|**PSNR [dB]↑**|**LPIPS↓**|**FID↓**|
> |-|-|-|-|
> |NonRepuls-RLSD|**30.56**|0.073|**41.11**|
> |RED-Diff|28.55|0.074|62.87|
> |DPS|26.48|0.15|95.44|
> |ReSample|27.49|**0.062**|54.42|
>
> We did not include these inverse problems in the main text due to space limitations. Instead, we prioritized those with
> 1. A high level of ambiguity where diversity is important
> 2. Multi-modal non-linear inverse problems (phase retrieval).
> ### **Q3: Computational efficiency.**
> RLSD involves two optimization steps due to the augmentation.
> The optimization w.r.t. $\mathbf{x}$ is cheap: in particular, when the operator $f(.)$ is linear, the loss function is quadratic.
> This allows acceleration by computing the gradient in a closed-form expression.
> On the other hand, the optimization w.r.t. $\mathbf{z}$ is more expensive: we need to backpropagate through the decoder and compute the repulsion term (when used).
> Despite this, RLSD is more efficient than its baselines, as explained below.
>
> Compared to PSLD:
> - RLSD does not need to compute the score Jacobian.
> - RLSD can solve inverse problems with fewer steps (e.g., 200 steps for inpainting as shown in appendix) due to its plug-and-play formulation. In constrast, PSLD requires more steps due to its posterior sampling formulation.
>  - The repulsion term indeed increases the computational cost as we need to compute the gradients of the kernel w.r.t. particles; however, we can trade-off between number of particles and diversity/quality depending on the downstream task. Furthermore, as mentioned in Section 5.1 and in response to reviewer 3 (point W1), we can consider a hybrid approach, where only a subset of particles interact, reducing the computational cost.
>
> Compared to ReSample, RLSD do not have an inner optimization (the hard consistency step in [1]). As explained in the answer to your Q1, this yields a marked difference in running time.

---

> > ### Comment · Reviewer_GZBS · 2024-11-25
> >
> > I appreciate your answers to my questions. The additional analysis and experiments address my concerns. I decided to raise the rating to 6.

---

> > > ### Author Response · Authors · 2024-11-25
> > >
> > > Thank you for appreciating our response! If there is anything else that can help to improve the manuscript, we would be happy to respond/address.

---

### Official Review · Reviewer_DJhq · 2024-11-04

**Soundness:** 3
**Presentation:** 3
**Contribution:** 3
**Rating:** 6
**Confidence:** 4

**Summary:**

The paper introduces Repulsive Latent Score Distillation (RLSD), a novel variational posterior sampling framework designed to address key challenges faced by Score Distillation Sampling (SDS) in high-dimensional data, specifically mode collapse and latent space inversion. By leveraging an interactive particle-based variational approach with a repulsion mechanism, RLSD encourages diversity among solutions through a kernel-based regularization strategy. Additionally, the framework incorporates an augmented variational distribution that decouples the latent space from the data space, optimizing computational efficiency while preserving detail and quality. The authors validate RLSD through extensive experiments on high-resolution (512 × 512) inverse tasks, showcasing improvements in linear inverse problems like inpainting, deblurring and nonlinear inverse problems like phase retrieval using Stable Diffusion models.

**Strengths:**

1. In Table 1, the authors compared RLSD with all the relevant prior works in a very clear way, emphasizing the scalability over dimensions and linear / nonlinear inverse problems as well as the avoidance of score Jacobian computations.
2. The introduction of a repulsion regularization mechanism that leverages kernel-based similarity to prevent particles from collapsing into a single mode is a step forward. This approach promotes exploration of diverse solutions, a common challenge in high-dimensional generative tasks.
3. The presentation of this paper is good, the reference is relatively complete.
4. The paper provides robust experimental results across different types of inverse problems, including linear and nonlinear tasks. The diversity and quality trade-off demonstrated by RLSD is well-supported by quantitative metrics and visual comparisons.

**Weaknesses:**

1. While the proposed approach is well-executed, its novelty compared to existing methods could be more explicitly justified. I think using repulsive forces for diversity and variational formulations for latent space decoupling are not truly unique and novel, compared the existing framework of variational formulations.
2. For the experiment section, is it possible for you to add more discussion or results on the performance of RLSD with larger-scale datasets and higher-dimensional spaces to clarify practical usage limits?
3. For the related works section, can you include a part (or put in appendix) with the comparison to other variational samplers as well as the difference between their pseudo-codes?

**Questions:**

Please refer to the "weaknesses" section for questions.

**Details Of Ethics Concerns:**

Ethics concerns have been written clearly in the paper.

---

> ### Author Response · Authors · 2024-11-21
>
> We thank the reviewer for the positive comments about our paper and the constructive feedback.
> Please find the comments addressed below, and in Appendix E and G in blue in the main paper.
> ### **Q1: Novelty**
> We thank the reviewer for raising this point.
> Indeed, repulsion and augmentation have been studied in the past (see [1], [2]).
> Also, augmentation for inverse problems (decoupling) can be related to plug-and-play methods (we added details on this in Appendix G).
> However, as far as we know, nobody has **combined repulsion and augmentation** in the context of variational inference.
> Moreover, repulsion in the context of diffusion models has been applied only in a few works (see Appendix F in the paper), but none for solving inverse problems with diffusion priors.
> See also Appendix F for our description of how the proposed repulsion can be related to copula models.
>
> More concretely, our method is novel in the following aspects:
> 1. It proposes a variational approximation by leveraging both particle coupling (repulsion) and distribution augmentation.
>     This is the first work to **explicitly promote diversity for solving inverse problems**.
>  2. It proposes to solve inverse problems with latent diffusion model as an expressive prior in a more **efficient** way than previous methods such as ReSample (see the response to Reviewer 2) and PSLD. This markedly reduces the latency in image generation (see response to Q1 from Reviewer 2)
> ### **Q2: Large-scale, high-dimensional data and practical limitations**
> Thanks for the comment.
> Regarding the limitations of our method, we expanded the related discussion in Appendix D.9.
> Concerning the dimensionality of the data, it is constrained by the existing pretrained diffusion priors (currently, stable diffusion provides resolutions up to 512x512). This is still considered a high dimensional data, compared with existing methods that work with resolutions up to 256x256; see e.g., [5, 7]. Also, note that since our method works in the latent space, it is scalable to the dimension as all the diffusion operates in a low-dimensional kernel space. A comparison between different domains where the kernel is defined is in Appendix D.10.2.
>
> Notice that since diffusion and the diversity kernel works in the latest space, our method is scalable with respect to the dimension. A comparison between different domains where the kernel is defined is in Appendix D.10.2.
> ### **Q3: Comparison with other variational samplers**
> We added a new section in Appendix G (see the updated pdf for more details) with a comparison with previous variational samplers for solving inverse problems.
> To summarize, the new section includes:
>
> 1. A comparison with traditional variational samplers, such as [3], and more recent ones like [4].
> 2. We expanded on the discussion with two recent works that used diffusion priors within a variational framework [5] and [6].
>
>     Compared to [5], our method has three main differences:
>     1. While RED-Diff is formulated in the pixel domain, our formulation is in a **latent space**, allowing us to exploit the latent diffusion model
>     2. We consider a **multi-modal variational distribution** with a repulsion term instead of a Gaussian
>     3. We decouple the data and prior term to handle the challenges of latent inversion.
>
>      Compared to [6], there are also a few differences.
>
>    1. They also consider pixel-based diffusion models, which simplifies the formulation.
>    2. They incorporate the diffusion prior by computing the log probabilities: this requires solving the underlying ODE and estimating a divergence, which is computationally expensive.
>    3. They consider a normalizing flow as a variational distribution. While this differs from our particle-based variational distribution, our framework allows incorporating a normalizing flow as a variational distribution. Exploring this type of parametric distribution opens an interesting research direction, where we define an amortized distribution that can generate solutions in only a few steps and it relates to the modern one-step distillation; we leave this as future work.
>
> [1] D'Angelo, Francesco, and Vincent Fortuin. "Repulsive deep ensembles are bayesian." NeurIPS 2021
>
> [2] Wild, Veit David, et al. "A rigorous link between deep ensembles and (variational) Bayesian methods." NeurIPS 2024.
>
> [3] Kobler, Erich, et al. "Variational networks: connecting variational methods and deep learning." GCPR, 2017.
>
> [4] Tonolini, Francesco, et al. "Variational inference for computational imaging inverse problems." JMLR 2020.
>
> [5] Mardani, Morteza, et al. "A Variational Perspective on Solving Inverse Problems with Diffusion Models." ICLR 2024.
>
> [6] Feng, Berthy T., et al. "Score-based diffusion models as principled priors for inverse imaging." CVPR 2023.
>
> [7] Chung, Hyungjin, et al. "Diffusion Posterior Sampling for General Noisy Inverse Problems." ICLR 2023.

---

> > ### Comment · Reviewer_DJhq · 2024-11-25
> > **Official Comment**
> >
> > Thanks so much for your clear response. I think it answers my questions and I keep my original score.

---

> > > ### Author Response · Authors · 2024-11-25
> > >
> > > Thank you again! If there is anything else that can help to improve the manuscript, we would be happy to respond/address.

---

> ### Author Response · Authors · 2024-11-28
>
> We want to let you know that we ran new experiments that further showed the advantages of RLSD; you can find these results in the last response to reviewer 3.
>
> Besides these experiments, please let us know if there is something else that we can include to improve your impression of the manuscript. After addressing your original comments and the addition of these experiments, if there is any reason why you still think that the paper is only “marginally above the acceptance threshold”, we would like the opportunity to improve our manuscript further.

---

### Author Response · Authors · 2024-12-02
**General response and summary of contributions and modifications**

We want to thank all the reviewers for the great feedback. The quality of the reviews demonstrated the commitment of the reviewers and the time and effort spent on our work. Addressing all the different comments improved our work.

Before the end of the rebuttal period, we want to summarize the contributions of our work and the modifications/improvements we have made.

### Contributions
1. The paper is **well-written** and easy to follow (all Reviewers), and **overall a strong contribution** (Reviewer GZBS)
2. The motivation behind our method is **well-founded** (Reviewer XHxy) and with a theoretically sound foundation on Wasserstein gradient flow (Reviewer Xthg)
3. All Reviewers emphasized and appreciated our **two main contributions**: the **repulsion mechanism** to enhance diversity and **augmented variational distribution** for decoupling latent and data spaces.
4. **Robust experimental results** that support our work (Reviewers DJhq, GZBS)

### Modifications/improvements.
1. Further **discussions on related works** (variational sampler) and limitations. As expressed by Reviewer DJhq, we added a Appendix G with a discussion with variational samplers in the context of inverse problems. In addition, we expanded the analysis of limitations in Appendix D.10.2.
2. **Additional baselines**. As expressed by Reviewers GZBS and Xthg, we added new baselines ($\PiGDM, FPS-SMC, Re-Sample) and included RED-Diff and DPS in all the inverse problems (Reviewer GZBS) in all the experiments of the paper.
3. **Advantages of Repulsion for performance**. As expressed by Reviewer XHxy, we show that our method consistently outperforms (except Phase Retrieval) the baselines when using the best sample across particles while increasing diversity.
4. **Additional ablation**. As expressed by Reviewer XHxy, we added **quantitative** baseline experiments in addition to the qualitative ones from the original submission.
5. **Clarifications**. We discussed some clarifications of our work, such as sensibility to initialization, strategies to improve convergence (Reviewer Xthg), and how we compute the metrics (Reviewer XHxy).

Lastly, we thank all the reviewers for reading our rebuttal, engaging with the discussion, and raising the general score.

---

### Meta-Review · Area_Chair_VTZc · 2024-12-17

**Metareview:**

This paper introduces Repulsive Latent Score Distillation (RLSD), a method designed to overcome mode collapse and latent space inversion in Score Distillation Sampling (SDS) for high-dimensional data. RLSD incorporates a repulsion mechanism to enhance solution diversity and employs an augmented variational distribution to effectively disentangle the latent and data spaces. Inspired by the Wasserstein gradient flow, RLSD achieves a balance between computational efficiency, solution quality, and diversity, making it well-suited for inverse problems such as inpainting, phase retrieval, and HDR reconstruction.

All reviewers find the paper well-written, and the idea interesting and well evaluated.

**Additional Comments On Reviewer Discussion:**

The authors have conducted experiments with extra baselines for validating the results.

---

### Decision · Program_Chairs · 2025-01-22

Accept (Poster)